# Generalization Bound of Gradient Flow through Training Trajectory and Data-dependent Kernel

**Yilan Chen**[1], **Zhichao Wang**[2,3], **Wei Huang**[4,5], **Andi Han**[6,4],
**Taiji Suzuki**[7,4], **Arya Mazumdar**[1]

[1] University of California San Diego  [2] University of California Berkeley
[3] International Computer Science Institute  [4] RIKEN AIP
[5] The Institute of Statistical Mathematics  [6] The University of Sydney  [7] The University of Tokyo
yic031@ucsd.edu; zhichao.wang@berkeley.edu; wei.huang.vr@riken.jp;
andi.han@sydney.edu.au; taiji@mist.i.u-tokyo.ac.jp; arya@ucsd.edu;

## Abstract

Gradient-based optimization methods have shown remarkable empirical success, yet their theoretical generalization properties remain only partially understood. In this paper, we establish a generalization bound for gradient flow that aligns with the classical Rademacher complexity bounds for kernel methods-specifically those based on the RKHS norm and kernel trace-through a data-dependent kernel called the loss path kernel (LPK). Unlike static kernels such as NTK, the LPK captures the entire training trajectory, adapting to both data and optimization dynamics, leading to tighter and more informative generalization guarantees. Moreover, the bound highlights how the norm of the training loss gradients along the optimization trajectory influences the final generalization performance. The key technical ingredients in our proof combine stability analysis of gradient flow with uniform convergence via Rademacher complexity. Our bound recovers existing kernel regression bounds for overparameterized neural networks and shows the feature learning capability of neural networks compared to kernel methods. Numerical experiments on real-world datasets validate that our bounds correlate well with the true generalization gap.

## 1  Introduction

Gradient-based optimization lies at the heart of modern deep learning, yet the theoretical understanding of why these methods generalize so well is still incomplete. Classical bounds attribute the generalization of machine learning (ML) models to the complexity of the hypothesis class [62], which fails to explain the power of deep neural networks (NNs) with billions of parameters [31, 14]. Recent studies reveal that the training algorithm, data distribution, and network architecture together impose an implicit inductive bias, effectively restricting the vast parameter space to a much smaller "effective region" that improves the generalization ability [33, 48, 60, 28, 59, 21]. This observation motivates the need for *algorithm-dependent* generalization bounds—ones that capture how gradient-based dynamics carve out the truly relevant portion of the hypothesis class during training.

A variety of theoretical frameworks have been proposed to address this challenge. Algorithmic stability [16] bounds the generalization error by the stability of the learning algorithm. Hardt et al. [29] first proved the stability of stochastic gradient descent (SGD) for both convex and non-convex functions. However, these bounds are often data-independent, require decaying step sizes for non-convex objectives, and grow linearly with training time. Moreover, for non-convex functions, full-batch gradient descent (GD) is typically considered not uniformly stable [29, 18].

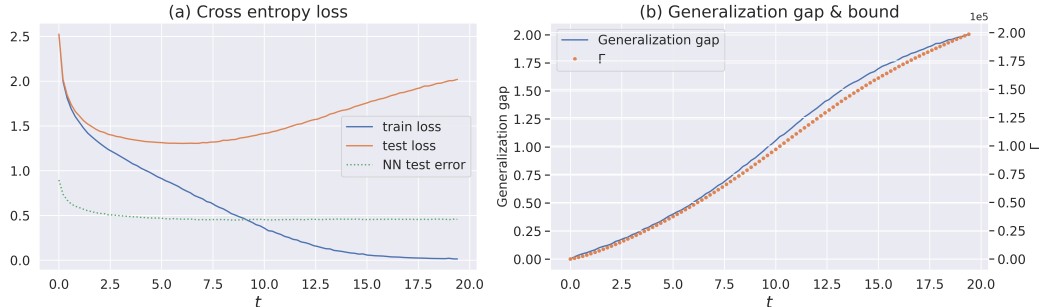

Figure 1: ResNet 18 trained by SGD on full CIFAR-10. (a) NN's training loss, test loss, and test error. (b) The generalization bound $\Gamma$ we derive in Theorem 5.4 correlates well with the true generalization gap.

Information-theoretic (IT) approaches [58, 66, 30] bound the expected generalization error with the mutual information between training data and learned parameters. To control this, researchers introduce noise into the learning process, employing techniques like stochastic gradient Langevin dynamics (SGLD) [57, 46, 64] or perturb parameters [47]. The PAC-Bayesian framework [39, 26] bounds the expected generalization error by the KL divergence between the model's posterior and prior distributions. To establish algorithm-dependent bounds, they also consider gradient descent with continuous noise like SGLD [45, 36], similar to the IT approach. But these noise-based approaches can diverge from SGD and their bounds can grow large when the noise variance is small.

In this work, we propose a novel perspective that combines *stability analysis of gradient flow* with *uniform convergence* tools grounded in Rademacher complexity. Specifically, we utilize a connection between loss dynamics and loss path kernel (LPK) proposed by Chen et al. [20]. By studying the stability of gradient flow, we show the concentration of LPKs trained with different datasets. This allows us to construct a function class explored by gradient flow with high probability while being substantially smaller than the full function class, leading to a tighter generalization bound. We summarize our main contributions as follows.

- We prove $O(1/n)$ stability for gradient flow on convex, strongly-convex, and non-convex losses, where $n$ is the number of training samples, and show that, as a result, LPKs concentrate tightly. This localization dramatically shrinks the effective hypothesis class and leads to a tighter bound than the previous result of Chen et al. [20].

- Using the above results, we derive a generalization bound for gradient flow that parallels classical Rademacher complexity bounds for kernel methods—specifically those involving the RKHS norm and kernel trace [10]—but adapts to the actual training trajectory. The generalization gap is controlled by an explicit term $\Gamma$, determined by the norm of the training loss gradients along the optimization trajectory. A similar bound is also proved for stochastic gradient flow.

- Our bound recovers known results in the NTK regime and kernel ridge regression, and exposes the feature-learning advantage of NNs. Extensive experiments on real-world datasets show that our bound $\Gamma$ correlates tightly with the true generalization gap (Fig. 1).

## 2  Related Work

**Generalization theory in deep learning.** Generalization has long been a central theme in deep learning theory, and various techniques have been proposed to study it. Beyond the algorithmic stability, PAC-Bayesian, and IT frameworks discussed earlier, Bartlett et al. [12] obtained tight bounds for the VC dimension of ReLU networks. Other works measure network capacity via norms, margins [10, 49, 11, 52], or sharpness-based metrics [50, 51, 4] to explain why deep NNs can generalize despite their large parameter counts.

**Algorithm-dependent generalization bound.** PAC-Bayesian, stability-based, and IT approaches can all yield algorithm-dependent bounds. Li et al. [36] combine PAC-Bayesian theory with algorithmic stability to derive an expected bound for SGLD that depends on the expected norm of the training loss gradient along the trajectory, which is similar to our bound, yet the bound blows up as the injected noise vanishes. Neu et al. [47] use mutual-information arguments to control the expected gap by the local gradient variance. Nikolakakis et al. [54] analyze the expected output stability to get an

expected generalization bound for full-batch GD on smooth loss that depends on the training loss gradient norm along the trajectory and the expected optimization error. In contrast, our result is a *high-probability* uniform-convergence bound whose leading term is not only tighter but is also straightforward to compute. Amir et al. [3] study generalization bounds for linear models trained by gradient descent on convex losses by constructing a function class centered around the expected trajectory, whereas our approach handles more general losses and models.

**Neural tangent kernel (NTK) and feature learning.** There is a line of work showing that over-parameterized NNs trained by GD converge to a global minimum and the trained parameters are close to their initialization [32, 25, 24, 2, 6] — so-called NTK regime. Arora et al. [5] study the generalization capacity of ultra-wide, two-layer NNs trained by GD and square loss, while Cao & Gu [17] examine the generalization of deep, ultra-wide NNs trained by SGD and logistic loss. Both establish generalization bounds of trained NNs, but NNs perform like a fixed kernel machine in this case. Going beyond the NTK regime, recent works [7, 15, 27, 9, 23, 43] have explored feature learning of NNs trained by GD for efficiently learning low-dimensional features which outperform the fixed kernel. Our approach is considerably more general and not restricted to overparameterized NNs. We present our bound in these two regimes as case studies in Sec. 6.

## 3 Notation and Preliminaries

Consider a supervised learning problem where the task is to predict an output variable in $\mathcal{Y} \subseteq \mathbb{R}^k$ using a vector of input variables in $\mathcal{X} \subseteq \mathbb{R}^d$. Let $\mathcal{Z} \triangleq \mathcal{X} \times \mathcal{Y}$. Denote the training set by $\mathcal{S} \triangleq \{z_i\}_{i=1}^n$ with $z_i \triangleq (x_i, y_i) \in \mathcal{Z}$. Assume each point is drawn i.i.d. from a distribution $\mu$. Let $\mathbf{X} = [x_1, \cdots, x_n] \in \mathbb{R}^{d \times n}$, $\mathbf{Y} = [y_1, \cdots, y_n] \in \mathbb{R}^{k \times n}$, and $\mathbf{Z} = [\mathbf{X}; \mathbf{Y}] \in \mathbb{R}^{(d+k) \times n}$.

We express a NN as $f(w, x) : \mathbb{R}^p \times \mathbb{R}^d \to \mathbb{R}^k$, where $w$ are its trainable parameters and $x$ is an input data. A learning algorithm $\mathcal{A} : \mathcal{Z}^n \mapsto \mathbb{R}^p$ takes a training set $\mathcal{S}$ and returns trained parameters $w$. The ultimate goal is to minimize the population risk $L_\mu(w) \triangleq \mathbb{E}_{z \sim \mu}[\ell(w, z)]$ where $\ell(w, z) \triangleq \ell(f(w, x), y)$ is a loss function. We assume $\ell(w, z) \in [0, 1]$. In practice, since the distribution $\mu$ is unknown, we instead minimize the empirical risk on the training set $\mathcal{S}$: $L_\mathcal{S}(w) \triangleq \frac{1}{n} \sum_{i=1}^n \ell(w, z_i)$. The *generalization gap* is defined as $L_\mu(w) - L_\mathcal{S}(w)$.

Below, we recall the definition of Rademacher complexity and a generalization upper bound.

**Definition 3.1** (Empirical Rademacher complexity $\hat{\mathcal{R}}_\mathcal{S}(\mathcal{G})$)**.** Let $\mathcal{F}$ be a hypothesis class of functions from $\mathcal{X}$ to $\mathbb{R}^k$. Let $\mathcal{G}$ be the set of loss functions associated with functions in $\mathcal{F}$, defined by $\mathcal{G} = \{g : (x, y) \to \ell(f(x), y), f \in \mathcal{F}\}$. The empirical Rademacher complexity of $\mathcal{G}$ with respect to sample $\mathcal{S} = \{z_1, \ldots, z_n\}$ is defined as $\hat{\mathcal{R}}_\mathcal{S}(\mathcal{G}) = \frac{1}{n} \mathbb{E}_\sigma \left[ \sup_{g \in \mathcal{G}} \sum_{i=1}^n \sigma_i g(z_i) \right]$, where $\sigma = (\sigma_1, \ldots, \sigma_n)$ is a sample of independent uniform random variables taking values in $\{+1, -1\}$, and $\mathbb{E}_\sigma$ is the expectation over $\sigma$ conditioned on all other random variables.

**Theorem 3.2** (Theorem 3.3 in [41])**.** *Let $\mathcal{G}$ be a family of functions mapping from $\mathcal{Z}$ to $[0, 1]$. Then for any $\delta \in (0, 1)$, with probability at least $1 - \delta$ over the draw of an i.i.d. sample set $\mathcal{S} = \{z_1, \ldots, z_n\}$, the following holds for all $g \in \mathcal{G}$: $\mathbb{E}_z[g(z)] - \frac{1}{n} \sum_{i=1}^n g(z_i) \leq 2\hat{\mathcal{R}}_\mathcal{S}(\mathcal{G}) + 3\sqrt{\frac{\log(2/\delta)}{2n}}$.*

### 3.1 Kernel Method and Loss Path Kernel

Recall that a kernel is a function $K : \mathcal{X} \times \mathcal{X} \to \mathbb{R}$ for which there exists a mapping $\Phi : \mathcal{X} \to \mathcal{H}$ into a reproducing kernel Hilbert space (RKHS) $\mathcal{H}$ such that $K(x, x') = \langle \Phi(x), \Phi(x') \rangle_\mathcal{H}$ for all $x, x' \in \mathcal{X}$, where $\langle \cdot, \cdot \rangle_\mathcal{H}$ denotes the inner product in $\mathcal{H}$. A function $K$ is a kernel if and only if it is symmetric and positive definite (Chapter 4 in [61]). A kernel machine $g : \mathcal{X} \to \mathbb{R}$ is a linear function in $\mathcal{H}$, and can be written as $g(x) = \langle \beta, \Phi(x) \rangle + b$, where its weight vector $\beta$ is a linear combination of the training points $\beta = \sum_{i=1}^n a_i \Phi(x_i)$ and $b$ is a constant bias. The RKHS norm of $g$ is $\|g\|_\mathcal{H} = \|\sum_{i=1}^n a_i \Phi(x_i)\| = \sqrt{\sum_{i,j} a_i a_j K(x_i, x_j)}$. The kernel machine with bounded RKHS norm has a classic Rademacher complexity bound as follows:

**Lemma 3.3** (Lemma 22 in [10])**.** *Denote a function class $\mathcal{F} = \{g(x) = \sum_{i=1}^n a_i K(x, x_i) : n \in \mathbb{N}, x_i \in \mathcal{X}, \sum_{i,j} a_i a_j K(x_i, x_j) \leq B^2\}$ for a constant $B > 0$. Then its Rademacher complexity is bounded by $\hat{\mathcal{R}}_\mathcal{S}(\mathcal{F}) \leq \frac{B}{n} \sqrt{\sum_{i=1}^n K(x_i, x_i)}$.*

Next we introduce the loss path kernel, which calculates the inner product of loss gradients and integrates along a given parameter path governed by (stochastic) gradient flows. Previous NTK theory cannot fully capture the training dynamics of NNs since trained parameters could move far away from initialization. The loss path kernel addresses this limitation by capturing the entire training trajectory.

**Definition 3.4** (Loss Path Kernel (LPK) $\mathsf{K}_T$ in [20]). Suppose the weights follow a continuous path $\boldsymbol{w}(t) : [0, T] \to \mathbb{R}^p$ in their domain with a starting point $\boldsymbol{w}(0) = \boldsymbol{w}_0$, where $T$ is a predetermined constant. This path is determined by the learning algorithm $\mathcal{A}$, the training set $\mathcal{S}$, and the training time $T$, i.e. $\boldsymbol{w}(t) = \mathcal{A}_t(\mathcal{S})$. We define the loss path kernel associated with the loss function $\ell(\boldsymbol{w}, \boldsymbol{z})$ along the path as

$$\mathsf{K}_T(\boldsymbol{z}, \boldsymbol{z}'; \mathcal{S}) \triangleq \int_0^T \langle \nabla_{\boldsymbol{w}} \ell(\mathcal{A}_t(\mathcal{S}), \boldsymbol{z}), \nabla_{\boldsymbol{w}} \ell(\mathcal{A}_t(\mathcal{S}), \boldsymbol{z}') \rangle \, \mathrm{d}t.$$

LPK is a valid kernel by definition. Intuitively, it measures the similarity between data points $\boldsymbol{z}$ and $\boldsymbol{z}'$ by comparing their loss gradients and accumulating over the training trajectory.

## 3.2 Loss Dynamics of Gradient Flow (GF) and Its Equivalence with Kernel Machine

Consider the GF dynamics (gradient descent with infinitesimal step size):

$$\frac{\mathrm{d}\boldsymbol{w}(t)}{\mathrm{d}t} = -\nabla_{\boldsymbol{w}} L_S(\boldsymbol{w}(t)) = -\frac{1}{n} \sum_{i=1}^n \nabla_{\boldsymbol{w}} \ell(\boldsymbol{w}(t), \boldsymbol{z}_i).$$

Chen et al. [20] showed that the loss of the NN at a certain fixed time is a kernel machine with LPK plus the loss function at initialization: $\ell(\boldsymbol{w}_T, \boldsymbol{z}) = \sum_{i=1}^n -\frac{1}{n} \mathsf{K}_T(\boldsymbol{z}, \boldsymbol{z}_i; \mathcal{S}) + \ell(\boldsymbol{w}_0, \boldsymbol{z})$. Here, the LPK is a data-dependent kernel that depends on the training set $\mathcal{S}$. Using this equivalence, define the following set of LPKs and the function class of the loss function

$$\mathcal{K}_T \triangleq \{\mathsf{K}_T(\cdot, \cdot; \mathcal{S}') : \mathcal{S}' \in \mathrm{supp}(\mu^{\otimes n}), \frac{1}{n^2} \sum_{i,j} \mathsf{K}_T(\boldsymbol{z}'_i, \boldsymbol{z}'_j; \mathcal{S}') \leq B^2\},$$

$$\mathcal{G}_T \triangleq \left\{ \ell(\mathcal{A}_T(\mathcal{S}'), \boldsymbol{z}) = \sum_{i=1}^n -\frac{1}{n} \mathsf{K}(\boldsymbol{z}, \boldsymbol{z}'_i; \mathcal{S}') + \ell(\boldsymbol{w}_0, \boldsymbol{z}) : \mathsf{K}(\cdot, \cdot; \mathcal{S}') \in \mathcal{K}_T \right\}, \tag{1}$$

where $B > 0$ is some constant, $\mathcal{S}' = \{\boldsymbol{z}'_1, \ldots, \boldsymbol{z}'_n\}$, $\mu^{\otimes n}$ is the joint distribution of $n$ i.i.d. samples drawn from $\mu$, $\mathrm{supp}(\mu^{\otimes n})$ is the support set of $\mu^{\otimes n}$, and $\mathcal{A}_T(\mathcal{S}')$ is the parameters obtained by GF algorithm at time $T$ and trained with $\mathcal{S}'$. Then Chen et al. [20] derived the following generalization bound:

$$\hat{\mathcal{R}}_{\mathcal{S}}(\mathcal{G}_T) \leq \frac{B}{n} \sqrt{\sup_{\mathsf{K}(\cdot, \cdot; \mathcal{S}') \in \mathcal{K}_T} \mathrm{Tr}(\mathsf{K}(\mathbf{Z}, \mathbf{Z}; \mathcal{S}')) + \sum_{i \neq j} \Delta(\boldsymbol{z}_i, \boldsymbol{z}_j)},$$

where $\Delta(\boldsymbol{z}_i, \boldsymbol{z}_j) = \frac{1}{2} \left[ \sup_{\mathsf{K}(\cdot, \cdot; \mathcal{S}') \in \mathcal{K}_T} \mathsf{K}(\boldsymbol{z}_i, \boldsymbol{z}_j; \mathcal{S}') - \inf_{\mathsf{K}(\cdot, \cdot; \mathcal{S}') \in \mathcal{K}_T} \mathsf{K}(\boldsymbol{z}_i, \boldsymbol{z}_j; \mathcal{S}') \right]$. However, the above bound suffers from several limitations: 1) It involves a supremum over an infinite family of LPKs, making it intractable to compute in practice; 2) The term $\sum_{i \neq j} \Delta(\boldsymbol{z}_i, \boldsymbol{z}_j)$ can be as large as $O(n^2)$ in the worst case, leading to a loose bound; 3) The bound must be evaluated on datasets distinct from the training set, limiting its practical applicability. In this paper, we use the stability property of GF to substantially reduce the size of the function class, resulting in a significantly tighter generalization bound that depends only on the training set. Our new bound matches the classical kernel method bound in Lemma 3.3, but instead of relying on a fixed kernel, it utilizes the *data-dependent* loss path kernel. Adapting to the data and algorithm, this learned kernel can outperform static kernels in traditional methods, thereby achieving improved generalization performance.

## 4 Uniform Stability of Gradient Flow and Concentration of LPKs

In this section, we show that the GF is uniformly stable. This uniform stability property implies the LPK concentration and connects LPKs trained from different datasets. Instead of transforming the stability to a generalization bound directly, we then combine the stability analysis with uniform convergence via Rademacher complexity to get a data-dependent bound in Sec. 5.

## 4.1 Uniform Stability of Gradient Flow

**Definition 4.1** (Uniform argument stability [16, 13])**.** A randomized algorithm $\mathcal{A}$ is $\epsilon_n$-uniformly argument stable if for all datasets $\mathcal{S}, \mathcal{S}^{(i)} \in \mathcal{Z}^n$ such that they differ by at most one data point, we have $\mathbb{E}_{\mathcal{A}} \left[ \left\| \mathcal{A}(\mathcal{S}) - \mathcal{A}(\mathcal{S}^{(i)}) \right\| \right] \leq \epsilon_n$, where the expectation is taken over the randomness of $\mathcal{A}$.

Here we consider the uniform argument stability, which can be easily transformed to uniform stability with respect to loss if the loss function is Lipschitz. In this paper, we mainly consider full-batch GF so there is no randomness in $\mathcal{A}$. To analyze the GF dynamics and LPKs, we make the following standard assumptions.

**Assumption 4.2.** Assume $\ell(\boldsymbol{w}, \cdot)$ is $L$-Lipschitz and $\beta$-smooth with respect to $\boldsymbol{w}$, that is, $\|\ell(\boldsymbol{w}, \cdot) - \ell(\boldsymbol{w}', \cdot)\| \leq L \|\boldsymbol{w} - \boldsymbol{w}'\|$ and $\|\nabla_{\boldsymbol{w}}\ell(\boldsymbol{w}, \cdot) - \nabla_{\boldsymbol{w}}\ell(\boldsymbol{w}', \cdot)\| \leq \beta \|\boldsymbol{w} - \boldsymbol{w}'\|$.

Let $\mathcal{S}$ and $\mathcal{S}^{(i)}$ be two datasets that differ only in the $i$-th data point. We prove the following stability results of GF for convex, strongly convex (S.C.), and non-convex losses. Similar stability results of GD (for convex case) and SGD were proved in [13, 29].

**Lemma 4.3.** *Under Assumption 4.2, for any two data sets $\mathcal{S}$ and $\mathcal{S}^{(i)}$, let $\boldsymbol{w}_t = \mathcal{A}_t(\mathcal{S})$ and $\boldsymbol{w}'_t = \mathcal{A}_t(\mathcal{S}^{(i)})$ be the parameters trained from same initialization $\boldsymbol{w}_0 = \boldsymbol{w}'_0$, then*

$$\|\boldsymbol{w}_t - \boldsymbol{w}'_t\| \leq \begin{cases} \frac{2L}{\gamma n}, & L_S(\boldsymbol{w}) \text{ is } \gamma\text{-S.C.,} \\ \frac{2Lt}{n}, & L_S(\boldsymbol{w}) \text{ is convex,} \\ \frac{2L}{\beta n}(e^{\beta t} - 1), & L_S(\boldsymbol{w}) \text{ is non-convex.} \end{cases}$$

For convex losses, uniform stability increases linearly with $T$. For strongly convex losses, it holds without increasing with training time. Unfortunately, for non-convex losses, the bound exponentially increases with time $T$ in the worst case, leading to an exponential stability generalization bound. Our Theorem 5.2 avoids this case by combining stability analysis with Rademacher complexity.

For non-convex loss, Hardt et al. [29] obtain $O(T/n)$ stability bound of SGD with decayed learning rate $\eta = c/t$, which is equivalent to training $c \ln T$ time in our case since $\sum_t c/t \approx c \ln T$. In our case, using a learning rate of $\eta = 1/\beta(t+1)$ will allow us to have $\|\boldsymbol{w}_T - \boldsymbol{w}'_T\| = \frac{2LT}{\beta n}$ [1].

## 4.2 Concentration of LPKs under Stability

We now derive useful concentration properties of LPKs using uniform stability. These properties will be used when defining the function class explored by GF and proving the generalization bound. First of all, one can show that the LPK concentrates for a fixed pair of $\boldsymbol{z}, \boldsymbol{z}'$.

**Lemma 4.4.** *Under Assumption 4.2, for any fixed $\boldsymbol{z}, \boldsymbol{z}'$, with probability at least $1 - \delta$ over the randomness of $\mathcal{S}'$,*

$$\left| \mathsf{K}_T(\boldsymbol{z}, \boldsymbol{z}'; \mathcal{S}') - \mathop{\mathbb{E}}_{\mathcal{S}'} \mathsf{K}_T(\boldsymbol{z}, \boldsymbol{z}'; \mathcal{S}') \right| \leq \begin{cases} \frac{4L^2\beta T}{\gamma}\sqrt{\frac{\ln\frac{2}{\delta}}{2n}}, & L_S(\boldsymbol{w}) \text{ is } \gamma\text{-S.C.,} \\ 2L^2\beta T^2\sqrt{\frac{\ln\frac{2}{\delta}}{2n}}, & L_S(\boldsymbol{w}) \text{ is convex,} \\ \frac{4L^2}{\beta}(e^{\beta T} - \beta T - 1)\sqrt{\frac{\ln\frac{2}{\delta}}{2n}}, & L_S(\boldsymbol{w}) \text{ is non-convex.} \end{cases}$$

Next, using a stability argument and Chernoff bound, we are able to bound the difference between $\sum_{i=1}^{n} \mathsf{K}_T(\boldsymbol{z}_i, \boldsymbol{z}_i; \mathcal{S}')$ and $\sum_{i=1}^{n} \mathsf{K}_T(\boldsymbol{z}_i, \boldsymbol{z}_i; \mathcal{S})$.

**Lemma 4.5.** *Under Assumption 4.2, for two datasets $\mathcal{S}$ and $\mathcal{S}'$, with probability at least $1 - \delta$ over the randomness of $\mathcal{S}$ and $\mathcal{S}'$,*

$$\left| \sum_{i=1}^{n} \mathsf{K}_T(\boldsymbol{z}_i, \boldsymbol{z}_i; \mathcal{S}) - \sum_{i=1}^{n} \mathsf{K}_T(\boldsymbol{z}_i, \boldsymbol{z}_i; \mathcal{S}') \right| \leq \begin{cases} \tilde{O}(T\sqrt{n}), & L_S(\boldsymbol{w}) \text{ is } \gamma\text{-S.C.,} \\ \tilde{O}(T^2\sqrt{n}), & L_S(\boldsymbol{w}) \text{ is convex,} \\ \tilde{O}(e^T\sqrt{n}), & L_S(\boldsymbol{w}) \text{ is non-convex.} \end{cases}$$

---

[1]However, training with a decayed learning rate may not converge and requires to change the definition of the LPK. Therefore, we stick to the constant learning rate.

Table 1: The rate of $\epsilon$ in Theorem 5.2 under different training time $T$ scales. Boldface indicates the cases where $\Gamma$ computed by (3) is the dominant term compared with $\epsilon$.

| $T$ | $O(1)$ | $O(\ln \sqrt{n})$ | $O(\sqrt{n})$ | $O(n)$ |
|---|---|---|---|---|
| S.C. | $\tilde{O}(n^{-3/4})$ | $\tilde{O}(n^{-3/4})$ | $\tilde{O}(n^{-1/2})$ | $\tilde{O}(n^{-1/4})$ |
| Convex | $\tilde{O}(n^{-3/4})$ | $\tilde{O}(n^{-3/4})$ | $O(n^{-1/4})$ | $O(1)$ |
| Non-convex | $\tilde{O}(n^{-3/4})$ | $\tilde{O}(n^{-1/2})$ | $O(n^{-1/4})$ | $O(1)$ |

# 5 Main Results

## 5.1 Generalization Bound of Gradient Flow (GF)

With the above preparations, we are ready to prove our generalization bound. We define the loss function class $\mathcal{G}_T$ as in (1) at time $T$ by constraining the LPK class $\mathcal{K}_T$ as follows

$$\mathcal{K}_T \triangleq \left\{ \mathsf{K}_T(\cdot, \cdot; \mathcal{S}') : \frac{1}{n^2} \sum_{i,j} \mathsf{K}_T(z_i', z_j'; \mathcal{S}') \leq B^2, \mathcal{S}' \in \mathbb{S}' \subseteq \mathrm{supp}(\mu^{\otimes n}), \sup_{z,z'} |\mathsf{K}_T(z, z'; \mathcal{S}')| \leq \Delta \right\}.$$

where $B, \Delta > 0$ are some constants and $\mathbb{S}'$ is a subset of $\mathrm{supp}(\mu^{\otimes n})$. Note this function class includes $\ell(\mathcal{A}_T(\mathcal{S}), z)$ if the conditions are satisfied on $\mathcal{S}$. For example, the first condition is satisfied if $\frac{1}{n^2} \sum_{i,j} \mathsf{K}_T(z_i, z_j; \mathcal{S}) \leq B^2$. For this function class, we can improve the Rademacher complexity below since the conditions in $\mathcal{K}_T$ significantly reduce the size of the function class.

**Lemma 5.1.** *Recall Definition 3.1 for $\hat{\mathcal{R}}_\mathcal{S}(\mathcal{G}_T)$, we have*

$$\hat{\mathcal{R}}_\mathcal{S}(\mathcal{G}_T) \leq \frac{B}{n} \sqrt{\sup_{\mathsf{K}_T(\cdot,\cdot;\mathcal{S}') \in \mathcal{K}_T} \sum_{i=1}^{n} \mathsf{K}_T(z_i, z_i; \mathcal{S}') + 4\Delta\sqrt{6n \ln 2n} + 8\Delta}.$$

As we have shown above, the conditions in the function class are satisfied with $B$ being some data-dependent quantity, and the trace term can be bounded as in Lemma 4.5. With a covering argument, we can prove our main result of the generalization bound for GF dynamics.

**Theorem 5.2.** *Denote by $\Gamma \triangleq \frac{2}{n^2} \sqrt{\sum_{i=1}^{n} \sum_{j=1}^{n} \mathsf{K}_T(z_i, z_j; \mathcal{S})} \sqrt{\sum_{i=1}^{n} \mathsf{K}_T(z_i, z_i; \mathcal{S})}$. Under Assumption 4.2, with probability at least $1 - \delta$ over the randomness of $\mathcal{S}$,*

$$L_\mu(\mathcal{A}_T(\mathcal{S})) - L_\mathcal{S}(\mathcal{A}_T(\mathcal{S})) \leq \Gamma + \epsilon + 3\sqrt{\frac{\ln(4n/\delta)}{2n}}, \tag{2}$$

*where* $\epsilon = \begin{cases} \tilde{O}\left(\frac{\sqrt{T}}{n^{\frac{3}{4}}}\right), & S.C., \\ \min\left\{\tilde{O}\left(\frac{T}{n^{\frac{3}{4}}}\right), O\left(\sqrt{\frac{T}{n}}\right)\right\}, & convex, \\ \min\left\{\tilde{O}\left(\frac{e^{\frac{T}{2}}}{n^{\frac{3}{4}}}\right), O\left(\sqrt{\frac{T}{n}}\right)\right\}, & non\text{-}convex. \end{cases}$

We now study which term in (2) dominates the bound in Theorem 5.2. We summarize the rate of $\epsilon$ for different training scaling of $T$ in Table 1. A rough analysis implies that the first term $\Gamma$ in the bound can be upper bounded by $O\left(L\sqrt{T/n}\right)$. In many cases, $\Gamma$ may not achieve this upper bound; Sec. 6 shows $\Gamma$ typically grows sub-linearly for $T$ if the training loss converges sufficiently fast.

*Remark* 5.3 (**Leading order for non-convex case**). For the non-convex case, when $T = O(1)$, $\epsilon = \tilde{O}(n^{-3/4})$ and when $T = O(\ln \sqrt{n})$, $\epsilon = \tilde{O}(n^{-1/2})$. In these cases, $\epsilon$ has a faster-decreasing rate compared with other terms. When $T = \Omega(\ln \sqrt{n})$, $\epsilon = O(\sqrt{T/n})$ which has a rate similar to $\Gamma$. Especially, when the loss is non-convex but satisfies the Polyak-Łojasiewicz (PL) condition with parameter $\alpha$, $L_\mathcal{S}(w_t) - L_\mathcal{S}(w^*) \leq e^{-\alpha t}(L_\mathcal{S}(w_0) - L_\mathcal{S}(w^*))$, $T = O(\frac{1}{\alpha} \ln \sqrt{n})$ is sufficient to achieve $O(1/\sqrt{n})$ optimization error.

Our results show that the generalization ability of GF is mainly affected by the first term $\Gamma$, which can also be rewritten as

$$\Gamma = \frac{2}{n}\sqrt{L_{\mathcal{S}}(\boldsymbol{w}_0) - L_{\mathcal{S}}(\boldsymbol{w}_T)}\sqrt{\sum_{i=1}^{n}\int_0^T \|\nabla_{\boldsymbol{w}}\ell(\boldsymbol{w}_t, \boldsymbol{z}_i)\|^2\, dt}, \qquad (3)$$

due to the definition of LPK and $\frac{dL_{\mathcal{S}}(\boldsymbol{w}_t)}{dt} = \nabla_{\boldsymbol{w}}L_{\mathcal{S}}(\boldsymbol{w}_t)^\top \frac{d\boldsymbol{w}_t}{dt} = -\|\nabla_{\boldsymbol{w}}L_{\mathcal{S}}(\boldsymbol{w}_t)\|^2$. This bound matches the Radamecher bound of the classic kernel methods in Lemma 3.3. In $\Gamma$, $\sqrt{\frac{1}{n^2}\sum_{i=1}^{n}\sum_{j=1}^{n}\mathsf{K}_T(\boldsymbol{z}_i, \boldsymbol{z}_j; \mathcal{S})}$ serves as the RKHS norm of the kernel, while $\sum_{i=1}^{n}\mathsf{K}_T(\boldsymbol{z}_i, \boldsymbol{z}_i; \mathcal{S})$ is the trace of the kernel. The RKHS norm in our setting remains below 1 due to the bounded loss.

Unlike a kernel method with a fixed kernel, GF learns a *data-dependent* kernel LPK, thus adapting the underlying feature map to the training set. Consequently, our bound can surpass the fixed-kernel scenario because the gradient norms $\|\nabla_{\boldsymbol{w}}L_{\mathcal{S}}(\boldsymbol{w}_t)\|, \|\nabla_{\boldsymbol{w}}\ell(\boldsymbol{w}_t, \boldsymbol{z}_i)\|$ shrink during training—whereas a fixed kernel's bound remains static. Moreover, the RKHS norm here stays below 1, in contrast to fixed-kernel methods, whose RKHS norm may grow with the sample size or dimensionality (see Corollary 6.2). Overall, our bound highlights that a more favorable optimization landscape and faster convergence can promote stronger generalization.

## 5.2 Generalization Bound of Stochastic Gradient Flow (SGF)

Above, we derived a generalization bound for NNs trained from full-batch GF. Here we extend our analysis to SGF and derive a corresponding generalization bound. To start with, we recall the dynamics of SGF (SGD with infinitesimal step size):

$$\frac{d\boldsymbol{w}_t}{dt} = -\nabla_{\boldsymbol{w}}L_{\mathcal{S}_t}(\boldsymbol{w}_t) = -\frac{1}{m}\sum_{i\in\mathcal{S}_t}\nabla_{\boldsymbol{w}}\ell(\boldsymbol{w}_t, \boldsymbol{z}_i) \qquad (4)$$

where $\mathcal{S}_t \subseteq \{1, \ldots, n\}$ is the indices of batch data used in time interval $[t, t+1]$ and $|\mathcal{S}_t| = m$ is the batch size. Define $\mathsf{K}_{t,t+1}(\boldsymbol{z}, \boldsymbol{z}'; \mathcal{S}) = \int_t^{t+1}\langle \nabla_{\boldsymbol{w}}\ell(\boldsymbol{w}_t, \boldsymbol{z}), \nabla_{\boldsymbol{w}}\ell(\boldsymbol{w}_t, \boldsymbol{z}')\rangle\, dt$ to be the LPK over time interval $[t, t+1]$.

**Theorem 5.4.** *Under Assumption 4.2, for a fixed sequence $\mathcal{S}_0, \ldots, \mathcal{S}_{T-1}$, with probability at least $1 - \delta$ over the randomness of $\mathcal{S}$, the generalization gap of SGF defined by* (4) *is upper bounded by*

$$L_\mu(\mathcal{A}_T(\mathcal{S})) - L_{\mathcal{S}}(\mathcal{A}_T(\mathcal{S})) \leq \frac{2}{n}\sum_{t=0}^{T-1}\sqrt{\frac{1}{m^2}\sum_{i,j\in\mathcal{S}_t}\mathsf{K}_{t,t+1}(\boldsymbol{z}_i, \boldsymbol{z}_j; \mathcal{S})}\sqrt{\sum_{i=1}^{n}\mathsf{K}_{t,t+1}(\boldsymbol{z}_i, \boldsymbol{z}_i; \mathcal{S})} + \tilde{O}(\frac{T}{\sqrt{n}}).$$

Similarly, we define the first term as $\Gamma$ for the SGF case. When $\mathcal{S}_t = \mathcal{S}$, SGF becomes GF and the bound becomes similar to (2). This bound can be extended to any random sampling algorithm by taking the expectation over the randomness of the algorithm.

# 6 Case Study

## 6.1 Overparameterized Neural Network under NTK Regime

The NTK associated with the NN $f(\boldsymbol{w}, \boldsymbol{x})$ at $\boldsymbol{w}$ is defined by $\hat{\Theta}(\boldsymbol{w}; \boldsymbol{x}, \boldsymbol{x}') = \nabla_{\boldsymbol{w}}f(\boldsymbol{w}, \boldsymbol{x})\nabla_{\boldsymbol{w}}f(\boldsymbol{w}, \boldsymbol{x}')^\top \in \mathbb{R}^{k\times k}$. Since the LPK has a natural connection with NTK, our bound $\Gamma$ in (3) can be calculated using the NTK during the training: $\Gamma = \frac{2}{n}\sqrt{L_{\mathcal{S}}(\boldsymbol{w}_0) - L_{\mathcal{S}}(\boldsymbol{w}_T)}\sqrt{\sum_{i=1}^{n}\int_0^T \nabla_f\ell(\boldsymbol{w}_t, \boldsymbol{z}_i)\hat{\Theta}(\boldsymbol{w}_t; \boldsymbol{x}_i, \boldsymbol{x}_i)\nabla_f\ell(\boldsymbol{w}_t, \boldsymbol{z}_i)dt}$. When the output dimension $k = 1$ and using a mean-square loss $L_{\mathcal{S}}(\boldsymbol{w}_t) = \frac{1}{2n}\|f(\boldsymbol{w}_t, \mathbf{X}) - \boldsymbol{y}\|^2$, as previous work [25, 24] showed, as long as the smallest eigenvalue of NTK is lower bounded from 0, the training loss enjoys an exponential convergence, $\|f(\boldsymbol{w}_t, \mathbf{X}) - \boldsymbol{y}\|^2 \leq e^{-\frac{2\lambda_{\min}}{n}t}\|f(\boldsymbol{w}_0, \mathbf{X}) - \boldsymbol{y}\|^2$. In this setting, the generalization can be upper bounded by the condition number of the NTK as follows.

**Corollary 6.1.** *Suppose that $\lambda_{max}(\hat{\Theta}(\boldsymbol{w}_t; \mathbf{X}, \mathbf{X})) \leq \lambda_{max}$ and $\lambda_{min}(\hat{\Theta}(\boldsymbol{w}_t; \mathbf{X}, \mathbf{X})) \geq \lambda_{min} > 0$ for $t \in [0, T]$. Then*

$$\Gamma \leq \sqrt{\frac{2\lambda_{max}\cdot\|f(\boldsymbol{w}_0, \mathbf{X}) - \boldsymbol{y}\|^2}{\lambda_{min}\cdot n}}(1 - e^{-\frac{2\lambda_{min}}{n}T}).$$

This bound shows that the generalization of overparameterized NNs depends on the condition number of the NTK. With a smaller condition number, the network converges faster and generalizes better. This bound is always upper-bounded even when $T \to \infty$. As $n/T$ increases, the bound decreases. Since $\|f(\boldsymbol{w}_0, \mathbf{X}) - \boldsymbol{y}\|^2 = O(n)$ for NTK initialization [25] and $1 - e^{-\frac{2\lambda_{\min}T}{n}} \leq \frac{2\lambda_{\min}T}{n}$, our bound has a faster rate than $O(\sqrt{\lambda_{\max}T/n})$. When $\frac{\lambda_{\max}}{\lambda_{\min}} = O(1)$, our bound has a rate of $O(\sqrt{1 - e^{-\frac{2\lambda_{\min}T}{n}}})$. For overparameterized NNs, NTK does not change much from initialization, hence $\lambda_{\max}$ and $\lambda_{\min}$ can be specified using the $\lambda_{\max}(\hat{\Theta}(\boldsymbol{w}_0; \mathbf{X}, \mathbf{X}))$ and $\lambda_{\min}(\hat{\Theta}(\boldsymbol{w}_0; \mathbf{X}, \mathbf{X}))$, see [25, 24, 53, 44, 65].

## 6.2 Kernel Ridge Regression

Given a kernel $K(\boldsymbol{x}, \boldsymbol{x}') = \langle \phi(\boldsymbol{x}), \phi(\boldsymbol{x}') \rangle$, where $\phi : \mathbb{R}^d \mapsto \mathbb{R}^p$, consider kernel ridge regression $f(\boldsymbol{w}, \boldsymbol{x}) = \boldsymbol{w}^\top \phi(\boldsymbol{x})$ with $L_{\mathcal{S}}(\boldsymbol{w}) = \frac{1}{2n} \|\phi(\mathbf{X})^\top \boldsymbol{w} - \boldsymbol{y}\|^2 + \frac{\lambda}{2} \|\boldsymbol{w}\|^2$ and $\ell(\boldsymbol{w}, \boldsymbol{z}) = \frac{1}{2}(\boldsymbol{w}^\top \phi(\boldsymbol{x}) - y)^2 + \frac{\lambda}{2} \|\boldsymbol{w}\|^2$, where $\phi(\mathbf{X}) \in \mathbb{R}^{p \times n}$ and $\boldsymbol{w} \in \mathbb{R}^p$. Denote the optimal solution as $\boldsymbol{w}^* = \frac{1}{n} \phi(\mathbf{X}) \left(\frac{1}{n} K(\mathbf{X}, \mathbf{X}) + \lambda \mathbf{I}_n\right)^{-1} \boldsymbol{y}$. Then we have the following bound for the kernel regression.

**Corollary 6.2.** *Suppose $K(\boldsymbol{x}_i, \boldsymbol{x}_i) \leq K_{\max}$ for all $i \in [n]$ and $K(\mathbf{X}, \mathbf{X})$ is full-rank. We have that*

$$
\Gamma \leq \begin{cases} \frac{1}{n} \sqrt{K_{\max}} \|\boldsymbol{w}_0 - \boldsymbol{w}^*\| \|\phi(\mathbf{X})^\top (\boldsymbol{w}_0 - \boldsymbol{w}^*)\|, & \text{when } \lambda = 0, \\ \frac{1}{n} \sqrt{K_{\max}} \sqrt{\boldsymbol{y}^\top (K(\mathbf{X}, \mathbf{X}))^{-1} \boldsymbol{y}} \|\boldsymbol{y}\|, & \text{when } \lambda = 0, \ \boldsymbol{w}_0 = 0. \end{cases} \tag{5}
$$

Here (5) recovers the Rademacher complexity bound for kernel regression [5]. Compared with the classic bound in Lemma 3.3, when $\|\boldsymbol{y}\| \leq \sqrt{n}$, (5) is tighter since $K_{\max} \leq \sum_i K(\boldsymbol{x}_i, \boldsymbol{x}_i)$. In the high-dimensional regime, if $\boldsymbol{w}^*$ is standard Gaussian and $\boldsymbol{w}_0 = 0$, (5) has a rate of $O(\sqrt{p/n})$. Similar rates can be found in [37, 38] for fixed kernel regression.

## 6.3 Feature Learning

Consider a single-index model $y = f_*(\langle \boldsymbol{\theta}^*, \boldsymbol{x} \rangle) + \xi$, where $\boldsymbol{\theta}^* \in \mathbb{S}^{d-1}$ is a fixed unit vector, data $\boldsymbol{x} \sim \mathcal{N}(0, \mathbf{I}_d)$, $f_*$ is an unknown link function, and $\xi \sim \mathcal{N}(0, \sigma^2)$ is an independent Gaussian noise. The sample complexity of this problem is usually $O(d^s)$ [7, 15] or $O(d^{s/2})$ [23], where $s$ is the information exponent of $f_*$, defined as the smallest nonzero coefficient of the Hermite expansion of $f_*$. Bietti et al. [15] trained a two-layer NN with gradient flow to learn this single-index model. Specifically, the NN is $f(\boldsymbol{\theta}, \boldsymbol{c}; \boldsymbol{x}) = \frac{1}{\sqrt{N}} \sum_{i=1}^N c_i \phi(\sigma_i \langle \boldsymbol{\theta}, \boldsymbol{x} \rangle + b_i)$, where $\boldsymbol{\theta} \in \mathbb{S}^{d-1}$, $\phi(u) = \max\{0, u\}$ is the ReLU activation function, $b_i \sim \mathcal{N}(0, \tau^2)$ with $\tau > 1$ are random biases that are frozen during training, and $\sigma_i$ are random Rademacher variables. Let $L_{\mathcal{S}}(\boldsymbol{\theta}, \boldsymbol{c}) = \frac{1}{n} \sum_{i=1}^n (f(\boldsymbol{\theta}, \boldsymbol{c}; \boldsymbol{x}_i) - y_i)^2 + \lambda \|\boldsymbol{c}\|^2$ be a regularized squared loss. The NN is trained by a two-stage gradient flow:

$$
\frac{d\boldsymbol{\theta}_t}{dt} = -\nabla_{\boldsymbol{\theta}}^{\mathbb{S}^{d-1}} L_{\mathcal{S}}(\boldsymbol{\theta}_t, \boldsymbol{c}_t), \qquad \frac{d\boldsymbol{c}_t}{dt} = -\mathbf{1}(t > T_0) \nabla_{\boldsymbol{c}} L_{\mathcal{S}}(\boldsymbol{\theta}_t, \boldsymbol{c}_t),
$$

where $\nabla_{\boldsymbol{\theta}}^{\mathbb{S}^{d-1}}$ is the Riemannian gradient on the unit sphere, $T_0 = \tilde{\Theta}(d^{\frac{s}{2}-1})$ and $s$ is the information exponent. They show that $n = \tilde{\Omega}(\frac{(d+N)d^{s-1}}{\lambda^4})$ is sufficient to guarantee weakly recovering the feature vector $\boldsymbol{\theta}^*$. Here we compute our bound in their setting.

**Corollary 6.3.** *Under the settings of Theorem 6.1 in Bietti et al. [15] (provided in Theorem G.2),*

$$
\Gamma \leq \tilde{O}\left(\sqrt{\frac{d^{\frac{s}{2}+1}}{n\lambda^2} + \lambda^2 d}\right),
$$

*with high probability as $n, d \to \infty$. As long as $\lambda = o_d(1/\sqrt{d})$ and $n = \tilde{\Omega}(d^{\frac{s}{2}+2})$, $\Gamma = o_{n,d}(1)$. Taking $\lambda = \Theta(\frac{d^{\frac{s}{2}}}{n})^{\frac{1}{4}}$, we have $\Gamma \leq \tilde{O}\left(\left(d^{\frac{s}{2}+2}/n\right)^{\frac{1}{4}}\right)$.*

Our bound is compatible with the requirements of $n = \tilde{\Omega}((d+N)d^{s-1}/\lambda^4)$ in Bietti et al. [15]. The sample complexity of $n = \tilde{\Omega}(d^{\frac{s}{2}+2})$ almost matches the correlational statistical query (CSQ) lower bound $n = \tilde{\Omega}(d^{\frac{s}{2}})$ [22, 1] and outperforms the kernel methods that require $n = \tilde{\Omega}(d^p)$ where $p$ is the degree of the polynomial of $f_*$ ($s \leq p$). Compared with Corollary 6.2, the bound of $\Gamma$ in the feature learning case is *vanishing*, while the bound in (5) is $\Theta(1)$, indicating the benefits of feature learning from the generalization gap bound.

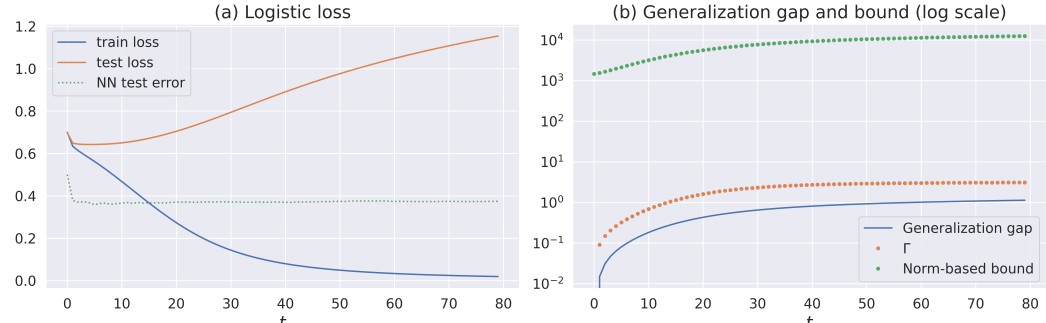

Figure 2: **Experiment (I)**. Two-layer NN trained by gradient descent on CIFAR-10 cat and dog. (a) shows NN's training loss, test loss, and test error. (b) shows that the complexity bound $\Gamma$ in Theorem 5.2 captures the generalization gap $L_\mu(\boldsymbol{w}_T) - L_\mathcal{S}(\boldsymbol{w}_T)$ well. It first increases and then converges after sufficient training time.

# 7 Numerical Experiments

We conduct comprehensive numerical experiments to demonstrate that our generalization bounds correlate well with the true generalization gap. For more simulations and details, see the Appendix.

**(I) Generalization bound of gradient flow in Theorem 5.2.** In Fig. 2, we use logistic loss to train a two-layer NN of 400 hidden nodes and Softplus activation function for binary classification on 4000 CIFAR-10 cat and dog [35] data by full-batch gradient descent and compute $\Gamma$, the main term in our bound. The integration in $\Gamma$ (3) is estimated with a Riemann sum. After training, the norm-based bound in Bartlett et al. [11] is 12557.3, which is much larger than our bound, as shown in the figure.

**(II) Generalization bound of SGF in Theorem 5.4.** In Fig. 1, we train a randomly initialized ResNet 18 by SGD on full CIFAR-10 [35] and estimate $\Gamma$ in our bound. Fig. 4, 5, and 6 in the Appendix show more experiments on ResNet 34 and two-layer NNs. Our generalization bound characterizes the *overfitting* and feature unlearning behavior [43] of overparameterized NNs after long-term training (when $T = O(n)$ in Theorem 5.2).

**(III) Generalization bound with label noise.** We corrupt the labels in the experiment (I) with random labels and plot the generalization gap and $\Gamma$ in Fig. 3. $\Gamma$ captures the generalization gap well and increases with the portion of label noise, explaining the random label phenomenon [67]. This behavior follows naturally: noisier labels force larger norm of loss gradients during training, which directly inflates $\Gamma$ and generalization gap.

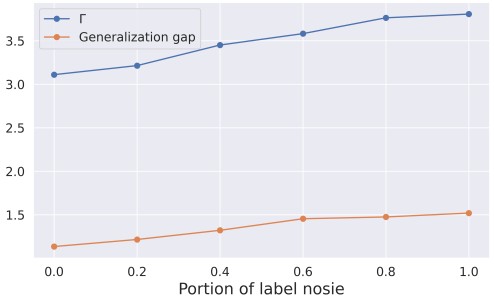

Figure 3: Generalization gap and our bound $\Gamma$ with label noise.

# 8 Conclusion and Future Work

In this paper, by combining the stability analysis and uniform convergence via Rademacher complexity, we derive a generalization bound for GF that parallels classical Rademacher complexity bounds for kernel methods by leveraging the data-dependent kernel LPK. Our results show that NNs trained by GF may outperform a fixed kernel by learning data-dependent kernels. Our bound also shows how the norm of the training loss gradients along the optimization trajectory affects the generalization. Recently, Montanari & Urbani [43] applied dynamical mean–field theory (DMFT) to two-layer NNs and showed that GF exhibits three distinct phases—an initial feature-learning regime ($T = O(1)$), a prolonged generalization plateau, and a late overfitting phase ($T = O(n)$) (Fig.2). Our bound (Theorem 5.2) reproduces similar qualitative behavior. Unlike the mean–field analysis, our approach applies to general architectures and data distributions. A promising direction is to integrate the DMFT's phase-wise insights with our LPK framework to obtain finer generalization guarantees.

For practice-relevant applications, by monitoring the evolution of our bound $\Gamma$ during training, one can predict the overfitting for overparameterized NNs and identify optimal stopping time for training without access test data [3]. Our $\Gamma$ can also serve as a proxy to compare model architectures in Neural Architecture Search (NAS) [55, 40, 19, 42, 20].

For future directions, extending the analysis to GD and SGD with large learning rates can bring the bound closer to practice. Second, our analysis uses a function class larger than $\mathcal{G}_T$ when bounding the Rademacher complexity. Refining this step could further tighten the bound.

## Acknowledgments

Yilan Chen and Arya Mazumdar were supported by NSF TRIPODS Institute grant 2217058 (En-CORE) and NSF 2217058 (TILOS). Zhichao Wang was supported by the NSF under Grant No. DMS-1928930, while he was in residence at the Simons Laufer Mathematical Sciences Institute in Berkeley, California, during the Spring 2025 semester. Wei Huang was supported by JSPS KAK-ENHI (24K20848) and JST BOOST (JPMJBY24G6). Taiji Suzuki was partially supported by JSPS KAKENHI (24K02905) and JST CREST (JPMJCR2015).

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

# Appendices

## Limitations and Impact Statement

Our analysis focuses on the generalization bound of the gradient flow algorithm. The behaviors of other algorithms, such as gradient descent (GD), stochastic gradient descent (SGD), and Adam, are still unclear. Extending the analysis to GD and SGD with large learning rates can bring the bound closer to practice.

This paper presents work whose goal is to advance the field of Machine Learning. There are many potential societal consequences of our work, none of which we feel must be specifically highlighted here.

## A    Additional Experiments

In experiment (I), we train the two-layer NN with a learning rate of $\eta = 0.01$ for $8000$ steps. The training time is calculated by $T = \eta \times$steps. The integration in $\Gamma$ (3) is estimated by computing the gradient norm at each training step and summing over the steps. For experiment (II) and Fig. 4, we train Resnet 18 and Resnet 34 with a learning rate of 0.001 and batch size of 128 for 50 epochs. For Fig. 5, we train a two-layer NN of 1000 hidden nodes with a learning rate of 0.01 and batch size 128 for 100 epochs. For Fig. 6, we train a two-layer NN of 1000 hidden nodes with a learning rate of 0.1 and batch size 200 for 10 epochs. Experiments are implemented with PyTorch [56] on 24G A5000 and V100 GPUs.

Fig. 4 and Fig. 5 have similar behavior with Fig. 1. The models first learn the features, then overfit. Fig. 6 has less overfitting. Our bound correlates well with the true generalization gap in both cases and for all models.

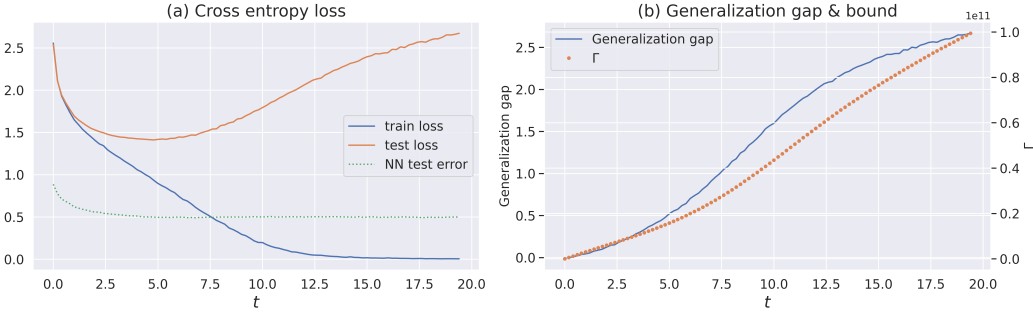

Figure 4: **Experiment (II)**. ResNet 34 trained by SGD on full CIFAR-10.

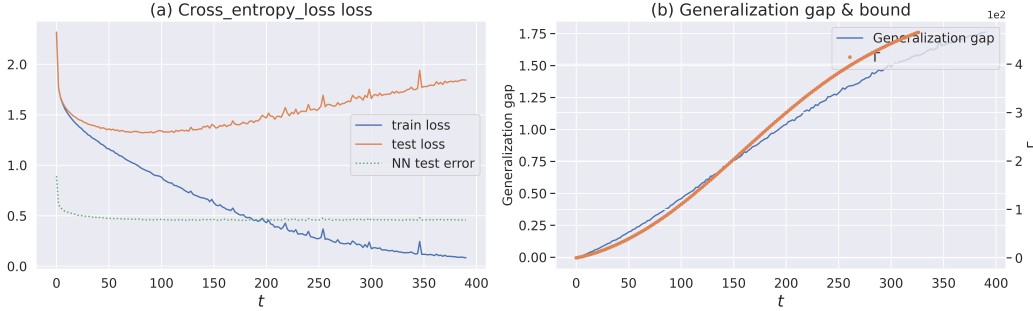

Figure 5: **Experiment (II)**. Two-layer NN trained by SGD on full CIFAR-10.

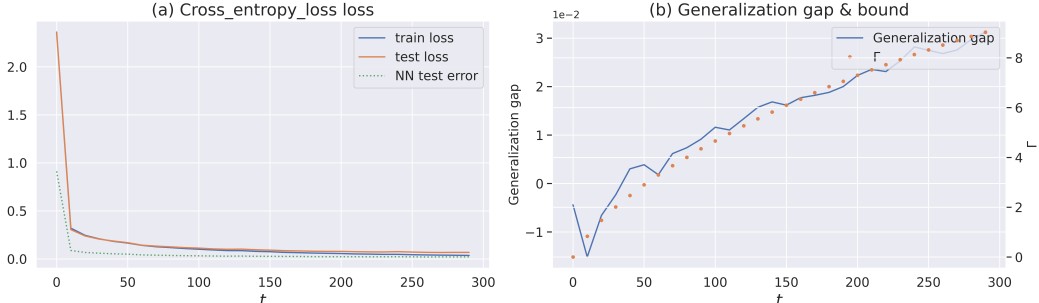

Figure 6: **Experiment (II)**. Two-layer NN trained by SGD on full MNIST.

# B  Uniform Stability of Gradient Flow

**Lemma 4.3.** *Under Assumption 4.2, for any two data sets $\mathcal{S}$ and $\mathcal{S}^{(i)}$, let $\boldsymbol{w}_t = \mathcal{A}_t(\mathcal{S})$ and $\boldsymbol{w}'_t = \mathcal{A}_t(\mathcal{S}^{(i)}))$ be the parameters trained from same initialization $\boldsymbol{w}_0 = \boldsymbol{w}'_0$, then*

$$\|\boldsymbol{w}_t - \boldsymbol{w}'_t\| \leq \begin{cases} \frac{2L}{\gamma n}, & L_S(\boldsymbol{w}) \text{ is } \gamma\text{-S.C.,} \\ \frac{2Lt}{n}, & L_S(\boldsymbol{w}) \text{ is convex,} \\ \frac{2L}{\beta n}(e^{\beta t} - 1), & L_S(\boldsymbol{w}) \text{ is non-convex.} \end{cases}$$

*Proof.* **Convex Case.** Notice that

$$\frac{d\|\boldsymbol{w}_t - \boldsymbol{w}'_t\|^2}{dt}$$
$$= \left\langle \frac{\partial\|\boldsymbol{w}_t - \boldsymbol{w}'_t\|^2}{\partial(\boldsymbol{w}_t - \boldsymbol{w}'_t)}, \frac{d(\boldsymbol{w}_t - \boldsymbol{w}'_t)}{dt} \right\rangle$$
$$= 2(\boldsymbol{w}_t - \boldsymbol{w}'_t)^\top \frac{d(\boldsymbol{w}_t - \boldsymbol{w}'_t)}{dt}$$
$$= 2(\boldsymbol{w}_t - \boldsymbol{w}'_t)^\top (-\nabla_{\boldsymbol{w}} L_{\mathcal{S}}(\boldsymbol{w}_t) + \nabla_{\boldsymbol{w}} L_{\mathcal{S}^{(i)}}(\boldsymbol{w}'_t))$$
$$= 2(\boldsymbol{w}_t - \boldsymbol{w}'_t)^\top (-\nabla_{\boldsymbol{w}} L_{\mathcal{S}}(\boldsymbol{w}_t) + \nabla_{\boldsymbol{w}} L_{\mathcal{S}^{(i)}}(\boldsymbol{w}_t) - \nabla_{\boldsymbol{w}} L_{\mathcal{S}^{(i)}}(\boldsymbol{w}_t) + \nabla_{\boldsymbol{w}} L_{\mathcal{S}^{(i)}}(\boldsymbol{w}'_t))$$
$$= \frac{2}{n}(\boldsymbol{w}_t - \boldsymbol{w}'_t)^\top (\nabla_{\boldsymbol{w}} \ell(\boldsymbol{w}_t, \boldsymbol{z}'_i) - \nabla_{\boldsymbol{w}} \ell(\boldsymbol{w}_t, \boldsymbol{z}_i)) - 2(\boldsymbol{w}_t - \boldsymbol{w}'_t)^\top (\nabla_{\boldsymbol{w}} L_{\mathcal{S}^{(i)}}(\boldsymbol{w}_t) - \nabla_{\boldsymbol{w}} L_{\mathcal{S}^{(i)}}(\boldsymbol{w}'_t))$$
$$\leq \frac{2}{n}(\boldsymbol{w}_t - \boldsymbol{w}'_t)^\top (\nabla_{\boldsymbol{w}} \ell(\boldsymbol{w}_t, \boldsymbol{z}'_i) - \nabla_{\boldsymbol{w}} \ell(\boldsymbol{w}_t, \boldsymbol{z}_i)) \qquad\qquad \text{(convexity)}$$
$$\leq \frac{4L}{n}\|\boldsymbol{w}_t - \boldsymbol{w}'_t\|.$$

Since also $\frac{d\|\boldsymbol{w}_t - \boldsymbol{w}'_t\|^2}{dt} = 2\|\boldsymbol{w}_t - \boldsymbol{w}'_t\|\frac{d\|\boldsymbol{w}_t - \boldsymbol{w}'_t\|}{dt}$, we have

$$2\|\boldsymbol{w}_t - \boldsymbol{w}'_t\|\frac{d\|\boldsymbol{w}_t - \boldsymbol{w}'_t\|}{dt} \leq \frac{4L}{n}\|\boldsymbol{w}_t - \boldsymbol{w}'_t\|.$$

When $\|\boldsymbol{w}_t - \boldsymbol{w}'_t\| = 0$, the result already hold. When $\|\boldsymbol{w}_t - \boldsymbol{w}'_t\| > 0$,

$$\frac{d\|\boldsymbol{w}_t - \boldsymbol{w}'_t\|}{dt} \leq \frac{2L}{n}.$$

Solve the differential equation, we have

$$\|\boldsymbol{w}_t - \boldsymbol{w}'_t\| \leq \frac{2Lt}{n}.$$

Thus, we complete the proof of the convex case.

$\gamma$**-Strongly Convex Case.** Notice that

$$\frac{d\left\|\boldsymbol{w}_t - \boldsymbol{w}_t'\right\|^2}{dt}$$

$$= 2\left(\boldsymbol{w}_t - \boldsymbol{w}_t'\right)^\top \frac{d\left(\boldsymbol{w}_t - \boldsymbol{w}_t'\right)}{dt}$$

$$= 2\left(\boldsymbol{w}_t - \boldsymbol{w}_t'\right)^\top \left(-\nabla_{\boldsymbol{w}} L_{\mathcal{S}}(\boldsymbol{w}_t) + \nabla_{\boldsymbol{w}} L_{\mathcal{S}^{(i)}}(\boldsymbol{w}_t')\right)$$

$$= 2\left(\boldsymbol{w}_t - \boldsymbol{w}_t'\right)^\top \left(-\nabla_{\boldsymbol{w}} L_{\mathcal{S}}(\boldsymbol{w}_t) + \nabla_{\boldsymbol{w}} L_{\mathcal{S}^{(i)}}(\boldsymbol{w}_t) - \nabla_{\boldsymbol{w}} L_{\mathcal{S}^{(i)}}(\boldsymbol{w}_t) + \nabla_{\boldsymbol{w}} L_{\mathcal{S}^{(i)}}(\boldsymbol{w}_t')\right)$$

$$= \frac{2}{n}\left(\boldsymbol{w}_t - \boldsymbol{w}_t'\right)^\top \left(\nabla_{\boldsymbol{w}}\ell(\boldsymbol{w}_t, \boldsymbol{z}_i') - \nabla_{\boldsymbol{w}}\ell(\boldsymbol{w}_t, \boldsymbol{z}_i)\right) - 2\left(\boldsymbol{w}_t - \boldsymbol{w}_t'\right)^\top \left(\nabla_{\boldsymbol{w}} L_{\mathcal{S}^{(i)}}(\boldsymbol{w}_t) - \nabla_{\boldsymbol{w}} L_{\mathcal{S}^{(i)}}(\boldsymbol{w}_t')\right)$$

$$\leq \frac{4L}{n}\left\|\boldsymbol{w}_t - \boldsymbol{w}_t'\right\| - 2\gamma \left\|\boldsymbol{w}_t - \boldsymbol{w}_t'\right\|^2$$

$$= 2\left\|\boldsymbol{w}_t - \boldsymbol{w}_t'\right\|\left(\frac{2L}{n} - \gamma \left\|\boldsymbol{w}_t - \boldsymbol{w}_t'\right\|\right). \tag{6}$$

Now we prove $\|\boldsymbol{w}_t - \boldsymbol{w}_t'\| \leq \frac{2L}{\gamma n}$ by contradition. Recall $\|\boldsymbol{w}_0 - \boldsymbol{w}_0'\| = 0$. Suppose that there is some time $T$ such that $\|\boldsymbol{w}_T - \boldsymbol{w}_T'\| > \frac{2L}{\gamma n}$, then there must be some $T' < T$ such that $\|\boldsymbol{w}_{T'} - \boldsymbol{w}_{T'}'\| = \frac{2L}{\gamma n}$ and $\|\boldsymbol{w}_t - \boldsymbol{w}_t'\|$ is increasing at some point between $[T', T]$. However, when $\|\boldsymbol{w}_t - \boldsymbol{w}_t'\| > \frac{2L}{\gamma n}$, by (6), $\frac{d\|\boldsymbol{w}_t - \boldsymbol{w}_t'\|^2}{dt} < 0$ and $\|\boldsymbol{w}_t - \boldsymbol{w}_t'\|$ must decrease. Therefore contradict and we must have $\|\boldsymbol{w}_t - \boldsymbol{w}_t'\| \leq \frac{2L}{\gamma n}$.

**Non-Convex Case.** First of all, we have that

$$\frac{d\left\|\boldsymbol{w}_t - \boldsymbol{w}_t'\right\|^2}{dt}$$

$$= 2\left(\boldsymbol{w}_t - \boldsymbol{w}_t'\right)^\top \frac{d\left(\boldsymbol{w}_t - \boldsymbol{w}_t'\right)}{dt}$$

$$= 2\left(\boldsymbol{w}_t - \boldsymbol{w}_t'\right)^\top \left(-\nabla_{\boldsymbol{w}} L_{\mathcal{S}}(\boldsymbol{w}_t) + \nabla_{\boldsymbol{w}} L_{\mathcal{S}^{(i)}}(\boldsymbol{w}_t')\right)$$

$$= 2\left(\boldsymbol{w}_t - \boldsymbol{w}_t'\right)^\top \left(-\nabla_{\boldsymbol{w}} L_{\mathcal{S}}(\boldsymbol{w}_t) + \nabla_{\boldsymbol{w}} L_{\mathcal{S}^{(i)}}(\boldsymbol{w}_t) - \nabla_{\boldsymbol{w}} L_{\mathcal{S}^{(i)}}(\boldsymbol{w}_t) + \nabla_{\boldsymbol{w}} L_{\mathcal{S}^{(i)}}(\boldsymbol{w}_t')\right)$$

$$= \frac{2}{n}\left(\boldsymbol{w}_t - \boldsymbol{w}_t'\right)^\top \left(\nabla_{\boldsymbol{w}}\ell(\boldsymbol{w}_t, \boldsymbol{z}_i') - \nabla_{\boldsymbol{w}}\ell(\boldsymbol{w}_t, \boldsymbol{z}_i)\right) - 2\left(\boldsymbol{w}_t - \boldsymbol{w}_t'\right)^\top \left(\nabla_{\boldsymbol{w}} L_{\mathcal{S}^{(i)}}(\boldsymbol{w}_t) - \nabla_{\boldsymbol{w}} L_{\mathcal{S}^{(i)}}(\boldsymbol{w}_t')\right)$$

$$\leq \frac{2}{n}\left\|\boldsymbol{w}_t - \boldsymbol{w}_t'\right\| \left\|\nabla_{\boldsymbol{w}}\ell(\boldsymbol{w}_t, \boldsymbol{z}_i') - \nabla_{\boldsymbol{w}}\ell(\boldsymbol{w}_t, \boldsymbol{z}_i)\right\| + 2\left\|\boldsymbol{w}_t - \boldsymbol{w}_t'\right\| \left\|\nabla_{\boldsymbol{w}} L_{\mathcal{S}^{(i)}}(\boldsymbol{w}_t) - \nabla_{\boldsymbol{w}} L_{\mathcal{S}^{(i)}}(\boldsymbol{w}_t')\right\|$$

$$\leq \frac{4L}{n}\left\|\boldsymbol{w}_t - \boldsymbol{w}_t'\right\| + 2\beta \left\|\boldsymbol{w}_t - \boldsymbol{w}_t'\right\|^2.$$

Since also $\frac{d\|\boldsymbol{w}_t - \boldsymbol{w}_t'\|^2}{dt} = 2\|\boldsymbol{w}_t - \boldsymbol{w}_t'\|\frac{d\|\boldsymbol{w}_t - \boldsymbol{w}_t'\|}{dt}$, we have

$$2\left\|\boldsymbol{w}_t - \boldsymbol{w}_t'\right\|\frac{d\left\|\boldsymbol{w}_t - \boldsymbol{w}_t'\right\|}{dt} \leq \frac{4L}{n}\left\|\boldsymbol{w}_t - \boldsymbol{w}_t'\right\| + 2\beta \left\|\boldsymbol{w}_t - \boldsymbol{w}_t'\right\|^2.$$

When $\|\boldsymbol{w}_t - \boldsymbol{w}_t'\| = 0$, the result already hold. When $\|\boldsymbol{w}_t - \boldsymbol{w}_t'\| > 0$,

$$\frac{d\left\|\boldsymbol{w}_t - \boldsymbol{w}_t'\right\|}{dt} \leq \frac{2L}{n} + \beta \left\|\boldsymbol{w}_t - \boldsymbol{w}_t'\right\|.$$

From this we have

$$\frac{d\left\|\boldsymbol{w}_t - \boldsymbol{w}_t'\right\|}{\frac{2L}{n\beta} + \left\|\boldsymbol{w}_t - \boldsymbol{w}_t'\right\|} \leq \beta dt.$$

Solve the differential equation, we have

$$\left\|\boldsymbol{w}_t - \boldsymbol{w}_t'\right\| \leq \frac{2L}{\beta n}(e^{\beta t} - 1).$$

$\square$

## C Concentration of Loss Path Kernels under Stability

In the following, we will only show the proofs for the convex case. The proofs for strongly convex and non-convex cases are similar.

**Lemma C.1.** *Let $\mathcal{S}$ and $\mathcal{S}^{(i)}$ be two datasets that only differ in $i$-th data point. Under Assumption 4.2, for any $\boldsymbol{z}, \boldsymbol{z}'$,*

$$\left|\mathsf{K}_T(\boldsymbol{z}, \boldsymbol{z}'; \mathcal{S}) - \mathsf{K}_T(\boldsymbol{z}, \boldsymbol{z}'; \mathcal{S}^{(i)})\right| \leq \begin{cases} \frac{4L^2\beta T}{\gamma n}, & \text{when } L_S(\boldsymbol{w}) \text{ is } \gamma\text{-strongly convex,} \\ \frac{2L^2\beta T^2}{n}, & \text{when } L_S(\boldsymbol{w}) \text{ is convex,} \\ \frac{4L^2}{\beta n}(e^{\beta T} - \beta T - 1), & \text{when } L_S(\boldsymbol{w}) \text{ is non-convex.} \end{cases}$$

*Proof.* For convex loss, by the smoothness and Lemma 4.3, we have $\left\|\nabla_{\boldsymbol{w}}\ell(\mathcal{A}_t(\mathcal{S}), \boldsymbol{z}) - \nabla_{\boldsymbol{w}}\ell(\mathcal{A}_t(\mathcal{S}^{(i)}), \boldsymbol{z})\right\| \leq \beta \left\|\mathcal{A}_t(\mathcal{S}) - \mathcal{A}_t(\mathcal{S}^{(i)})\right\| \leq \beta\frac{2Lt}{n}$ for all $\boldsymbol{z}$. Then

$$\mathsf{K}_T(\boldsymbol{z}, \boldsymbol{z}'; \mathcal{S}) - \mathsf{K}_T(\boldsymbol{z}, \boldsymbol{z}'; \mathcal{S}^{(i)})$$

$$= \int_0^T \langle \nabla_{\boldsymbol{w}}\ell(\mathcal{A}_t(\mathcal{S}), \boldsymbol{z}), \nabla_{\boldsymbol{w}}\ell(\mathcal{A}_t(\mathcal{S}), \boldsymbol{z}')\rangle - \left\langle \nabla_{\boldsymbol{w}}\ell(\mathcal{A}_t(\mathcal{S}^{(i)}), \boldsymbol{z}), \nabla_{\boldsymbol{w}}\ell(\mathcal{A}_t(\mathcal{S}^{(i)}), \boldsymbol{z}')\right\rangle dt$$

$$= \int_0^T \langle \nabla_{\boldsymbol{w}}\ell(\mathcal{A}_t(\mathcal{S}), \boldsymbol{z}), \nabla_{\boldsymbol{w}}\ell(\mathcal{A}_t(\mathcal{S}), \boldsymbol{z}')\rangle - \left\langle \nabla_{\boldsymbol{w}}\ell(\mathcal{A}_t(\mathcal{S}), \boldsymbol{z}), \nabla_{\boldsymbol{w}}\ell(\mathcal{A}_t(\mathcal{S}^{(i)}), \boldsymbol{z}')\right\rangle$$

$$+ \left\langle \nabla_{\boldsymbol{w}}\ell(\mathcal{A}_t(\mathcal{S}), \boldsymbol{z}), \nabla_{\boldsymbol{w}}\ell(\mathcal{A}_t(\mathcal{S}^{(i)}), \boldsymbol{z}')\right\rangle - \left\langle \nabla_{\boldsymbol{w}}\ell(\mathcal{A}_t(\mathcal{S}^{(i)}), \boldsymbol{z}), \nabla_{\boldsymbol{w}}\ell(\mathcal{A}_t(\mathcal{S}^{(i)}), \boldsymbol{z}')\right\rangle dt$$

$$= \int_0^T \left\langle \nabla_{\boldsymbol{w}}\ell(\mathcal{A}_t(\mathcal{S}), \boldsymbol{z}), \nabla_{\boldsymbol{w}}\ell(\mathcal{A}_t(\mathcal{S}), \boldsymbol{z}') - \nabla_{\boldsymbol{w}}\ell(\mathcal{A}_t(\mathcal{S}^{(i)}), \boldsymbol{z}')\right\rangle$$

$$+ \left\langle \nabla_{\boldsymbol{w}}\ell(\mathcal{A}_t(\mathcal{S}), \boldsymbol{z}) - \nabla_{\boldsymbol{w}}\ell(\mathcal{A}_t(\mathcal{S}^{(i)}), \boldsymbol{z}), \nabla_{\boldsymbol{w}}\ell(\mathcal{A}_t(\mathcal{S}^{(i)}), \boldsymbol{z}')\right\rangle dt$$

$$\leq \int_0^T \|\nabla_{\boldsymbol{w}}\ell(\mathcal{A}_t(\mathcal{S}), \boldsymbol{z})\| \left\|\nabla_{\boldsymbol{w}}\ell(\mathcal{A}_t(\mathcal{S}), \boldsymbol{z}') - \nabla_{\boldsymbol{w}}\ell(\mathcal{A}_t(\mathcal{S}^{(i)}), \boldsymbol{z}')\right\|$$

$$+ \left\|\nabla_{\boldsymbol{w}}\ell(\mathcal{A}_t(\mathcal{S}), \boldsymbol{z}) - \nabla_{\boldsymbol{w}}\ell(\mathcal{A}_t(\mathcal{S}^{(i)}), \boldsymbol{z})\right\| \left\|\nabla_{\boldsymbol{w}}\ell(\mathcal{A}_t(\mathcal{S}^{(i)}), \boldsymbol{z}')\right\| dt$$

$$\leq \int_0^T L\beta\frac{2Lt}{n} + \beta\frac{2Lt}{n}L\, dt$$

$$= \frac{2L^2\beta T^2}{n}$$

Similarly $\mathsf{K}_T(\boldsymbol{z}, \boldsymbol{z}'; \mathcal{S}^{(i)}) - \mathsf{K}_T(\boldsymbol{z}, \boldsymbol{z}'; \mathcal{S}) \leq \frac{2L^2\beta T^2}{n}$. Thus $\left|\mathsf{K}_T(\boldsymbol{z}, \boldsymbol{z}'; \mathcal{S}) - \mathsf{K}_T(\boldsymbol{z}, \boldsymbol{z}'; \mathcal{S}^{(i)})\right| \leq \frac{2L^2\beta T^2}{n}$. $\qquad\square$

With this, we can show that $\mathsf{K}_T(\boldsymbol{z}, \boldsymbol{z}'; \mathcal{S}')$ concentrate to its expectation.

**Lemma 4.4.** *Under Assumption 4.2, for any fixed $\boldsymbol{z}, \boldsymbol{z}'$, with probability at least $1 - \delta$ over the randomness of $\mathcal{S}'$,*

$$\left|\mathsf{K}_T(\boldsymbol{z}, \boldsymbol{z}'; \mathcal{S}') - \mathop{\mathbb{E}}_{\mathcal{S}'} \mathsf{K}_T(\boldsymbol{z}, \boldsymbol{z}'; \mathcal{S}')\right| \leq \begin{cases} \frac{4L^2\beta T}{\gamma}\sqrt{\frac{\ln\frac{2}{\delta}}{2n}}, & L_S(\boldsymbol{w}) \text{ is } \gamma\text{-S.C.,} \\ 2L^2\beta T^2\sqrt{\frac{\ln\frac{2}{\delta}}{2n}}, & L_S(\boldsymbol{w}) \text{ is convex,} \\ \frac{4L^2}{\beta}(e^{\beta T} - \beta T - 1)\sqrt{\frac{\ln\frac{2}{\delta}}{2n}}, & L_S(\boldsymbol{w}) \text{ is non-convex.} \end{cases}$$

*Proof.* We prove for the convex case. Strongly convex and non-convex cases are similar. Let $\mathcal{S}'$ and $\mathcal{S}'^{(i)}$ be two datasets that differ only in the $i$-th data point. By Lemma C.1, $\left|\mathsf{K}_T(\boldsymbol{z}, \boldsymbol{z}', \mathcal{S}') - \mathsf{K}_T(\boldsymbol{z}, \boldsymbol{z}', \mathcal{S}'^{(i)})\right| \leq \frac{2L^2\beta T^2}{n}$. Then by McDiarmid's inequality, for any $\delta \in (0, 1)$, with probability at least $1 - \delta$,

$$\left|\mathsf{K}_T(\boldsymbol{z}, \boldsymbol{z}'; \mathcal{S}') - \mathop{\mathbb{E}}_{\mathcal{S}'} \mathsf{K}_T(\boldsymbol{z}, \boldsymbol{z}'; \mathcal{S}')\right| \leq 2L^2\beta T^2\sqrt{\frac{\ln\frac{2}{\delta}}{2n}}.$$

$\square$

Similarly, we show that LPK on the training set concentrates to its expectation.

**Lemma C.2.** *Under Assumption 4.2, with probability at least $1 - \delta$ over the randomness of $\mathcal{S}$,*

$$\left| \sum_{i=1}^{n} \mathsf{K}_T(\boldsymbol{z}_i, \boldsymbol{z}_i; \mathcal{S}) - \mathbb{E}_{\mathcal{S}}\left[ \sum_{i=1}^{n} \mathsf{K}_T(\boldsymbol{z}_i, \boldsymbol{z}_i; \mathcal{S}) \right] \right|$$
$$\leq \begin{cases} \left( L^2 T + \frac{2L^2 \beta T}{\gamma} \right) \sqrt{2n \log \frac{2}{\delta}}, & L_S(\boldsymbol{w}) \text{ is } \gamma\text{-strongly convex,} \\ \left( L^2 T + L^2 \beta T^2 \right) \sqrt{2n \log \frac{2}{\delta}}, & L_S(\boldsymbol{w}) \text{ is convex,} \\ \left( L^2 T + \frac{2L^2}{\beta} (e^{\beta T} - \beta T - 1) \right) \sqrt{2n \log \frac{2}{\delta}}, & L_S(\boldsymbol{w}) \text{ is non-convex.} \end{cases}$$

*Proof.* For any fixed $j \in [n]$, let $\mathcal{S}$ and $\mathcal{S}^{(j)}$ be two datasets that only differ in $j$-th data point.

$$\left| \sum_{i=1}^{n} \mathsf{K}_T(\boldsymbol{z}_i, \boldsymbol{z}_i, \mathcal{S}) - \sum_{i=1}^{n} \mathsf{K}_T(\boldsymbol{z}_i, \boldsymbol{z}_i, \mathcal{S}^{(j)}) \right|$$
$$= \left| \sum_{i \neq j} \left( \mathsf{K}_T(\boldsymbol{z}_i, \boldsymbol{z}_i, \mathcal{S}) - \mathsf{K}_T(\boldsymbol{z}_i, \boldsymbol{z}_i, \mathcal{S}^{(j)}) \right) + \mathsf{K}_T(\boldsymbol{z}_j, \boldsymbol{z}_j, \mathcal{S}) - \mathsf{K}_T(\boldsymbol{z}'_j, \boldsymbol{z}'_j, \mathcal{S}^{(j)}) \right|$$
$$\leq \sum_{i \neq j} \left| \mathsf{K}_T(\boldsymbol{z}_i, \boldsymbol{z}_i, \mathcal{S}) - \mathsf{K}_T(\boldsymbol{z}_i, \boldsymbol{z}_i, \mathcal{S}^{(j)}) \right| + \left| \mathsf{K}_T(\boldsymbol{z}_j, \boldsymbol{z}_j, \mathcal{S}) - \mathsf{K}_T(\boldsymbol{z}'_j, \boldsymbol{z}'_j, \mathcal{S}^{(j)}) \right|$$

When $j \neq i$, by Lemma C.1, for convex loss,

$$\left| \mathsf{K}_T(\boldsymbol{z}_i, \boldsymbol{z}_i, \mathcal{S}) - \mathsf{K}_T(\boldsymbol{z}_i, \boldsymbol{z}_i, \mathcal{S}^{(j)}) \right| \leq \frac{2L^2 \beta T^2}{n}.$$

When $j = i$, by the definition of LPK, it can be bound by the Lipschitz constant,

$$\left| \mathsf{K}_T(\boldsymbol{z}_j, \boldsymbol{z}_j, \mathcal{S}) - \mathsf{K}_T(\boldsymbol{z}'_j, \boldsymbol{z}'_j, \mathcal{S}^{(j)}) \right| \leq 2L^2 T.$$

Therefore

$$\left| \sum_{i=1}^{n} \mathsf{K}_T(\boldsymbol{z}_i, \boldsymbol{z}_i, \mathcal{S}) - \sum_{i=1}^{n} \mathsf{K}_T(\boldsymbol{z}_i, \boldsymbol{z}_i, \mathcal{S}^{(j)}) \right| \leq (n-1) \frac{2L^2 \beta T^2}{n} + 2L^2 T$$
$$\leq 2L^2 \beta T^2 + 2L^2 T.$$

Then by McDiarmid's inequality, for any $\delta \in (0, 1)$, with probability at least $1 - \delta$ over the randomness of $\mathcal{S}$,

$$\left| \sum_{i=1}^{n} \mathsf{K}_T(\boldsymbol{z}_i, \boldsymbol{z}_i; \mathcal{S}) - \mathbb{E}_{\mathcal{S}}\left[ \sum_{i=1}^{n} \mathsf{K}_T(\boldsymbol{z}_i, \boldsymbol{z}_i; \mathcal{S}) \right] \right| \leq \left( L^2 T + L^2 \beta T^2 \right) \sqrt{2n \log \frac{2}{\delta}}.$$

$\square$

## C.1 Bound the Trace Term

With the above results, we are able to bound the difference between $\sum_{i=1}^{n} \mathsf{K}_T(\boldsymbol{z}_i, \boldsymbol{z}_i; \mathcal{S}')$ and $\sum_{i=1}^{n} \mathsf{K}_T(\boldsymbol{z}_i, \boldsymbol{z}_i; \mathcal{S})$.

**Lemma 4.5.** *Under Assumption 4.2, for two datasets $\mathcal{S}$ and $\mathcal{S}'$, with probability at least $1 - \delta$ over the randomness of $\mathcal{S}$ and $\mathcal{S}'$,*

$$\left| \sum_{i=1}^{n} \mathsf{K}_T(\boldsymbol{z}_i, \boldsymbol{z}_i; \mathcal{S}) - \sum_{i=1}^{n} \mathsf{K}_T(\boldsymbol{z}_i, \boldsymbol{z}_i; \mathcal{S}') \right| \leq \begin{cases} \tilde{O}(T \sqrt{n}), & \text{when } L_S(\boldsymbol{w}) \text{ is } \gamma\text{-strongly convex,} \\ \tilde{O}(T^2 \sqrt{n}), & \text{when } L_S(\boldsymbol{w}) \text{ is convex,} \\ \tilde{O}(e^T \sqrt{n}), & \text{when } L_S(\boldsymbol{w}) \text{ is non-convex.} \end{cases}$$

*Proof.* For any $\lambda > 0$,

$$\underset{\mathcal{S}}{\mathbb{E}}\, e^{\lambda \sum_i \mathsf{K}_T(\boldsymbol{z}_i, \boldsymbol{z}_i; \mathcal{S})} = \underset{\mathcal{S}}{\mathbb{E}}\, \underset{\mathcal{S}'}{\mathbb{E}}\, e^{\lambda \sum_i \mathsf{K}_T(\boldsymbol{z}_i', \boldsymbol{z}_i'; \mathcal{S}^{(i)})} \qquad \text{(replace } \boldsymbol{z}_i \text{ with } \boldsymbol{z}_i')$$

$$= \underset{\mathcal{S}}{\mathbb{E}}\, \underset{\mathcal{S}'}{\mathbb{E}}\, e^{\lambda \sum_i \left( \mathsf{K}_T(\boldsymbol{z}_i', \boldsymbol{z}_i'; \mathcal{S}) + \mathsf{K}_T(\boldsymbol{z}_i', \boldsymbol{z}_i'; \mathcal{S}^{(i)}) - \mathsf{K}_T(\boldsymbol{z}_i', \boldsymbol{z}_i'; \mathcal{S}) \right)}$$

If $\boldsymbol{z}_i' = \boldsymbol{z}_i$, $\mathsf{K}_T(\boldsymbol{z}_i', \boldsymbol{z}_i'; \mathcal{S}^{(i)}) - \mathsf{K}_T(\boldsymbol{z}_i', \boldsymbol{z}_i'; \mathcal{S}) = 0$. If $\boldsymbol{z}_i' \neq \boldsymbol{z}_i$, by Lemma C.1, for convex loss, $\left| \mathsf{K}_T(\boldsymbol{z}_i', \boldsymbol{z}_i'; \mathcal{S}^{(i)}) - \mathsf{K}_T(\boldsymbol{z}_i', \boldsymbol{z}_i'; \mathcal{S}) \right| \leq \frac{2L^2 \beta T^2}{n}$. Therefore,

$$\underset{\mathcal{S}}{\mathbb{E}}\, e^{\lambda \sum_i \mathsf{K}_T(\boldsymbol{z}_i, \boldsymbol{z}_i; \mathcal{S})} \geq \underset{\mathcal{S}}{\mathbb{E}}\, \underset{\mathcal{S}'}{\mathbb{E}}\, e^{\lambda \sum_i \mathsf{K}_T(\boldsymbol{z}_i', \boldsymbol{z}_i'; \mathcal{S}) - 2\lambda L^2 \beta T^2}$$

$$= \underset{\mathcal{S}}{\mathbb{E}}\, \underset{\mathcal{S}'}{\mathbb{E}}\, e^{\lambda \sum_i \mathsf{K}_T(\boldsymbol{z}_i, \boldsymbol{z}_i; \mathcal{S}') - 2\lambda L^2 \beta T^2}. \qquad \text{(exchange the name of } \mathcal{S} \text{ and } \mathcal{S}')$$

Hence, we have

$$\underset{\mathcal{S}}{\mathbb{E}}\, \underset{\mathcal{S}'}{\mathbb{E}}\, e^{\lambda \sum_i \mathsf{K}_T(\boldsymbol{z}_i, \boldsymbol{z}_i; \mathcal{S}')} \leq e^{2\lambda L^2 \beta T^2} \underset{\mathcal{S}}{\mathbb{E}}\, e^{\lambda \sum_i \mathsf{K}_T(\boldsymbol{z}_i, \boldsymbol{z}_i; \mathcal{S})}.$$

By Markov's inequality,

$$\mathbb{P}\left( \sum_{i=1}^{n} \mathsf{K}_T(\boldsymbol{z}_i, \boldsymbol{z}_i; \mathcal{S}') \geq t \right) = \mathbb{P}\left( e^{\lambda \sum_i \mathsf{K}_T(\boldsymbol{z}_i, \boldsymbol{z}_i; \mathcal{S}')} \geq e^{\lambda t} \right)$$

$$\leq \frac{\mathbb{E}_{\mathcal{S}} \, \mathbb{E}_{\mathcal{S}'} \, e^{\lambda \sum_i \mathsf{K}_T(\boldsymbol{z}_i, \boldsymbol{z}_i; \mathcal{S}')}}{e^{\lambda t}}$$

$$\leq \frac{e^{2\lambda L^2 \beta T^2} \, \mathbb{E}_{\mathcal{S}} \, e^{\lambda \sum_i \mathsf{K}_T(\boldsymbol{z}_i, \boldsymbol{z}_i; \mathcal{S})}}{e^{\lambda t}}.$$

Set the RHS as $\delta$, we have at least $1 - \delta$ over the randomness of $\mathcal{S}$ and $\mathcal{S}'$,

$$\sum_{i=1}^{n} \mathsf{K}_T(\boldsymbol{z}_i, \boldsymbol{z}_i; \mathcal{S}') \leq \frac{1}{\lambda} \left( \ln \underset{\mathcal{S}}{\mathbb{E}}\, e^{\lambda \sum_{i=1}^{n} \mathsf{K}_T(\boldsymbol{z}_i, \boldsymbol{z}_i; \mathcal{S})} + \ln \frac{1}{\delta} \right) + 2L^2 \beta T^2$$

Take $\lambda = 1$, we have

$$\sum_{i=1}^{n} \mathsf{K}_T(\boldsymbol{z}_i, \boldsymbol{z}_i; \mathcal{S}') \leq \ln \underset{\mathcal{S}}{\mathbb{E}}\, e^{\sum_{i=1}^{n} \mathsf{K}_T(\boldsymbol{z}_i, \boldsymbol{z}_i; \mathcal{S})} + 2L^2 \beta T^2 + \ln \frac{1}{\delta}.$$

By Lemma C.2 and $\mathsf{K}_T(\boldsymbol{z}_i, \boldsymbol{z}_i; \mathcal{S}) \leq L^2 T$ in worst case, for any $\delta' \in (0, 1)$,

$$\ln \underset{\mathcal{S}}{\mathbb{E}}\, e^{\sum_{i=1}^{n} \mathsf{K}_T(\boldsymbol{z}_i, \boldsymbol{z}_i; \mathcal{S})} \leq \ln e^{(1-\delta')\left( \mathbb{E}_{\mathcal{S}}\left[ \sum_{i=1}^{n} \mathsf{K}_T(\boldsymbol{z}_i, \boldsymbol{z}_i; \mathcal{S}) \right] + \left( L^2 T + L^2 \beta T^2 \right) \sqrt{2n \log \frac{2}{\delta'}} \right) + \delta' n L^2 T}$$

$$= (1 - \delta') \left( \underset{\mathcal{S}}{\mathbb{E}}\left[ \sum_{i=1}^{n} \mathsf{K}_T(\boldsymbol{z}_i, \boldsymbol{z}_i; \mathcal{S}) \right] + \left( L^2 T + L^2 \beta T^2 \right) \sqrt{2n \log \frac{2}{\delta'}} \right) + \delta' n L^2 T$$

Take $\delta' = \frac{1}{n}$,

$$\ln \underset{\mathcal{S}}{\mathbb{E}}\, e^{\sum_{i=1}^{n} \mathsf{K}_T(\boldsymbol{z}_i, \boldsymbol{z}_i; \mathcal{S})} \leq \underset{\mathcal{S}}{\mathbb{E}}\left[ \sum_{i=1}^{n} \mathsf{K}_T(\boldsymbol{z}_i, \boldsymbol{z}_i; \mathcal{S}) \right] + \left( L^2 T + L^2 \beta T^2 \right) \sqrt{2n \log 2n} + L^2 T$$

Combing with the above, with probability at least $1 - \delta$,

$$\sum_{i=1}^{n} \mathsf{K}_T(\boldsymbol{z}_i, \boldsymbol{z}_i; \mathcal{S}') \leq \underset{\mathcal{S}}{\mathbb{E}}\left[ \sum_{i=1}^{n} \mathsf{K}_T(\boldsymbol{z}_i, \boldsymbol{z}_i; \mathcal{S}) \right] + \left( L^2 T + L^2 \beta T^2 \right) \sqrt{2n \log 2n} + L^2 T + 2L^2 \beta T^2 + \ln \frac{1}{\delta}$$

By Lemma C.2 and a union bound, with probability at least $1 - \delta$,

$$\sum_{i=1}^{n} \mathsf{K}_T(\boldsymbol{z}_i, \boldsymbol{z}_i; \mathcal{S}') \leq \sum_{i=1}^{n} \mathsf{K}_T(\boldsymbol{z}_i, \boldsymbol{z}_i; \mathcal{S}) + \left( L^2 T + L^2 \beta T^2 \right) \left( \sqrt{2n \log 2n} + \sqrt{2n \log \frac{4}{\delta}} \right) + L^2 T$$

$$+ 2L^2 \beta T^2 + \ln \frac{2}{\delta}$$

$$= \sum_{i=1}^{n} \mathsf{K}_T(\boldsymbol{z}_i, \boldsymbol{z}_i; \mathcal{S}) + \tilde{O}(T^2 \sqrt{n}).$$

Because of the symmetry between $\mathcal{S}$ and $\mathcal{S}'$, we also have with probability at least $1 - \delta$,

$$\sum_{i=1}^{n} \mathsf{K}_T(\boldsymbol{z}_i, \boldsymbol{z}_i; \mathcal{S}) \leq \sum_{i=1}^{n} \mathsf{K}_T(\boldsymbol{z}_i, \boldsymbol{z}_i; \mathcal{S}') + \tilde{O}(T^2\sqrt{n}).$$

$\square$

# D Proofs for the Generalization Bound

The following decoupling inequality is a slight variation of a result found for instance in Vershynin [63].

**Lemma D.1** (Decoupling (Theorem 2.4 in [34])). *Let $F$ be a convex function, $\mathcal{D}$ a collection of matrices and $\boldsymbol{\sigma}'$ be an independent copy of $\boldsymbol{\sigma}$, then*

$$\mathbb{E} \sup_{\mathbf{D} \in \mathcal{D}} F\left(\sum_{i \neq j} \sigma_i \sigma_j \mathbf{D}_{ij}\right) \leq \mathbb{E} \sup_{\mathbf{D} \in \mathcal{D}} F\left(4 \sum_{i \neq j} \sigma_i \sigma_j' \mathbf{D}_{ij}\right).$$

**Lemma D.2** (Hoeffding's inequality for Rademacher random variables (Theorem 2.2.5 in [63])). *Let $\sigma_1, \ldots, \sigma_n$ be independent Rademacher random variables, and $\boldsymbol{a} = (a_1, \ldots, a_n) \in \mathbb{R}^n$, then*

$$\mathbb{P}\left(\left|\sum_{i=1}^{n} a_i \sigma_i\right| \geq t\right) \leq 2e^{-\frac{t^2}{2\|\boldsymbol{a}\|_2^2}}.$$

**Lemma 5.1.** *Recalling the Rademacher complexity in Definition 3.1, we have*

$$\hat{\mathcal{R}}_\mathcal{S}(\mathcal{G}_T) \leq \frac{B}{n}\sqrt{\sup_{\mathsf{K}_T(\cdot,\cdot;\mathcal{S}') \in \mathcal{K}_T} \sum_{i=1}^{n} \mathsf{K}_T(\boldsymbol{z}_i, \boldsymbol{z}_i; \mathcal{S}') + 4\Delta\sqrt{6n \ln 2n}} + 8\Delta,$$

*where $\mathcal{G}_T$ and $\mathcal{K}_T$ are defined by* (1) *and Section 5 respectively.*

*Proof.* Recall

$$\mathcal{G}_T = \left\{\ell(\mathcal{A}_T(\mathcal{S}'), \boldsymbol{z}) = \sum_{i=1}^{n} -\frac{1}{n}\mathsf{K}_T(\boldsymbol{z}, \boldsymbol{z}_i'; \mathcal{S}') + \ell(\boldsymbol{w}_0, \boldsymbol{z}) : \mathsf{K}_T(\cdot, \cdot; \mathcal{S}') \in \mathcal{K}_T\right\},$$

$$\mathcal{K}_T = \left\{\mathsf{K}_T(\cdot, \cdot; \mathcal{S}') : \frac{1}{n^2}\sum_{i,j} \mathsf{K}_T(\boldsymbol{z}_i', \boldsymbol{z}_j'; \mathcal{S}') \leq B^2, \mathcal{S}' \in \mathbb{S}' \subseteq \text{supp}(\mu^{\otimes n}), \sup_{\boldsymbol{z}, \boldsymbol{z}'}|\mathsf{K}_T(\boldsymbol{z}, \boldsymbol{z}'; \mathcal{S}')| \leq \Delta\right\}.$$

Suppose $\mathsf{K}_T(\boldsymbol{z}, \boldsymbol{z}'; \mathcal{S}') = \langle \Phi_{\mathcal{S}'}(\boldsymbol{z}), \Phi_{\mathcal{S}'}(\boldsymbol{z}')\rangle$. Define

$$\mathcal{G}_T' = \{g(\boldsymbol{z}) = \langle \boldsymbol{\beta}, \Phi_{\mathcal{S}'}(\boldsymbol{z})\rangle + \ell(\boldsymbol{w}_0, \boldsymbol{z}) : \|\boldsymbol{\beta}\| \leq B, \mathsf{K}_T(\cdot, \cdot; \mathcal{S}') \in \mathcal{K}_T\}. \tag{7}$$

We first show $\mathcal{G}_T \subseteq \mathcal{G}_T'$. For $\forall g(\boldsymbol{z}) \in \mathcal{G}_T$,

$$g(\boldsymbol{z}) = \sum_{i=1}^{n} -\frac{1}{n}\mathsf{K}_T(\boldsymbol{z}, \boldsymbol{z}_i'; \mathcal{S}') + \ell(\boldsymbol{w}_0, \boldsymbol{z})$$

$$= \sum_{i=1}^{n} -\frac{1}{n}\langle \Phi_{\mathcal{S}'}(\boldsymbol{z}), \Phi_{\mathcal{S}'}(\boldsymbol{z}_i')\rangle + \ell(\boldsymbol{w}_0, \boldsymbol{z})$$

$$= \left\langle \Phi_{\mathcal{S}'}(\boldsymbol{z}), \sum_{i=1}^{n} -\frac{1}{n}\Phi_{\mathcal{S}'}(\boldsymbol{z}_i')\right\rangle + \ell(\boldsymbol{w}_0, \boldsymbol{z})$$

$$= \langle \boldsymbol{\beta}_{\mathcal{S}'}, \Phi_{\mathcal{S}'}(\boldsymbol{z})\rangle + \ell(\boldsymbol{w}_0, \boldsymbol{z}),$$

where we denote $\boldsymbol{\beta}_{\mathcal{S}'} = \sum_{i=1}^{n} -\frac{1}{n}\Phi_{\mathcal{S}'}(\boldsymbol{z}_i')$. By definition of $\mathcal{G}_T$, $\|\boldsymbol{\beta}_{\mathcal{S}'}\|^2 = \frac{1}{n^2}\sum_{i,j} \mathsf{K}_T(\boldsymbol{z}_i', \boldsymbol{z}_j'; \mathcal{S}') \leq B^2$. Thus $g(\boldsymbol{z}) \in \mathcal{G}_T'$. Since $\forall g(\boldsymbol{z}) \in \mathcal{G}_T$, $g(\boldsymbol{z}) \in \mathcal{G}_T'$, $\mathcal{G}_T \subseteq \mathcal{G}_T'$.

$\mathcal{G}'_T$ is strictly larger than $\mathcal{G}_T$ because $\boldsymbol{\beta}_{\mathcal{S}'}$ is a fixed vector for a fixed $\mathsf{K}_T(\cdot, \cdot; \mathcal{S}')$ while $\boldsymbol{\beta}$ in $\mathcal{G}'_T$ is a vector of any direction. Then by the property of Rademacher complexity,

$$\hat{\mathcal{R}}_{\mathcal{S}}(\mathcal{G}_T) \leq \hat{\mathcal{R}}_{\mathcal{S}}(\mathcal{G}'_T)$$

$$= \frac{1}{n} \mathbb{E}_{\boldsymbol{\sigma}} \left[ \sup_{g \in \mathcal{G}'_T} \sum_{i=1}^n \sigma_i g(\boldsymbol{z}_i) \right]$$

$$= \frac{1}{n} \mathbb{E}_{\boldsymbol{\sigma}} \left[ \sup_{\mathsf{K}_T(\cdot,\cdot;\mathcal{S}') \in \mathcal{K}_T} \sum_{i=1}^n \sigma_i \left( \langle \boldsymbol{\beta}, \Phi_{\mathcal{S}'}(\boldsymbol{z}_i) \rangle + \ell(\boldsymbol{w}_0, \boldsymbol{z}_i) \right) \right]$$

$$= \frac{1}{n} \mathbb{E}_{\boldsymbol{\sigma}} \left[ \sup_{\mathsf{K}_T(\cdot,\cdot;\mathcal{S}') \in \mathcal{K}_T} \sum_{i=1}^n \sigma_i \langle \boldsymbol{\beta}, \Phi_{\mathcal{S}'}(\boldsymbol{z}_i) \rangle \right] + \frac{1}{n} \mathbb{E}_{\boldsymbol{\sigma}} \left[ \sup_{\mathsf{K}_T(\cdot,\cdot;\mathcal{S}') \in \mathcal{K}_T} \sum_{i=1}^n \sigma_i \ell(\boldsymbol{w}_0, \boldsymbol{z}_i) \right]$$

$$= \frac{1}{n} \mathbb{E}_{\boldsymbol{\sigma}} \left[ \sup_{\mathsf{K}_T(\cdot,\cdot;\mathcal{S}') \in \mathcal{K}_T} \left\langle \boldsymbol{\beta}, \sum_{i=1}^n \sigma_i \Phi_{\mathcal{S}'}(\boldsymbol{z}_i) \right\rangle \right].$$

By the dual norm property, we have

$$\frac{1}{n} \mathbb{E}_{\boldsymbol{\sigma}} \left[ \sup_{\mathsf{K}_T(\cdot,\cdot;\mathcal{S}') \in \mathcal{K}_T} \left\langle \boldsymbol{\beta}, \sum_{i=1}^n \sigma_i \Phi_{\mathcal{S}'}(\boldsymbol{z}_i) \right\rangle \right]$$

$$= \frac{B}{n} \mathbb{E}_{\boldsymbol{\sigma}} \left[ \sup_{\mathsf{K}_T(\cdot,\cdot;\mathcal{S}') \in \mathcal{K}_T} \left\| \sum_{i=1}^n \sigma_i \Phi_{\mathcal{S}'}(\boldsymbol{z}_i) \right\| \right]$$

$$= \frac{B}{n} \mathbb{E}_{\boldsymbol{\sigma}} \left[ \sup_{\mathsf{K}_T(\cdot,\cdot;\mathcal{S}') \in \mathcal{K}_T} \left( \sum_{i=1}^n \sum_{j=1}^n \sigma_i \sigma_j \mathsf{K}_T(\boldsymbol{z}_i, \boldsymbol{z}_j; \mathcal{S}') \right)^{\frac{1}{2}} \right]$$

$$= \frac{B}{n} \mathbb{E}_{\boldsymbol{\sigma}} \left[ \left( \sup_{\mathsf{K}_T(\cdot,\cdot;\mathcal{S}') \in \mathcal{K}_T} \sum_{i=1}^n \sum_{j=1}^n \sigma_i \sigma_j \mathsf{K}_T(\boldsymbol{z}_i, \boldsymbol{z}_j; \mathcal{S}') \right)^{\frac{1}{2}} \right].$$

Then by Jensen's inequality,

$$\frac{B}{n} \mathbb{E}_{\boldsymbol{\sigma}} \left[ \left( \sup_{\mathsf{K}_T(\cdot,\cdot;\mathcal{S}') \in \mathcal{K}_T} \sum_{i=1}^n \sum_{j=1}^n \sigma_i \sigma_j \mathsf{K}_T(\boldsymbol{z}_i, \boldsymbol{z}_j; \mathcal{S}') \right)^{\frac{1}{2}} \right]$$

$$\leq \frac{B}{n} \left( \mathbb{E}_{\boldsymbol{\sigma}} \left[ \sup_{\mathsf{K}_T(\cdot,\cdot;\mathcal{S}') \in \mathcal{K}_T} \sum_{i=1}^n \sum_{j=1}^n \sigma_i \sigma_j \mathsf{K}_T(\boldsymbol{z}_i, \boldsymbol{z}_j; \mathcal{S}') \right] \right)^{\frac{1}{2}} \qquad \text{(Jensen's inequality)}$$

$$= \frac{B}{n} \left( \mathbb{E}_{\boldsymbol{\sigma}} \left[ \sup_{\mathsf{K}_T(\cdot,\cdot;\mathcal{S}') \in \mathcal{K}_T} \left( \sum_{i=1}^n \mathsf{K}_T(\boldsymbol{z}_i, \boldsymbol{z}_i; \mathcal{S}') + \sum_{i \neq j} \sigma_i \sigma_j \mathsf{K}_T(\boldsymbol{z}_i, \boldsymbol{z}_j; \mathcal{S}') \right) \right] \right)^{\frac{1}{2}}$$

$$\leq \frac{B}{n} \left( \mathbb{E}_{\boldsymbol{\sigma}} \left[ \sup_{\mathsf{K}_T(\cdot,\cdot;\mathcal{S}') \in \mathcal{K}_T} \sum_{i=1}^n \mathsf{K}_T(\boldsymbol{z}_i, \boldsymbol{z}_i; \mathcal{S}') + \sup_{\mathsf{K}_T(\cdot,\cdot;\mathcal{S}') \in \mathcal{K}_T} \sum_{i \neq j} \sigma_i \sigma_j \mathsf{K}_T(\boldsymbol{z}_i, \boldsymbol{z}_j; \mathcal{S}') \right] \right)^{\frac{1}{2}}$$

$$= \frac{B}{n} \left( \sup_{\mathsf{K}_T(\cdot,\cdot;\mathcal{S}') \in \mathcal{K}_T} \sum_{i=1}^n \mathsf{K}_T(\boldsymbol{z}_i, \boldsymbol{z}_i; \mathcal{S}') + \mathbb{E}_{\boldsymbol{\sigma}} \left[ \sup_{\mathsf{K}_T(\cdot,\cdot;\mathcal{S}') \in \mathcal{K}_T} \sum_{i \neq j} \sigma_i \sigma_j \mathsf{K}_T(\boldsymbol{z}_i, \boldsymbol{z}_j; \mathcal{S}') \right] \right)^{\frac{1}{2}}.$$

For the second term above, by the decoupling in Lemma D.1, we can obtain that

$$\mathbb{E}_{\boldsymbol{\sigma}} \left[ \sup_{\mathsf{K}_T(\cdot,\cdot;\mathcal{S}') \in \mathcal{K}_T} \sum_{i \neq j} \sigma_i \sigma_j \mathsf{K}_T(\boldsymbol{z}_i, \boldsymbol{z}_j; \mathcal{S}') \right] \leq \mathbb{E}_{\boldsymbol{\sigma}, \boldsymbol{\sigma}'} \left[ \sup_{\mathsf{K}_T(\cdot,\cdot;\mathcal{S}') \in \mathcal{K}_T} 4 \sum_{i=1}^n \sigma_i \sum_{j \neq i} \sigma'_j \mathsf{K}_T(\boldsymbol{z}_i, \boldsymbol{z}_j; \mathcal{S}') \right].$$

Since $|\mathsf{K}_T(z_i, z_j; \mathcal{S}')| \le \Delta$, by Lemma D.2, for any fixed $i$, with probability at least $1 - \delta'$,

$$\left| \sum_{j \ne i} \sigma'_j \mathsf{K}_T(z_i, z_j; \mathcal{S}') \right| \le \Delta \sqrt{2n \ln \frac{2}{\delta'}}.$$

By a union bound, for all $i \in [n]$, we know that

$$\left| \sum_{j \ne i} \sigma'_j \mathsf{K}_T(z_i, z_j; \mathcal{S}') \right| \le \Delta \sqrt{2n \ln \frac{2n}{\delta'}}.$$

Conditioned on this, by Lemma D.2, with probability at least $(1 - \delta'')(1 - \delta')$,

$$\sum_{i=1}^{n} \sigma_i \sum_{j \ne i} \sigma'_j \mathsf{K}_T(z_i, z_j; \mathcal{S}') \le 2\Delta n \sqrt{\ln \frac{2n}{\delta'} \ln \frac{2}{\delta''}}.$$

For the left $1 - (1 - \delta'')(1 - \delta')$ portion, in the worst case we have

$$\sum_{i=1}^{n} \sigma_i \sum_{j \ne i} \sigma'_j \mathsf{K}_T(z_i, z_j; \mathcal{S}') \le n(n-1)\Delta.$$

Combining these two cases, we can bound the expectation as

$$\mathbb{E}_{\boldsymbol{\sigma}} \left[ \sup_{\mathsf{K}_T(\cdot, \cdot; \mathcal{S}') \in \mathcal{K}_T} \sum_{i \ne j} \sigma_i \sigma_j \mathsf{K}_T(z_i, z_j; \mathcal{S}') \right]$$

$$\le \mathbb{E}_{\boldsymbol{\sigma}, \boldsymbol{\sigma}'} \left[ \sup_{\mathsf{K}_T(\cdot, \cdot; \mathcal{S}') \in \mathcal{K}_T} 4 \sum_{i=1}^{n} \sigma_i \sum_{j \ne i} \sigma'_j \mathsf{K}_T(z_i, z_j; \mathcal{S}') \right]$$

$$\le (1 - \delta'')(1 - \delta') 4\Delta \sqrt{2n \ln \frac{2n}{\delta'}} + (\delta' + \delta'' - \delta'\delta'') 4n(n-1)\Delta$$

$$\le 4\Delta \sqrt{6n \ln 2n} + 8\Delta \qquad \qquad \text{(take } \delta' = \delta'' = \frac{1}{n^2})$$

Therefore, in total we have

$$\hat{\mathcal{R}}_{\mathcal{S}}(\mathcal{G}_T) \le \hat{\mathcal{R}}_{\mathcal{S}}(\mathcal{G}'_T) \le \frac{B}{n} \sqrt{\sup_{\mathsf{K}_T(\cdot, \cdot; \mathcal{S}') \in \mathcal{K}_T} \sum_{i=1}^{n} \mathsf{K}_T(z_i, z_i; \mathcal{S}') + 4\Delta \sqrt{6n \ln 2n} + 8\Delta}.$$

$\square$

**Theorem 5.2.** *Under Assumption 4.2, with probability at least $1 - \delta$ over the randomness of $\mathcal{S}$,*

$$L_\mu(\mathcal{A}_T(\mathcal{S})) - L_{\mathcal{S}}(\mathcal{A}_T(\mathcal{S})) \le \frac{2}{n^2} \sqrt{\sum_{i=1}^{n} \sum_{j=1}^{n} \mathsf{K}_T(z_i, z_j; \mathcal{S})} \sqrt{\sum_{i=1}^{n} \mathsf{K}_T(z_i, z_i; \mathcal{S})} + 3\sqrt{\frac{\ln(4n/\delta)}{2n}} + \epsilon,$$

*where* $\epsilon = \begin{cases} \tilde{O}(\frac{\sqrt{T}}{n^{\frac{3}{4}}}), & \text{S.C.,} \\ \min\left\{ \tilde{O}(\frac{T}{n^{\frac{3}{4}}}), O(\sqrt{\frac{T}{n}}) \right\}, & \text{convex,} \\ \min\left\{ \tilde{O}(\frac{e^{\frac{T}{2}}}{n^{\frac{3}{4}}}), O(\sqrt{\frac{T}{n}}) \right\}, & \text{non-convex.} \end{cases}$

*Proof.* Since $|\mathsf{K}_T(z, z; \mathcal{S})| \le L^2 T$ by the Lipschitz assumption, we can take $\Delta = L^2 T$ such that for all $\mathcal{S}' \in \operatorname{supp}(\mu^{\otimes n})$,

$$\sup_{z, z'} |\mathsf{K}_T(z, z'; \mathcal{S}')| \le \Delta.$$

By Lemma 4.5, we know with probability at least $1 - \delta$ over the randomness of $\mathcal{S}$ and $\mathcal{S}'$,

$$\sum_{i=1}^{n} \mathsf{K}_T(\boldsymbol{z}_i, \boldsymbol{z}_i; \mathcal{S}') \leq \kappa \triangleq \sum_{i=1}^{n} \mathsf{K}_T(\boldsymbol{z}_i, \boldsymbol{z}_i; \mathcal{S}) + \tilde{O}(T^2\sqrt{n}),$$

for convex loss. Conditioned on this, we can find a set $\mathbb{S}' \subseteq \text{supp}(\mu^{\otimes n})$ for dataset $\mathcal{S}'$ such that

$$\sup_{\mathsf{K}_T(\cdot, \cdot; \mathcal{S}') \in \mathcal{K}_T} \sum_{i=1}^{n} \mathsf{K}_T(\boldsymbol{z}_i, \boldsymbol{z}_i; \mathcal{S}') \leq \kappa.$$

Also, take $B^2 = \frac{1}{n^2} \sum_{i,j} \mathsf{K}_T(\boldsymbol{z}_i, \boldsymbol{z}_j; \mathcal{S})$. Therefore, with probability at least $1 - \delta$, we have $\ell(\mathcal{A}_T(\mathcal{S}), \boldsymbol{z}) \in \mathcal{G}_T^B$, where $\mathcal{G}_T^B$ denotes $\mathcal{G}_T$ taking values of $B, \Delta$, and $\mathbb{S}'$.

Note $B^2 = \frac{1}{n^2} \sum_{i,j} \mathsf{K}_T(\boldsymbol{z}_i, \boldsymbol{z}_j; \mathcal{S}) = \int_0^T \|\nabla_{\boldsymbol{w}} L_{\mathcal{S}}(\boldsymbol{w}_t)\|^2 \, dt = L_{\mathcal{S}}(\boldsymbol{w}_0) - L_{\mathcal{S}}(\boldsymbol{w}_T) \leq 1$. Since $0 \leq B \leq 1$, let $B_i = \frac{1}{n}, \frac{2}{n}, \ldots, 1$. We have simultaneously for every $B_i$ that

$$\hat{\mathcal{R}}_{\mathcal{S}}(\mathcal{G}_T^{B_i}) \leq \frac{B_i}{n}\sqrt{\kappa + 4\Delta\sqrt{6n \ln 2n} + 8\Delta}.$$

Let $B_i^*$ be the number such that

$$\frac{1}{n}\sqrt{\sum_{i=1}^{n}\sum_{j=1}^{n} \mathsf{K}_T(\boldsymbol{z}_i, \boldsymbol{z}_j; \mathcal{S})} \leq B_i^* \leq \frac{1}{n}\sqrt{\sum_{i=1}^{n}\sum_{j=1}^{n} \mathsf{K}_T(\boldsymbol{z}_i, \boldsymbol{z}_j; \mathcal{S})} + \frac{1}{n}.$$

We have

$$\hat{\mathcal{R}}_{\mathcal{S}}(\mathcal{G}_T^{B_i^*})$$
$$\leq \frac{B_i^*}{n}\sqrt{\kappa + 4\Delta\sqrt{6n \ln 2n} + 8\Delta}$$
$$\leq \frac{1}{n}\left(\frac{1}{n}\sqrt{\sum_{i=1}^{n}\sum_{j=1}^{n} \mathsf{K}_T(\boldsymbol{z}_i, \boldsymbol{z}_j; \mathcal{S})} + \frac{1}{n}\right)\sqrt{\sum_{i=1}^{n} \mathsf{K}_T(\boldsymbol{z}_i, \boldsymbol{z}_i; \mathcal{S}) + \tilde{O}(T^2\sqrt{n}) + \tilde{O}(T\sqrt{n})}$$
$$\leq \frac{1}{n}\left(\frac{1}{n}\sqrt{\sum_{i=1}^{n}\sum_{j=1}^{n} \mathsf{K}_T(\boldsymbol{z}_i, \boldsymbol{z}_j; \mathcal{S})} + \frac{1}{n}\right)\left(\sqrt{\sum_{i=1}^{n} \mathsf{K}_T(\boldsymbol{z}_i, \boldsymbol{z}_i; \mathcal{S})} + \tilde{O}(Tn^{\frac{1}{4}})\right)$$
$$\leq \frac{1}{n^2}\sqrt{\sum_{i=1}^{n}\sum_{j=1}^{n} \mathsf{K}_T(\boldsymbol{z}_i, \boldsymbol{z}_j; \mathcal{S})}\sqrt{\sum_{i=1}^{n} \mathsf{K}_T(\boldsymbol{z}_i, \boldsymbol{z}_i; \mathcal{S})} + \tilde{O}(\frac{T}{n^{\frac{3}{4}}}).$$

Since $\mathsf{K}_T(\boldsymbol{z}, \boldsymbol{z}; \mathcal{S}) \leq L^2 T$ by the Lipschitz assumption, we also have

$$\sup_{\mathsf{K}_T(\cdot, \cdot; \mathcal{S}') \in \mathcal{K}_T} \sum_{i=1}^{n} \mathsf{K}_T(\boldsymbol{z}_i, \boldsymbol{z}_i; \mathcal{S}') \leq \sum_{i=1}^{n} \mathsf{K}_T(\boldsymbol{z}_i, \boldsymbol{z}_i; \mathcal{S}) + L^2 T n.$$

From this, we can conclude that

$$\hat{\mathcal{R}}_{\mathcal{S}}(\mathcal{G}_T^{B_i^*}) \leq \frac{1}{n^2}\sqrt{\sum_{i=1}^{n}\sum_{j=1}^{n} \mathsf{K}_T(\boldsymbol{z}_i, \boldsymbol{z}_j; \mathcal{S})}\sqrt{\sum_{i=1}^{n} \mathsf{K}_T(\boldsymbol{z}_i, \boldsymbol{z}_i; \mathcal{S})} + O(\sqrt{\frac{T}{n}}).$$

Therefore,

$$\hat{\mathcal{R}}_{\mathcal{S}}(\mathcal{G}_T^{B_i^*}) \leq \frac{1}{n^2}\sqrt{\sum_{i=1}^{n}\sum_{j=1}^{n} \mathsf{K}_T(\boldsymbol{z}_i, \boldsymbol{z}_j; \mathcal{S})}\sqrt{\sum_{i=1}^{n} \mathsf{K}_T(\boldsymbol{z}_i, \boldsymbol{z}_i; \mathcal{S})} + \min\left\{\tilde{O}(\frac{T}{n^{\frac{3}{4}}}), O(\sqrt{\frac{T}{n}})\right\}.$$

By Theorem 3.2 and applying a union bound over $B_i = \frac{1}{n}, \frac{2}{n}, \ldots, 1$, with probability at least $1 - \delta$ over the randomness of $\mathcal{S}$, for all $B_i$,

$$\sup_{g \in \mathcal{G}_T^{B_i}} \{L_\mu(g) - L_\mathcal{S}(g)\} \leq 2\hat{\mathcal{R}}_\mathcal{S}(\mathcal{G}_T^{B_i}) + 3\sqrt{\frac{\ln(2n/\delta)}{2n}}.$$

Finally, taking a union bound, we know that with probability at least $1 - 2\delta$, for some $B_i^*$, the following three conditions hold:

$$\ell(\mathcal{A}_T(\mathcal{S}), \boldsymbol{z}) \in \mathcal{G}_T^{B_i^*},$$

$$\hat{\mathcal{R}}_\mathcal{S}(\mathcal{G}_T^{B_i^*}) \leq \frac{1}{n^2}\sqrt{\sum_{i=1}^n \sum_{j=1}^n \mathsf{K}_T(\boldsymbol{z}_i, \boldsymbol{z}_j; \mathcal{S})}\sqrt{\sum_{i=1}^n \mathsf{K}_T(\boldsymbol{z}_i, \boldsymbol{z}_i; \mathcal{S})} + \min\left\{\tilde{O}(\frac{T}{n^{\frac{3}{4}}}), O(\sqrt{\frac{T}{n}})\right\},$$

$$\sup_{g \in \mathcal{G}_T^{B_i^*}} \{L_\mu(g) - L_\mathcal{S}(g)\} \leq 2\hat{\mathcal{R}}_\mathcal{S}(\mathcal{G}_T^{B_i^*}) + 3\sqrt{\frac{\ln(2n/\delta)}{2n}}.$$

These together imply that with probability at least $1 - \delta$, we have

$$L_\mu(\mathcal{A}_T(\mathcal{S})) - L_\mathcal{S}(\mathcal{A}_T(\mathcal{S})) \leq \frac{2}{n^2}\sqrt{\sum_{i=1}^n \sum_{j=1}^n \mathsf{K}_T(\boldsymbol{z}_i, \boldsymbol{z}_j; \mathcal{S})}\sqrt{\sum_{i=1}^n \mathsf{K}_T(\boldsymbol{z}_i, \boldsymbol{z}_i; \mathcal{S})}$$
$$+ 3\sqrt{\frac{\ln(4n/\delta)}{2n}} + \min\left\{\tilde{O}(\frac{T}{n^{\frac{3}{4}}}), O(\sqrt{\frac{T}{n}})\right\}.$$

$\square$

# E   A lower bound of $\hat{\mathcal{R}}_\mathcal{S}(\mathcal{G}_T')$

Here we give a lower bound of $\hat{\mathcal{R}}_\mathcal{S}(\mathcal{G}_T')$. Similar lower bounds for a linear model were proved in [8, 11] without the supremum. Our lower bound matches the upper bound, which shows the bound is nearly optimal for $\hat{\mathcal{R}}_\mathcal{S}(\mathcal{G}_T')$.

**Theorem E.1.** *Recall the function class $\mathcal{G}_T'$ defined in (7). We have*

$$\hat{\mathcal{R}}_\mathcal{S}(\mathcal{G}_T') \geq \frac{B}{\sqrt{2}n} \sup_{\mathsf{K}_T(\cdot, \cdot; \mathcal{S}') \in \mathcal{K}_T} \sqrt{\sum_{i=1}^n \mathsf{K}_T(\boldsymbol{z}_i, \boldsymbol{z}_i; \mathcal{S}')}.$$

*Proof.* Recall

$$\mathcal{G}_T' = \{g(\boldsymbol{z}) = \langle \boldsymbol{\beta}, \Phi_{\mathcal{S}'}(\boldsymbol{z}) \rangle + \ell(\boldsymbol{w}_0, \boldsymbol{z}) : \|\boldsymbol{\beta}\| \leq B, \mathsf{K}_T(\cdot, \cdot; \mathcal{S}') \in \mathcal{K}_T\}.$$

The Rademacher complexity of $\mathcal{G}_T'$ is

$$\hat{\mathcal{R}}_\mathcal{S}(\mathcal{G}_T') = \frac{1}{n} \underset{\boldsymbol{\sigma}}{\mathbb{E}} \left[ \sup_{g \in \mathcal{G}_T'} \sum_{i=1}^n \sigma_i g(\boldsymbol{z}_i) \right]$$

$$= \frac{1}{n} \underset{\boldsymbol{\sigma}}{\mathbb{E}} \left[ \sup_{\mathsf{K}_T(\cdot, \cdot; \mathcal{S}') \in \mathcal{K}_T} \sup_{\|\boldsymbol{\beta}\| \leq B} \sum_{i=1}^n \sigma_i \left( \langle \boldsymbol{\beta}, \Phi_{\mathcal{S}'}(\boldsymbol{z}_i) \rangle + \ell(\boldsymbol{w}_0, \boldsymbol{z}_i) \right) \right]$$

$$= \frac{1}{n} \underset{\boldsymbol{\sigma}}{\mathbb{E}} \left[ \sup_{\mathsf{K}_T(\cdot, \cdot; \mathcal{S}') \in \mathcal{K}_T} \sup_{\|\boldsymbol{\beta}\| \leq B} \left\langle \boldsymbol{\beta}, \sum_{i=1}^n \sigma_i \Phi_{\mathcal{S}'}(\boldsymbol{z}_i) \right\rangle \right] + \underset{\boldsymbol{\sigma}}{\mathbb{E}} \left[ \sum_{i=1}^n \sigma_i \ell(\boldsymbol{w}_0, \boldsymbol{z}_i) \right]$$

$$= \frac{1}{n} \underset{\boldsymbol{\sigma}}{\mathbb{E}} \left[ \sup_{\mathsf{K}_T(\cdot, \cdot; \mathcal{S}') \in \mathcal{K}_T} \sup_{\|\boldsymbol{\beta}\| \leq B} \left\langle \boldsymbol{\beta}, \sum_{i=1}^n \sigma_i \Phi_{\mathcal{S}'}(\boldsymbol{z}_i) \right\rangle \right]$$

$$= \frac{B}{n} \underset{\boldsymbol{\sigma}}{\mathbb{E}} \left[ \sup_{\mathsf{K}_T(\cdot, \cdot; \mathcal{S}') \in \mathcal{K}_T} \left\| \sum_{i=1}^n \sigma_i \Phi_{\mathcal{S}'}(\boldsymbol{z}_i) \right\| \right],$$

where in the last line we apply the dual norm property. Then by the subadditivity of the supremum, we have

$$
\hat{\mathcal{R}}_{\mathcal{S}}(\mathcal{G}_T') \geq \frac{B}{n} \sup_{\mathsf{K}_T(\cdot,\cdot;\mathcal{S}')\in\mathcal{K}_T} \mathbb{E}_{\sigma}\left[\left\|\sum_{i=1}^{n} \sigma_i \Phi_{\mathcal{S}'}(\boldsymbol{z}_i)\right\|\right]
$$

$$
\geq \frac{B}{n} \sup_{\mathsf{K}_T(\cdot,\cdot;\mathcal{S}')\in\mathcal{K}_T} \left\|\mathbb{E}_{\sigma}\left[\left|\sum_{i=1}^{n} \sigma_i \Phi_{\mathcal{S}'}(\boldsymbol{z}_i)\right|\right]\right\| \qquad \text{(norm sub-additivity)}
$$

$$
= \frac{B}{n} \sup_{\mathsf{K}_T(\cdot,\cdot;\mathcal{S}')\in\mathcal{K}_T} \left(\sum_{j\in\mathbb{N}_+} \left(\mathbb{E}_{\sigma}\left[\left|\sum_{i=1}^{n} \sigma_i [\Phi_{\mathcal{S}'}(\boldsymbol{z}_i)]_j\right|\right]\right)^2\right)^{\frac{1}{2}} \qquad \text{(by the definition of 2-norm)}
$$

$$
\geq \frac{B}{n} \sup_{\mathsf{K}_T(\cdot,\cdot;\mathcal{S}')\in\mathcal{K}_T} \left(\sum_{j\in\mathbb{N}_+} \left(\frac{1}{\sqrt{2}}\left|\sum_{i=1}^{n} [\Phi_{\mathcal{S}'}(\boldsymbol{z}_i)]_j^2\right|^{\frac{1}{2}}\right)^2\right)^{\frac{1}{2}}
$$

$$
\text{(Khintchine-Kahane inequality)}
$$

$$
= \frac{B}{\sqrt{2}n} \sup_{\mathsf{K}_T(\cdot,\cdot;\mathcal{S}')\in\mathcal{K}_T} \left(\sum_{j\in\mathbb{N}_+} \left|\sum_{i=1}^{n} [\Phi_{\mathcal{S}'}(\boldsymbol{z}_i)]_j^2\right|\right)^{\frac{1}{2}}
$$

$$
= \frac{B}{\sqrt{2}n} \sup_{\mathsf{K}_T(\cdot,\cdot;\mathcal{S}')\in\mathcal{K}_T} \left(\sum_{i=1}^{n} \sum_{j\in\mathbb{N}_+} [\Phi_{\mathcal{S}'}(\boldsymbol{z}_i)]_j^2\right)^{\frac{1}{2}} \qquad \text{(rearrange the summations)}
$$

$$
= \frac{B}{\sqrt{2}n} \sup_{\mathsf{K}_T(\cdot,\cdot;\mathcal{S}')\in\mathcal{K}_T} \left(\sum_{i=1}^{n} \|\Phi_{\mathcal{S}'}(\boldsymbol{z}_i)\|^2\right)^{\frac{1}{2}}
$$

$$
= \frac{B}{\sqrt{2}n} \sup_{\mathsf{K}_T(\cdot,\cdot;\mathcal{S}')\in\mathcal{K}_T} \sqrt{\sum_{i=1}^{n} \mathsf{K}_T(\boldsymbol{z}_i, \boldsymbol{z}_i; \mathcal{S}')}.
$$

Hence, we complete the proof of this theorem. $\qquad\square$

# F  Stochastic Gradient Flow

In the previous section, we derived a generalization bound for NNs trained from full-batch gradient flow. Here we extend our analysis to stochastic gradient flow and derive a corresponding generalization bound. To start with, we recall the dynamics of stochastic gradient flow (SGD with infinitesimal step size).

$$\frac{d\boldsymbol{w}_t}{dt} = -\nabla_{\boldsymbol{w}} L_{\mathcal{S}_t}(\boldsymbol{w}_t) = -\frac{1}{m} \sum_{i \in \mathcal{S}_t} \nabla_{\boldsymbol{w}} \ell(\boldsymbol{w}_t, \boldsymbol{z}_i)$$

where $\mathcal{S}_t \subseteq \{1, \ldots, n\}$ be the indices of batch data used in time interval $[t, t+1]$ and $|\mathcal{S}_t| = m$ be the batch size. Suppose each $\mathcal{S}_t$ is uniformly sampled without replacement from $\{1, \ldots, n\}$. We recall the connection between the loss dynamics of stochastic gradient flow and a general kernel machine in [20].

**Theorem F.1** (Theorem 4 in [20]). *Suppose $\boldsymbol{w}(T) = \boldsymbol{w}_T$ is a solution of stochastic gradient flow at time $T \in \mathbb{N}$ with initialization $\boldsymbol{w}(0) = \boldsymbol{w}_0$. Then for any $\boldsymbol{z} \in \mathcal{Z}$,*

$$\ell(\boldsymbol{w}_T, \boldsymbol{z}) = \sum_{t=0}^{T-1} \sum_{i \in \mathcal{S}_t} -\frac{1}{m} \mathsf{K}_{t,t+1}(\boldsymbol{z}, \boldsymbol{z}_i; \mathcal{S}) + \ell(\boldsymbol{w}_0, \boldsymbol{z}),$$

*where $\mathsf{K}_{t,t+1}(\boldsymbol{z}, \boldsymbol{z}_i; \mathcal{S}) = \int_t^{t+1} \langle \nabla_{\boldsymbol{w}} \ell(\boldsymbol{w}_t, \boldsymbol{z}), \nabla_{\boldsymbol{w}} \ell(\boldsymbol{w}_t, \boldsymbol{z}_i) \rangle \, dt$ is the LPK over time interval $[t, t+1]$.*

## F.1  Stability of Stochastic Gradient Flow (SGF)

**Lemma F.2.** *Suppose $L_{\mathcal{S}}(\boldsymbol{w})$ is convex for any $\mathcal{S}$ and Assumption 4.2 holds. For any two data sets $\mathcal{S}$ and $\mathcal{S}^{(i)}$, let $\boldsymbol{w}_t = \mathcal{A}_t(\mathcal{S})$ and $\boldsymbol{w}'_t = \mathcal{A}_t(\mathcal{S}^{(i)}))$ be the parameters trained with SGF from same initialization $\boldsymbol{w}_0 = \boldsymbol{w}'_0$, then*

$$\mathop{\mathbb{E}}_{\mathcal{A}_t} \|\boldsymbol{w}_t - \boldsymbol{w}'_t\| \leq \frac{2Lt}{n}.$$

*where the expectation is taken over the randomness of sampling the data batches $\mathcal{S}_t$.*

*Proof.* Notice that

$$\frac{d \|\boldsymbol{w}_t - \boldsymbol{w}'_t\|^2}{dt}$$
$$= \left\langle \frac{\partial \|\boldsymbol{w}_t - \boldsymbol{w}'_t\|^2}{\partial(\boldsymbol{w}_t - \boldsymbol{w}'_t)}, \frac{d(\boldsymbol{w}_t - \boldsymbol{w}'_t)}{dt} \right\rangle$$
$$= 2(\boldsymbol{w}_t - \boldsymbol{w}'_t)^\top \frac{d(\boldsymbol{w}_t - \boldsymbol{w}'_t)}{dt}$$
$$= 2(\boldsymbol{w}_t - \boldsymbol{w}'_t)^\top \left( -\nabla_{\boldsymbol{w}} L_{\mathcal{S}_t}(\boldsymbol{w}_t) + \nabla_{\boldsymbol{w}} L_{\mathcal{S}_t^{(i)}}(\boldsymbol{w}'_t) \right)$$

Since $\mathcal{S}_t$ and $\mathcal{S}_t^{(i)}$ are uniformly sampled without replacement, the probability that $\mathcal{S}_t$ and $\mathcal{S}_t^{(i)}$ are different is $\frac{m}{n}$. When $\mathcal{S}_t = \mathcal{S}_t^{(i)}$, by convexity,

$$2(\boldsymbol{w}_t - \boldsymbol{w}'_t)^\top \left( -\nabla_{\boldsymbol{w}} L_{\mathcal{S}_t}(\boldsymbol{w}_t) + \nabla_{\boldsymbol{w}} L_{\mathcal{S}_t}(\boldsymbol{w}'_t) \right) \leq 0.$$

Since also $\frac{d\|\boldsymbol{w}_t - \boldsymbol{w}'_t\|^2}{dt} = 2\|\boldsymbol{w}_t - \boldsymbol{w}'_t\| \frac{d\|\boldsymbol{w}_t - \boldsymbol{w}'_t\|}{dt}$, we have

$$2\|\boldsymbol{w}_t - \boldsymbol{w}'_t\| \frac{d\|\boldsymbol{w}_t - \boldsymbol{w}'_t\|}{dt} \leq 0.$$

Solve the differential equation for $[T-1, T]$, we have

$$\|\boldsymbol{w}_T - \boldsymbol{w}'_T\| \leq \|\boldsymbol{w}_{T-1} - \boldsymbol{w}'_{T-1}\|.$$

When $\mathcal{S}_t$ and $\mathcal{S}'_t$ differ with one data point,

$$2\left(\boldsymbol{w}_t - \boldsymbol{w}'_t\right)^\top \left(-\nabla_{\boldsymbol{w}} L_{\mathcal{S}_t}(\boldsymbol{w}_t) + \nabla_{\boldsymbol{w}} L_{\mathcal{S}_t^{(i)}}(\boldsymbol{w}'_t)\right)$$

$$= 2\left(\boldsymbol{w}_t - \boldsymbol{w}'_t\right)^\top \left(-\nabla_{\boldsymbol{w}} L_{\mathcal{S}_t}(\boldsymbol{w}_t) + \nabla_{\boldsymbol{w}} L_{\mathcal{S}_t^{(i)}}(\boldsymbol{w}_t) - \nabla_{\boldsymbol{w}} L_{\mathcal{S}_t^{(i)}}(\boldsymbol{w}_t) + \nabla_{\boldsymbol{w}} L_{\mathcal{S}_t^{(i)}}(\boldsymbol{w}'_t)\right)$$

$$= \frac{2}{m}\left(\boldsymbol{w}_t - \boldsymbol{w}'_t\right)^\top \left(\nabla_{\boldsymbol{w}}\ell(\boldsymbol{w}_t, \boldsymbol{z}'_i) - \nabla_{\boldsymbol{w}}\ell(\boldsymbol{w}_t, \boldsymbol{z}_i)\right) - 2\left(\boldsymbol{w}_t - \boldsymbol{w}'_t\right)^\top \left(\nabla_{\boldsymbol{w}} L_{\mathcal{S}_t^{(i)}}(\boldsymbol{w}_t) - \nabla_{\boldsymbol{w}} L_{\mathcal{S}_t^{(i)}}(\boldsymbol{w}'_t)\right)$$

$$\leq \frac{2}{m}\left(\boldsymbol{w}_t - \boldsymbol{w}'_t\right)^\top \left(\nabla_{\boldsymbol{w}}\ell(\boldsymbol{w}_t, \boldsymbol{z}'_i) - \nabla_{\boldsymbol{w}}\ell(\boldsymbol{w}_t, \boldsymbol{z}_i)\right) \qquad \text{(convexity)}$$

$$\leq \frac{4L}{m}\left\|\boldsymbol{w}_t - \boldsymbol{w}'_t\right\|.$$

Since also $\frac{d\|\boldsymbol{w}_t - \boldsymbol{w}'_t\|^2}{dt} = 2\|\boldsymbol{w}_t - \boldsymbol{w}'_t\|\frac{d\|\boldsymbol{w}_t - \boldsymbol{w}'_t\|}{dt}$, we have

$$2\|\boldsymbol{w}_t - \boldsymbol{w}'_t\|\frac{d\|\boldsymbol{w}_t - \boldsymbol{w}'_t\|}{dt} \leq \frac{4L}{m}\|\boldsymbol{w}_t - \boldsymbol{w}'_t\|.$$

When $\|\boldsymbol{w}_t - \boldsymbol{w}'_t\| = 0$, the result already hold. When $\|\boldsymbol{w}_t - \boldsymbol{w}'_t\| > 0$,

$$\frac{d\|\boldsymbol{w}_t - \boldsymbol{w}'_t\|}{dt} \leq \frac{2L}{m}.$$

Solve the differential equation for $[T-1, T]$, we have

$$\|\boldsymbol{w}_T - \boldsymbol{w}'_T\| \leq \frac{2L}{m} + \|\boldsymbol{w}_{T-1} - \boldsymbol{w}'_{T-1}\|.$$

Therefore, considering the two cases that whether $\mathcal{S}_t = \mathcal{S}_t^{(i)}$,

$$\mathop{\mathbb{E}}_{\mathcal{A}_T} \|\boldsymbol{w}_T - \boldsymbol{w}'_T\| \leq \frac{m}{n} \cdot \frac{2L}{m} + \left(1 - \frac{m}{n}\right) \cdot 0 + \mathop{\mathbb{E}}_{\mathcal{A}_T} \|\boldsymbol{w}_{T-1} - \boldsymbol{w}'_{T-1}\|$$

$$= \frac{2L}{n} + \mathop{\mathbb{E}}_{\mathcal{A}_T} \|\boldsymbol{w}_{T-1} - \boldsymbol{w}'_{T-1}\|$$

$$= \frac{2LT}{n}.$$

Thus, we complete the proof of this lemma.

$\square$

The proofs for strongly convex and nonconvex cases are analogous to those of full-batch gradient flow. Consequently, we omit the proof for strongly convex and proceed directly with the proof for the nonconvex case.

**Lemma F.3.** *Suppose $L_{\mathcal{S}}(\boldsymbol{w})$ is $\gamma$-strongly convex for any $\mathcal{S}$ and Assumption 4.2 holds. For any two data sets $\mathcal{S}$ and $\mathcal{S}^{(i)}$, let $\boldsymbol{w}_t = \mathcal{A}_t(\mathcal{S})$ and $\boldsymbol{w}'_t = \mathcal{A}_t(\mathcal{S}^{(i)})$ be the parameters trained with SGF from same initialization $\boldsymbol{w}_0 = \boldsymbol{w}'_0$, then*

$$\mathop{\mathbb{E}}_{\mathcal{A}_t} \|\boldsymbol{w}_t - \boldsymbol{w}'_t\| \leq \frac{2L}{\gamma n}.$$

*where the expectation is taken over the randomness of sampling the data batches $\mathcal{S}_t$.*

**Lemma F.4.** *Suppose $L_{\mathcal{S}}(\boldsymbol{w})$ is non-convex for any $\mathcal{S}$ and Assumption 4.2 holds. For any two data sets $\mathcal{S}$ and $\mathcal{S}^{(i)}$, let $\boldsymbol{w}_t = \mathcal{A}_t(\mathcal{S})$ and $\boldsymbol{w}'_t = \mathcal{A}_t(\mathcal{S}^{(i)})$ be the parameters trained with SGF from same initialization $\boldsymbol{w}_0 = \boldsymbol{w}'_0$, then*

$$\mathop{\mathbb{E}}_{\mathcal{A}_t} \|\boldsymbol{w}_t - \boldsymbol{w}'_t\| \leq \frac{2L}{\beta n}(e^{\beta t} - 1).$$

*where the expectation is taken over the randomness of sampling the data batches $\mathcal{S}_t$.*

*Proof.* Notice that

$$\frac{d\,\|\boldsymbol{w}_t - \boldsymbol{w}_t'\|^2}{dt}$$

$$= \left\langle \frac{\partial\,\|\boldsymbol{w}_t - \boldsymbol{w}_t'\|^2}{\partial(\boldsymbol{w}_t - \boldsymbol{w}_t')}, \frac{d\,(\boldsymbol{w}_t - \boldsymbol{w}_t')}{dt} \right\rangle$$

$$= 2\,(\boldsymbol{w}_t - \boldsymbol{w}_t')^\top \frac{d\,(\boldsymbol{w}_t - \boldsymbol{w}_t')}{dt}$$

$$= 2\,(\boldsymbol{w}_t - \boldsymbol{w}_t')^\top \left( -\nabla_{\boldsymbol{w}} L_{\mathcal{S}_t}(\boldsymbol{w}_t) + \nabla_{\boldsymbol{w}} L_{\mathcal{S}_t^{(i)}}(\boldsymbol{w}_t') \right).$$

When $\mathcal{S}_t = \mathcal{S}_t^{(i)}$, by the smoothness,

$$2\,(\boldsymbol{w}_t - \boldsymbol{w}_t')^\top \left( -\nabla_{\boldsymbol{w}} L_{\mathcal{S}_t}(\boldsymbol{w}_t) + \nabla_{\boldsymbol{w}} L_{\mathcal{S}_t}(\boldsymbol{w}_t') \right)$$

$$\leq 2\,\|\boldsymbol{w}_t - \boldsymbol{w}_t'\|\,\|-\nabla_{\boldsymbol{w}} L_{\mathcal{S}_t}(\boldsymbol{w}_t) + \nabla_{\boldsymbol{w}} L_{\mathcal{S}_t}(\boldsymbol{w}_t')\|$$

$$\leq 2\beta\,\|\boldsymbol{w}_t - \boldsymbol{w}_t'\|^2\,.$$

Again, because of $\frac{d\|\boldsymbol{w}_t - \boldsymbol{w}_t'\|^2}{dt} = 2\,\|\boldsymbol{w}_t - \boldsymbol{w}_t'\|\frac{d\|\boldsymbol{w}_t - \boldsymbol{w}_t'\|}{dt}$, we have

$$\frac{d\,\|\boldsymbol{w}_t - \boldsymbol{w}_t'\|}{dt} \leq \beta\,\|\boldsymbol{w}_t - \boldsymbol{w}_t'\|\,.$$

When $\mathcal{S}_t$ and $\mathcal{S}_t'$ differ with one data point, by a similar argument as the full-batch gradient flow, we have

$$\frac{d\,\|\boldsymbol{w}_t - \boldsymbol{w}_t'\|}{dt} \leq \frac{4L}{m} + 2\beta\,\|\boldsymbol{w}_t - \boldsymbol{w}_t'\|\,.$$

Combining the two cases, we get

$$\frac{d\,\mathbb{E}_{\mathcal{A}_T}\,\|\boldsymbol{w}_t - \boldsymbol{w}_t'\|}{dt} = \mathbb{E}_{\mathcal{A}_T}\,\frac{d\,\|\boldsymbol{w}_t - \boldsymbol{w}_t'\|}{dt}$$

$$\leq \frac{m}{n}\left( \frac{4L}{m} + 2\beta\,\mathbb{E}_{\mathcal{A}_T}\,\|\boldsymbol{w}_t - \boldsymbol{w}_t'\| \right) + (1 - \frac{m}{n}) \cdot \beta\,\mathbb{E}_{\mathcal{A}_T}\,\|\boldsymbol{w}_t - \boldsymbol{w}_t'\|$$

$$= \frac{4L}{n} + 2\beta\,\mathbb{E}_{\mathcal{A}_T}\,\|\boldsymbol{w}_t - \boldsymbol{w}_t'\|\,.$$

Solving the ODE, we get the result.

$\square$

## F.2 Concentrations of LPKs under SGF

For SGF, we can prove similar concentrations of LPKs as Lemma C.1, Lemma 4.4, Lemma C.2, and Lemma 4.5. The proofs are basically the same by simply replacing $\mathsf{K}_T(\boldsymbol{z}, \boldsymbol{z}'; \mathcal{S})$ with $\mathbb{E}_{\mathcal{A}_T}\,\mathsf{K}_{t,t+1}(\boldsymbol{z}, \boldsymbol{z}'; \mathcal{S})$. Hence, we only present the lemmas below. Note here we consider $\mathsf{K}_{t,t+1}$ instead of $\mathsf{K}_T(\boldsymbol{z}, \boldsymbol{z}'; \mathcal{S})$.

**Lemma F.5.** *Let $\mathcal{S}$ and $\mathcal{S}^{(i)}$ be two datasets that only differ in $i$-th data point. Under Assumption 4.2, for any $\boldsymbol{z}, \boldsymbol{z}'$,*

$$\left| \mathbb{E}_{\mathcal{A}_T} \left[ \mathsf{K}_{t,t+1}(\boldsymbol{z}, \boldsymbol{z}'; \mathcal{S}) - \mathsf{K}_{t,t+1}(\boldsymbol{z}, \boldsymbol{z}'; \mathcal{S}^{(i)}) \right] \right| \leq \begin{cases} \frac{4L^2\beta}{\gamma n}, & \gamma\text{-strongly convex,} \\ \frac{2L^2\beta(2t+1)}{n}, & \text{convex,} \\ \frac{4L^2}{\beta n}(e^{\beta(t+1)} - e^{\beta t} - \beta), & \text{non-convex.} \end{cases}$$

**Lemma F.6.** *Under Assumption 4.2, for any fixed $\boldsymbol{z}, \boldsymbol{z}'$, with probability at least $1 - \delta$ over the randomness of $\mathcal{S}'$,*

$$\left| \mathbb{E}_{\mathcal{A}_T} \left[ \mathsf{K}_{t,t+1}(\boldsymbol{z}, \boldsymbol{z}'; \mathcal{S}') - \mathbb{E}_{\mathcal{S}'}\,\mathsf{K}_{t,t+1}(\boldsymbol{z}, \boldsymbol{z}'; \mathcal{S}') \right] \right| \leq \begin{cases} \frac{4L^2\beta}{\gamma}\sqrt{\frac{\ln\frac{2}{\delta}}{2n}}, & \gamma\text{-strongly convex,} \\ 2L^2\beta(2t+1)\sqrt{\frac{\ln\frac{2}{\delta}}{2n}}, & \text{convex,} \\ \frac{4L^2}{\beta}(e^{\beta(t+1)} - e^{\beta t} - \beta)\sqrt{\frac{\ln\frac{2}{\delta}}{2n}}, & \text{non-convex.} \end{cases}$$

**Lemma F.7.** *Under Assumption 4.2, with probability at least $1 - \delta$ over the randomness of $\mathcal{S}$,*

$$\left| \mathbb{E}_{\mathcal{A}_T} \left[ \sum_{i=1}^{n} \mathsf{K}_{t,t+1}(\boldsymbol{z}_i, \boldsymbol{z}_i; \mathcal{S}) - \mathbb{E}_{\mathcal{S}} \left[ \sum_{i=1}^{n} \mathsf{K}_{t,t+1}(\boldsymbol{z}_i, \boldsymbol{z}_i; \mathcal{S}) \right] \right] \right|$$

$$\leq \begin{cases} \left( L^2 + \frac{2L^2\beta}{\gamma} \right) \sqrt{2n \log \frac{2}{\delta}}, & L_S(\boldsymbol{w}) \text{ is } \gamma\text{-strongly convex,} \\ \left( L^2 + L^2\beta(2t+1) \right) \sqrt{2n \log \frac{2}{\delta}}, & L_S(\boldsymbol{w}) \text{ is convex,} \\ \left( L^2 + \frac{2L^2}{\beta} (e^{\beta(t+1)} - e^{\beta t} - \beta) \right) \sqrt{2n \log \frac{2}{\delta}}, & L_S(\boldsymbol{w}) \text{ is non-convex.} \end{cases}$$

**Lemma F.8.** *Under Assumption 4.2, for two datasets $\mathcal{S}$ and $\mathcal{S}'$, with probability at least $1 - \delta$ over the randomness of $\mathcal{S}$ and $\mathcal{S}'$,*

$$\left| \mathbb{E}_{\mathcal{A}_T} \left[ \sum_{i=1}^{n} \mathsf{K}_{t,t+1}(\boldsymbol{z}_i, \boldsymbol{z}_i; \mathcal{S}) - \sum_{i=1}^{n} \mathsf{K}_{t,t+1}(\boldsymbol{z}_i, \boldsymbol{z}_i; \mathcal{S}') \right] \right| \leq \begin{cases} \tilde{O}(\sqrt{n}), & \gamma\text{-strongly convex,} \\ \tilde{O}(t\sqrt{n}), & \text{convex,} \\ \tilde{O}(e^t \sqrt{n}), & \text{non-convex.} \end{cases}$$

### F.3 Generalization bound of SGF

Given a sequence of $\mathcal{S}_0, \ldots, \mathcal{S}_{T-1}$, define the function class of SGF by

$$\mathcal{G}_T \triangleq \left\{ \ell(\mathcal{A}_T(\mathcal{S}'), \boldsymbol{z}) = \sum_{t=0}^{T-1} \sum_{i \in \mathcal{S}_t} -\frac{1}{m} \mathsf{K}_{t,t+1}(\boldsymbol{z}, \boldsymbol{z}_i'; \mathcal{S}') + \ell(\boldsymbol{w}_0, \boldsymbol{z}) : \mathsf{K}(\cdot, \cdot; \mathcal{S}') \in \mathcal{K}_T \right\}$$

where

$$\mathcal{K}_T = \Big\{ (\mathsf{K}_{0,1}(\cdot, \cdot; \mathcal{S}'), \cdots, \mathsf{K}_{T-1,T}(\cdot, \cdot; \mathcal{S}')) : \mathcal{S}' \in \mathsf{supp}(\mu^{\otimes n}),$$
$$\frac{1}{m^2} \sum_{i,j \in \mathcal{S}_t} \mathsf{K}_{t,t+1}(\boldsymbol{z}_i', \boldsymbol{z}_j'; \mathcal{S}') \leq B_t^2, |K_{t,t+1}(\cdot, \cdot; \mathcal{S}')| \leq \Delta \Big\}.$$

**Lemma F.9.** *Given a sequence of $\mathcal{S}_0, \ldots, \mathcal{S}_{T-1}$, we have*

$$\hat{\mathcal{R}}_{\mathcal{S}}(\mathcal{G}_T) \leq \sum_{t=0}^{T-1} \frac{B_t}{n} \left( \sup_{\mathsf{K}(\cdot, \cdot; \mathcal{S}') \in \mathcal{K}_T} \sum_{i=1}^{n} \mathsf{K}_{t,t+1}(\boldsymbol{z}_i, \boldsymbol{z}_i; \mathcal{S}') + 4\Delta\sqrt{6n \ln 2n} + 8\Delta \right)^{\frac{1}{2}}.$$

*Proof.* For $t = 0, 1, \cdots, T-1$, let

$$\mathcal{G}_t = \left\{ g(\boldsymbol{z}) = \sum_{i \in \mathcal{S}_t} -\frac{1}{m} \mathsf{K}_{t,t+1}(\boldsymbol{z}, \boldsymbol{z}_i'; \mathcal{S}') : \mathsf{K}(\cdot, \cdot; \mathcal{S}') \in \mathcal{K}_T \right\},$$

Then we have

$$\mathcal{G}_T \subseteq \mathcal{G}_0 \oplus \mathcal{G}_1 \oplus \cdots \oplus \mathcal{G}_{T-1} \oplus \{\ell(\boldsymbol{w}_0, \boldsymbol{z})\}.$$

Since the set on the RHS involves combinations of kernels induced from distinct training set $\mathcal{S}'$, it is a strictly larger set than the LHS. Apply Lemma 5.1 bound for each $\mathcal{G}_t$ on $\mathcal{S}$,

$$\hat{\mathcal{R}}_{\mathcal{S}}(\mathcal{G}_t) \leq \frac{B_t}{n} \left( \sup_{\mathsf{K}(\cdot, \cdot; \mathcal{S}') \in \mathcal{S}_t} \sum_{i=1}^{n} \mathsf{K}_{t,t+1}(\boldsymbol{z}_i, \boldsymbol{z}_i; \mathcal{S}') + 4\Delta\sqrt{6n \ln 2n} + 8\Delta \right)^{\frac{1}{2}}. \tag{8}$$

By the monotonicity and linear combination of Rademacher complexity [41] and take in (8),

$$\hat{\mathcal{R}}_{\mathcal{S}}(\mathcal{G}_T) \leq \hat{\mathcal{R}}_{\mathcal{S}}(\mathcal{G}_0 \oplus \mathcal{G}_1 \oplus \cdots \oplus \mathcal{G}_{T-1} \oplus \{\ell(\boldsymbol{w}_0, \boldsymbol{z})\})$$

$$= \sum_{t=0}^{T-1} \hat{\mathcal{R}}_{\mathcal{S}}(\mathcal{G}_t) + \hat{\mathcal{R}}_{\mathcal{S}}(\{\ell(\boldsymbol{w}_0, \boldsymbol{z})\})$$

$$\leq \sum_{t=0}^{T-1} \frac{B_t}{n} \left( \sup_{\mathsf{K}(\cdot, \cdot; \mathcal{S}') \in \mathcal{K}_T} \sum_{i=1}^{n} \mathsf{K}_{t,t+1}(\boldsymbol{z}_i, \boldsymbol{z}_i; \mathcal{S}') + 4\Delta\sqrt{6n \ln 2n} + 8\Delta \right)^{\frac{1}{2}}.$$

$\square$

**Theorem 5.4.** *Under Assumption 4.2, for a fixed sequence $\mathcal{S}_0, \ldots, \mathcal{S}_{T-1}$, with probability at least $1 - \delta$ over the randomness of $\mathcal{S}$, the generalization gap of SGF defined by (4) is upper bounded by*

$$L_\mu(\mathcal{A}_T(\mathcal{S})) - L_\mathcal{S}(\mathcal{A}_T(\mathcal{S})) \leq \frac{2}{n} \sum_{t=0}^{T-1} \sqrt{\frac{1}{m^2} \sum_{i,j \in \mathcal{S}_t} \mathsf{K}_{t,t+1}(\boldsymbol{z}_i, \boldsymbol{z}_j; \mathcal{S})} \sqrt{\sum_{i=1}^{n} \mathsf{K}_{t,t+1}(\boldsymbol{z}_i, \boldsymbol{z}_i; \mathcal{S})} + \tilde{O}(\frac{T}{\sqrt{n}}).$$

*Proof.* Let $\Delta = L^2$ in Lemma F.9. Take $B_t^2 = \frac{1}{m^2} \sum_{i,j \in \mathcal{S}_t} \mathsf{K}_{t,t+1}(\boldsymbol{z}_i, \boldsymbol{z}_j; \mathcal{S}) = L_S(\boldsymbol{w}_t) - L_S(\boldsymbol{w}_{t+1})$. Then $\ell(\mathcal{A}_T(\mathcal{S}), \boldsymbol{z}) \in \mathcal{G}_T^{B_0, \ldots, B_{T-1}}$, where $\mathcal{G}_T^{B_0, \ldots, B_{T-1}}$ denotes $\mathcal{G}_T$ taking values of $B_0, \ldots, B_{T-1}$.

By Lipchitz assumption,

$$\sup_{\mathsf{K}(\cdot, \cdot; \mathcal{S}') \in \mathcal{K}_T} \sum_{i=1}^{n} \mathsf{K}_{t,t+1}(\boldsymbol{z}_i, \boldsymbol{z}_i; \mathcal{S}') \leq \sum_{i=1}^{n} \mathsf{K}_{t,t+1}(\boldsymbol{z}_i, \boldsymbol{z}_i; \mathcal{S}) + L^2 n.$$

Since $0 \leq B_t \leq 1$, let $B_t^i = \frac{1}{n}, \frac{2}{n}, \ldots, 1$, $t = 0, \ldots, T-1$. We have simultaneously for every $B_0^i, \ldots, B_{T-1}^i$ that

$$\hat{\mathcal{R}}_\mathcal{S}(\mathcal{G}_T^{B_0^i, \ldots, B_{T-1}^i}) \leq \sum_{t=0}^{T-1} \frac{B_t^i}{n} \sqrt{\sum_{i=1}^{n} \mathsf{K}_{t,t+1}(\boldsymbol{z}_i, \boldsymbol{z}_i; \mathcal{S}) + L^2 n + 4L^2 \sqrt{6n \ln 2n} + 8L^2}.$$

Let $B_t^{i*}$ be the number such that

$$\sqrt{\frac{1}{m^2} \sum_{i,j \in \mathcal{S}_t} \mathsf{K}_{t,t+1}(\boldsymbol{z}_i, \boldsymbol{z}_j; \mathcal{S})} \leq B_t^{i*} \leq \sqrt{\frac{1}{m^2} \sum_{i,j \in \mathcal{S}_t} \mathsf{K}_{t,t+1}(\boldsymbol{z}_i, \boldsymbol{z}_j; \mathcal{S})} + \frac{1}{n}.$$

We have

$$\hat{\mathcal{R}}_\mathcal{S}(\mathcal{G}_T^{B_0^{i*}, \ldots, B_{T-1}^{i*}})$$

$$\leq \sum_{t=0}^{T-1} \frac{B_t^{i*}}{n} \sqrt{\sum_{i=1}^{n} \mathsf{K}_{t,t+1}(\boldsymbol{z}_i, \boldsymbol{z}_i; \mathcal{S}) + O(L^2 n)}$$

$$\leq \sum_{t=0}^{T-1} \frac{1}{n} \left( \sqrt{\frac{1}{m^2} \sum_{i,j \in \mathcal{S}_t} \mathsf{K}_{t,t+1}(\boldsymbol{z}_i, \boldsymbol{z}_j; \mathcal{S})} + \frac{1}{n} \right) \sqrt{\sum_{i=1}^{n} \mathsf{K}_{t,t+1}(\boldsymbol{z}_i, \boldsymbol{z}_i; \mathcal{S}) + O(L^2 n)}$$

$$= \sum_{t=0}^{T-1} \frac{1}{n} \left( \sqrt{\frac{1}{m^2} \sum_{i,j \in \mathcal{S}_t} \mathsf{K}_{t,t+1}(\boldsymbol{z}_i, \boldsymbol{z}_j; \mathcal{S})} \right) \sqrt{\sum_{i=1}^{n} \mathsf{K}_{t,t+1}(\boldsymbol{z}_i, \boldsymbol{z}_i; \mathcal{S})} + O(\frac{T}{\sqrt{n}}).$$

By Theorem 3.2 and applying a union bound over $B_t^i = \frac{1}{n}, \frac{2}{n}, \ldots, 1$, $t = 0, \ldots, T-1$, with probability at least $1 - \delta$, for all $B_i^t$,

$$\sup_{g \in \mathcal{G}_T^{B_0^i, \ldots, B_{T-1}^i}} \{L_\mu(g) - L_\mathcal{S}(g)\} \leq 2\hat{\mathcal{R}}_\mathcal{S}(\mathcal{G}_T^{B_0^i, \ldots, B_{T-1}^i}) + 3\sqrt{\frac{T \ln n + \ln(2/\delta)}{2n}}.$$

These together imply that with probability at least $1 - \delta$,

$$L_\mu(\mathcal{A}_T(\mathcal{S})) - L_\mathcal{S}(\mathcal{A}_T(\mathcal{S})) \leq \frac{2}{n} \sum_{t=0}^{T-1} \sqrt{\frac{1}{m^2} \sum_{i,j \in \mathcal{S}_t} \mathsf{K}_{t,t+1}(\boldsymbol{z}_i, \boldsymbol{z}_j; \mathcal{S})} \sqrt{\sum_{i=1}^{n} \mathsf{K}_{t,t+1}(\boldsymbol{z}_i, \boldsymbol{z}_i; \mathcal{S})} + \tilde{O}(\frac{T}{\sqrt{n}}).$$

$\square$

# G  Proofs for Case Study

## G.1  Overparameterized neural network under NTK regime

Recall the definition of NTK $\hat{\Theta}(\boldsymbol{w}; \boldsymbol{x}, \boldsymbol{x}') = \nabla_{\boldsymbol{w}} f(\boldsymbol{w}, \boldsymbol{x}) \nabla_{\boldsymbol{w}} f(\boldsymbol{w}, \boldsymbol{x}')^{\top} \in \mathbb{R}^{k \times k}$ for a neural network function $f(\boldsymbol{w}, \boldsymbol{x})$. We now prove our bound for the NTK case.

**Corollary 6.1.** *Suppose that* $\lambda_{max}(\hat{\Theta}(\boldsymbol{w}_t; \mathbf{X}, \mathbf{X})) \leq \lambda_{max}$ *and* $\lambda_{min}(\hat{\Theta}(\boldsymbol{w}_t; \mathbf{X}, \mathbf{X})) \geq \lambda_{min} > 0$ *for* $t \in [0, T]$. *Then*

$$\Gamma \leq \sqrt{\frac{2\lambda_{max} \cdot \|f(\boldsymbol{w}_0, \mathbf{X}) - \boldsymbol{y}\|^2}{\lambda_{min} \cdot n}(1 - e^{-\frac{2\lambda_{min}}{n}T})}.$$

*Proof.* Notice by the chain rule,

$$\sum_{i=1}^{n} \|\nabla_{\boldsymbol{w}} \ell(\boldsymbol{w}_t, \boldsymbol{z}_i)\|^2 = \sum_{i=1}^{n} \nabla_f \ell(\boldsymbol{w}_t, \boldsymbol{z}_i)^{\top} \hat{\Theta}(\boldsymbol{w}_t; \boldsymbol{x}_i, \boldsymbol{x}_i) \nabla_f \ell(\boldsymbol{w}_t, \boldsymbol{z}_i).$$

Therefore,

$$\Gamma = \frac{2}{n}\sqrt{L_{\mathcal{S}}(\boldsymbol{w}_0) - L_{\mathcal{S}}(\boldsymbol{w}_T)}\sqrt{\sum_{i=1}^{n} \int_0^T \nabla_f \ell(\boldsymbol{w}_t, \boldsymbol{z}_i)^{\top} \hat{\Theta}(\boldsymbol{w}_t; \boldsymbol{x}_i, \boldsymbol{x}_i) \nabla_f \ell(\boldsymbol{w}_t, \boldsymbol{z}_i) dt}$$

Since the loss is bounded in $[0, 1]$, $L_{\mathcal{S}}(\boldsymbol{w}_0) - L_{\mathcal{S}}(\boldsymbol{w}_T) \leq 1$. When using a mean squre loss $L_{\mathcal{S}}(\boldsymbol{w}_t) = \frac{1}{2n} \|f(\boldsymbol{w}_t, \mathbf{X}) - \boldsymbol{y}\|^2$ and $\ell(\boldsymbol{w}, \boldsymbol{z}) = \frac{1}{2}(f(\boldsymbol{w}, \boldsymbol{x}) - y)^2$,

$$\begin{aligned}
\sum_{i=1}^{n} \|\nabla_{\boldsymbol{w}} \ell(\boldsymbol{w}_t, \boldsymbol{z}_i)\|^2 &= \sum_{i=1}^{n} \hat{\Theta}(\boldsymbol{w}_t; \boldsymbol{x}_i, \boldsymbol{x}_i)(f(\boldsymbol{w}_t, \boldsymbol{x}_i) - \boldsymbol{y}_i)^2 \\
&\leq \max_{i \in [n]} \hat{\Theta}(\boldsymbol{w}_t; \boldsymbol{x}_i, \boldsymbol{x}_i) \|f(\boldsymbol{w}_t, \mathbf{X}) - \boldsymbol{y}\|^2 \\
&\leq \lambda_{\max}(\hat{\Theta}(\boldsymbol{w}_t; \mathbf{X}, \mathbf{X})) \|f(\boldsymbol{w}_t, \mathbf{X}) - \boldsymbol{y}\|^2.
\end{aligned}$$

In the case that the smallest eigenvalue of NTK $\lambda_{\min}(\hat{\Theta}(\boldsymbol{w}_t; \mathbf{X}, \mathbf{X})) \geq \lambda_{\min} > 0$ over the training, the loss converges exponentially $\|f(\boldsymbol{w}_t, \mathbf{X}) - \boldsymbol{y}\|^2 \leq e^{-\frac{2\lambda_{\min}}{n}t} \|f(\boldsymbol{w}_0, \mathbf{X}) - \boldsymbol{y}\|^2$. We can see from

$$\begin{aligned}
\frac{d \|f(\boldsymbol{w}_t, \mathbf{X}) - \boldsymbol{y}\|^2}{dt} &= 2(f(\boldsymbol{w}_t, \mathbf{X}) - \boldsymbol{y})^{\top} \frac{df(\boldsymbol{w}_t, \mathbf{X})}{dt} \\
&= 2(f(\boldsymbol{w}_t, \mathbf{X}) - \boldsymbol{y})^{\top} \nabla_{\boldsymbol{w}} f(\boldsymbol{w}_t, \mathbf{X}) \frac{d\boldsymbol{w}_t}{dt} \\
&= -2(f(\boldsymbol{w}_t, \mathbf{X}) - \boldsymbol{y})^{\top} \nabla_{\boldsymbol{w}} f(\boldsymbol{w}_t, \mathbf{X}) \nabla_{\boldsymbol{w}} L_{\mathcal{S}}(\boldsymbol{w}_t) \\
&= -2(f(\boldsymbol{w}_t, \mathbf{X}) - \boldsymbol{y})^{\top} \nabla_{\boldsymbol{w}} f(\boldsymbol{w}_t, \mathbf{X}) \nabla_{\boldsymbol{w}} f(\boldsymbol{w}_t, \mathbf{X})^{\top} \frac{1}{n}(f(\boldsymbol{w}_t, \mathbf{X}) - \boldsymbol{y}) \\
&= -\frac{2}{n}(f(\boldsymbol{w}_t, \mathbf{X}) - \boldsymbol{y})^{\top} \hat{\Theta}(\boldsymbol{w}_t; \mathbf{X}, \mathbf{X})(f(\boldsymbol{w}_t, \mathbf{X}) - \boldsymbol{y}) \\
&\leq -\frac{2\lambda_{\min}}{n} \|f(\boldsymbol{w}_t, \mathbf{X}) - \boldsymbol{y}\|^2.
\end{aligned}$$

Solving the ODE, we get

$$\|f(\boldsymbol{w}_t, \mathbf{X}) - \boldsymbol{y}\|^2 \leq e^{-\frac{2\lambda_{\min}}{n}t} \|f(\boldsymbol{w}_0, \mathbf{X}) - \boldsymbol{y}\|^2.$$

Then we have

$$\begin{aligned}
\sum_{i=1}^{n} \int_0^T \|\nabla_{\boldsymbol{w}} \ell(\boldsymbol{w}_t, \boldsymbol{z}_i)\|^2 dt &\leq \int_0^T \lambda_{\max} \|f(\boldsymbol{w}_t, \mathbf{X}) - \boldsymbol{y}\|^2 dt \\
&\leq \int_0^T \lambda_{\max} e^{-\frac{2\lambda_{\min}}{n}t} \|f(\boldsymbol{w}_0, \mathbf{X}) - \boldsymbol{y}\|^2 dt \\
&= \frac{n\lambda_{\max} \|f(\boldsymbol{w}_0, \mathbf{X}) - \boldsymbol{y}\|^2}{2\lambda_{\min}}(1 - e^{-\frac{2\lambda_{\min}}{n}T}).
\end{aligned}$$

Plugging this into our bound, we get

$$\Gamma \leq \sqrt{\frac{2\lambda_{\max} \left\| f(\boldsymbol{w}_0, \mathbf{X}) - \boldsymbol{y} \right\|^2}{\lambda_{\min} n} (1 - e^{-\frac{2\lambda_{\min}}{n}T})}.$$

$\square$

## G.2  Kernel Ridge Regression Case

**Corollary 6.2.** *Suppose $K(\boldsymbol{x}_i, \boldsymbol{x}_i) \leq K_{\max}$ for $i \in [n]$ and $K(\mathbf{X}, \mathbf{X})$ is full-rank,*

$$\Gamma \leq \frac{\sqrt{K_{\max}} \left\| \boldsymbol{w}_0 - \boldsymbol{w}^* \right\| \left\| \phi(\mathbf{X})^\top (\boldsymbol{w}_0 - \boldsymbol{w}^*) \right\|}{n}.$$

*When $\boldsymbol{w}_0 = 0$, the bound simplifies to*

$$\Gamma \leq \frac{\sqrt{K_{\max}} \sqrt{\boldsymbol{y}^\top \left( K(\mathbf{X}, \mathbf{X}) \right)^{-1} \boldsymbol{y}} \left\| \boldsymbol{y} \right\|}{n}. \tag{9}$$

*Proof.* The training loss gradient is

$$\begin{aligned}
\nabla_{\boldsymbol{w}} L_{\mathcal{S}}(\boldsymbol{w}_t) &= \frac{1}{n} \phi(\mathbf{X}) \left( \phi(\mathbf{X})^\top \boldsymbol{w}_t - \boldsymbol{y} \right) + \lambda \boldsymbol{w}_t \\
&= \frac{1}{n} \phi(\mathbf{X}) \phi(\mathbf{X})^\top \boldsymbol{w}_t - \frac{1}{n} \phi(\mathbf{X}) \boldsymbol{y} + \lambda \boldsymbol{w}_t \\
&= \left( \frac{1}{n} \phi(\mathbf{X}) \phi(\mathbf{X})^\top + \lambda \mathbf{I} \right) \boldsymbol{w}_t - \frac{1}{n} \phi(\mathbf{X}) \boldsymbol{y}
\end{aligned}$$

from where we can calculate

$$\boldsymbol{w}^* = \frac{1}{n} \left( \frac{1}{n} \phi(\mathbf{X}) \phi(\mathbf{X})^\top + \lambda \mathbf{I}_p \right)^{-1} \phi(\mathbf{X}) \boldsymbol{y} = \frac{1}{n} \phi(\mathbf{X}) \left( \frac{1}{n} \phi(\mathbf{X})^\top \phi(\mathbf{X}) + \lambda \mathbf{I}_n \right)^{-1} \boldsymbol{y}.$$

Thus, we have

$$\begin{aligned}
\frac{d\boldsymbol{w}_t}{dt} &= -\nabla_{\boldsymbol{w}} L_{\mathcal{S}}(\boldsymbol{w}_t) \\
&= -\left( \frac{1}{n} \phi(\mathbf{X}) \phi(\mathbf{X})^\top + \lambda \mathbf{I} \right) \boldsymbol{w}_t + \frac{1}{n} \phi(\mathbf{X}) \boldsymbol{y} \\
&= -\left( \frac{1}{n} \phi(\mathbf{X}) \phi(\mathbf{X})^\top + \lambda \mathbf{I} \right) \left( \boldsymbol{w}_t - \frac{1}{n} \left( \frac{1}{n} \phi(\mathbf{X}) \phi(\mathbf{X})^\top + \lambda \mathbf{I} \right)^{-1} \phi(\mathbf{X}) \boldsymbol{y} \right) \\
&= -\left( \frac{1}{n} \phi(\mathbf{X}) \phi(\mathbf{X})^\top + \lambda \mathbf{I} \right) \left( \boldsymbol{w}_t - \boldsymbol{w}^* \right).
\end{aligned}$$

Therefore,

$$\boldsymbol{w}_t = \boldsymbol{w}^* + e^{-\left( \frac{1}{n} \phi(\mathbf{X}) \phi(\mathbf{X})^\top + \lambda \mathbf{I} \right) t} \left( \boldsymbol{w}_0 - \boldsymbol{w}^* \right).$$

Calculate the norm of the gradient,

$$\begin{aligned}
\left\| \nabla_{\boldsymbol{w}} L_{\mathcal{S}}(\boldsymbol{w}_t) \right\|^2 &= \left\| \left( \frac{1}{n} \phi(\mathbf{X}) \phi(\mathbf{X})^\top + \lambda \mathbf{I} \right) \boldsymbol{w}_t - \frac{1}{n} \phi(\mathbf{X}) \boldsymbol{y} \right\|^2 \\
&= \left\| \left( \frac{1}{n} \phi(\mathbf{X}) \phi(\mathbf{X})^\top + \lambda \mathbf{I} \right) \left( \boldsymbol{w}^* + e^{-\left( \frac{1}{n} \phi(\mathbf{X}) \phi(\mathbf{X})^\top + \lambda \mathbf{I} \right) t} \left( \boldsymbol{w}_0 - \boldsymbol{w}^* \right) \right) - \frac{1}{n} \phi(\mathbf{X}) \boldsymbol{y} \right\|^2 \\
&= \left\| \left( \frac{1}{n} \phi(\mathbf{X}) \phi(\mathbf{X})^\top + \lambda \mathbf{I} \right) e^{-\left( \frac{1}{n} \phi(\mathbf{X}) \phi(\mathbf{X})^\top + \lambda \mathbf{I} \right) t} \left( \boldsymbol{w}_0 - \boldsymbol{w}^* \right) \right\|^2.
\end{aligned}$$

Suppose the eigen-decomposition of $\phi(\mathbf{X})\phi(\mathbf{X})^\top = \sum_{i=1}^p \lambda_i \boldsymbol{u}_i \boldsymbol{u}_i^\top$, then

$$
\begin{aligned}
\|\nabla_{\boldsymbol{w}} L_{\mathcal{S}}(\boldsymbol{w}_t)\|^2 &= \left\| \left( \sum_{i=1}^p \left( \frac{\lambda_i}{n} + \lambda \right) \boldsymbol{u}_i \boldsymbol{u}_i^\top \right) \left( \sum_{i=1}^p e^{-\left( \frac{\lambda_i}{n} + \lambda \right) t} \boldsymbol{u}_i \boldsymbol{u}_i^\top \right) (\boldsymbol{w}_0 - \boldsymbol{w}^*) \right\|^2 \\
&= \left\| \left( \sum_{i=1}^p \left( \frac{\lambda_i}{n} + \lambda \right) e^{-\left( \frac{\lambda_i}{n} + \lambda \right) t} \boldsymbol{u}_i \boldsymbol{u}_i^\top \right) (\boldsymbol{w}_0 - \boldsymbol{w}^*) \right\|^2 \\
&= (\boldsymbol{w}_0 - \boldsymbol{w}^*)^\top \left( \sum_{i=1}^p \left( \frac{\lambda_i}{n} + \lambda \right)^2 e^{-2\left( \frac{\lambda_i}{n} + \lambda \right) t} \boldsymbol{u}_i \boldsymbol{u}_i^\top \right) (\boldsymbol{w}_0 - \boldsymbol{w}^*) \\
&= \sum_{i=1}^p \left( \frac{\lambda_i}{n} + \lambda \right)^2 e^{-2\left( \frac{\lambda_i}{n} + \lambda \right) t} \left( \boldsymbol{u}_i^\top (\boldsymbol{w}_0 - \boldsymbol{w}^*) \right)^2 .
\end{aligned}
$$

Integrate the training loss gradient norm,

$$
\begin{aligned}
\int_0^T \|\nabla_{\boldsymbol{w}} L_{\mathcal{S}}(\boldsymbol{w}_t)\|^2 \, dt &= \int_0^T \sum_{i=1}^p \left( \frac{\lambda_i}{n} + \lambda \right)^2 e^{-2\left( \frac{\lambda_i}{n} + \lambda \right) t} \left( \boldsymbol{u}_i^\top (\boldsymbol{w}_0 - \boldsymbol{w}^*) \right)^2 dt \\
&= \frac{1}{2} \sum_{i=1}^p \left( \frac{\lambda_i}{n} + \lambda \right) \left( 1 - e^{-2\left( \frac{\lambda_i}{n} + \lambda \right) T} \right) \left( \boldsymbol{u}_i^\top (\boldsymbol{w}_0 - \boldsymbol{w}^*) \right)^2 \\
&\leq \frac{1}{2} \sum_{i=1}^p \left( \frac{\lambda_i}{n} + \lambda \right) \left( \boldsymbol{u}_i^\top (\boldsymbol{w}_0 - \boldsymbol{w}^*) \right)^2 \\
&= \frac{1}{2} (\boldsymbol{w}_0 - \boldsymbol{w}^*)^\top \sum_{i=1}^p \left( \frac{\lambda_i}{n} + \lambda \right) \boldsymbol{u}_i \boldsymbol{u}_i^\top (\boldsymbol{w}_0 - \boldsymbol{w}^*) \\
&= \frac{1}{2} (\boldsymbol{w}_0 - \boldsymbol{w}^*)^\top \left( \frac{1}{n} \phi(\mathbf{X})\phi(\mathbf{X})^\top + \lambda \mathbf{I} \right) (\boldsymbol{w}_0 - \boldsymbol{w}^*) .
\end{aligned}
$$

The individual gradient is

$$
\nabla_{\boldsymbol{w}} \ell(\boldsymbol{w}_t, \boldsymbol{z}_i) = \left( \phi(\boldsymbol{x}_i)^\top \boldsymbol{w}_t - y_i \right) \phi(\boldsymbol{x}_i) + \lambda \boldsymbol{w}_t = \left( \phi(\boldsymbol{x}_i)\phi(\boldsymbol{x}_i)^\top + \lambda \mathbf{I} \right) \boldsymbol{w}_t - y_i \phi(\boldsymbol{x}_i).
$$

Assume $K(\boldsymbol{x}_i, \boldsymbol{x}_i) \leq K_{\max}$. When $\lambda = 0$,

$$
\begin{aligned}
\sum_{i=1}^n \|\nabla_{\boldsymbol{w}} \ell(\boldsymbol{w}_t, \boldsymbol{z}_i)\|^2 &= \sum_{i=1}^n \left\| \phi(\boldsymbol{x}_i)\phi(\boldsymbol{x}_i)^\top \boldsymbol{w}_t - y_i \phi(\boldsymbol{x}_i) \right\|^2 \\
&= \sum_{i=1}^n \left\| \phi(\boldsymbol{x}_i)\phi(\boldsymbol{x}_i)^\top \left( \boldsymbol{w}^* + e^{-\frac{1}{n} \phi(\mathbf{X})\phi(\mathbf{X})^\top t} (\boldsymbol{w}_0 - \boldsymbol{w}^*) \right) - y_i \phi(\boldsymbol{x}_i) \right\|^2 \\
&= \sum_{i=1}^n \left\| y_i \phi(\boldsymbol{x}_i) + \phi(\boldsymbol{x}_i)\phi(\boldsymbol{x}_i)^\top e^{-\frac{1}{n} \phi(\mathbf{X})\phi(\mathbf{X})^\top t} (\boldsymbol{w}_0 - \boldsymbol{w}^*) - y_i \phi(\boldsymbol{x}_i) \right\|^2 \\
&= \sum_{i=1}^n \left\| \phi(\boldsymbol{x}_i)\phi(\boldsymbol{x}_i)^\top e^{-\frac{1}{n} \phi(\mathbf{X})\phi(\mathbf{X})^\top t} (\boldsymbol{w}_0 - \boldsymbol{w}^*) \right\|^2 \\
&\leq K_{\max} \sum_{i=1}^n (\boldsymbol{w}_0 - \boldsymbol{w}^*)^\top e^{-\frac{1}{n} \phi(\mathbf{X})\phi(\mathbf{X})^\top t} \phi(\boldsymbol{x}_i)\phi(\boldsymbol{x}_i)^\top e^{-\frac{1}{n} \phi(\mathbf{X})\phi(\mathbf{X})^\top t} (\boldsymbol{w}_0 - \boldsymbol{w}^*) \\
&= K_{\max} (\boldsymbol{w}_0 - \boldsymbol{w}^*)^\top e^{-\frac{1}{n} \phi(\mathbf{X})\phi(\mathbf{X})^\top t} \phi(\mathbf{X})\phi(\mathbf{X})^\top e^{-\frac{1}{n} \phi(\mathbf{X})\phi(\mathbf{X})^\top t} (\boldsymbol{w}_0 - \boldsymbol{w}^*) \\
&= K_{\max} (\boldsymbol{w}_0 - \boldsymbol{w}^*)^\top \sum_{i=1}^p \lambda_i e^{-\frac{2}{n} \lambda_i t} \boldsymbol{u}_i \boldsymbol{u}_i^\top (\boldsymbol{w}_0 - \boldsymbol{w}^*) \\
&= K_{\max} \sum_{i=1}^p \lambda_i e^{-\frac{2}{n} \lambda_i t} \left( \boldsymbol{u}_i^\top (\boldsymbol{w}_0 - \boldsymbol{w}^*) \right)^2 .
\end{aligned}
$$

Hence, we can obtain that

$$\int_0^T \sum_{i=1}^n \|\nabla_{\boldsymbol{w}} \ell(\boldsymbol{w}_t, \boldsymbol{z}_i)\|^2 \, dt \leq \int_0^T K_{\max} \sum_{i=1}^p \lambda_i e^{-\frac{2}{n} \lambda_i t} \left( \boldsymbol{u}_i^\top (\boldsymbol{w}_0 - \boldsymbol{w}^*) \right)^2 dt$$

$$= K_{\max} \sum_{i=1}^p \frac{n}{2} \left( 1 - e^{-\frac{2\lambda_i}{n} T} \right) \left( \boldsymbol{u}_i^\top (\boldsymbol{w}_0 - \boldsymbol{w}^*) \right)^2$$

$$\leq \frac{K_{\max} n}{2} \|\boldsymbol{w}_0 - \boldsymbol{w}^*\|^2 .$$

Therefore when $\lambda = 0$,

$$\Gamma \leq \frac{\sqrt{K_{\max}} \|\boldsymbol{w}_0 - \boldsymbol{w}^*\| \left\| \phi(\mathbf{X})^\top (\boldsymbol{w}_0 - \boldsymbol{w}^*) \right\|}{n}.$$

When $\boldsymbol{w}_0 = 0$, plunging in the expression of $\boldsymbol{w}^*$, the bound simplifies to

$$\Gamma \leq \frac{\sqrt{K_{\max}} \sqrt{\boldsymbol{y}^\top \left( K(\mathbf{X}, \mathbf{X}) \right)^{-1} \boldsymbol{y}} \, \|\boldsymbol{y}\|}{n}.$$

$\square$

## G.3 Feature Learning Case

We state the assumptions and results of Bietti et al. [15] below.

**Assumption G.1** (Regularity of $f_*$). We consider $f_* \in L^2(\gamma)$ with Hermite expansion $f_* = \sum_j \alpha_j h_j$, where $\gamma := \mathcal{N}(0, 1)$. Assume

1. $f_*$ is Lipschitz,

2. $\sum_j j^4 |\alpha_j|^2 < \infty$,

3. $f_*''(z) := \sum_j \sqrt{(j+2)(j+1)} \alpha_{j+2} h_j(z)$ is in $L^4(\gamma)$.

**Theorem G.2** (Theorem 6.1 in Bietti et al. [15]). *For $\delta \in (0, 1/4)$ and $f_*$ satisfying Assumption G.1, suppose the following are true: (i) $\lambda = O(1)$ and $\lambda = \Omega(\sqrt{\Delta_{crit}})$, where $\Delta_{crit} := \max\{\sqrt{\frac{d+N}{n}}, (\frac{d^2}{n})^{2s/(2s-1)}\}$, (ii) $n = \tilde{\Omega}(\max\{\frac{(d+N)d^{s-1}}{\lambda^4}, \frac{d^{(s+3)/2}}{\lambda^2}\})$, (iii) $N = \Omega(\frac{1}{\lambda \log \frac{1}{\lambda \delta}})$ and $N = \tilde{O}(\lambda \Delta_{crit}^{-1})$, (iv) $N_0 = \Theta(\log \frac{1}{\delta})$, (v) $\rho = \Theta(\sqrt{N} N_0^{-(2+s)/2}(\tau^2 + \lambda N/N_0)^{-1})$, (vi) $T_0 = \tilde{\Theta}(d^{s/2-1})$, and (vii) $T_1 = \tilde{\Theta}(\frac{\lambda^4 n}{d+N})$. Then, if we run Algorithm 1 for $T = T_0 + T_1$ time steps with the above parameters, with probability at least $\frac{1}{2} - \delta$ we have*

$$1 - |\langle \boldsymbol{\theta}, \boldsymbol{\theta}^* \rangle| = \tilde{O} \left( \lambda^{-4} \max \left\{ \frac{d+N}{n}, \frac{d^4}{n^2} \right\} \right).$$

---

**Algorithm 1** Gradient Flow

---

**Require:** $N_0, \rho, T_0, T_1, N$, and $\lambda$.
Initialize $\boldsymbol{\theta}(0) \sim \text{Unif}(\mathcal{S}^{d-1})$, $\boldsymbol{c}(0) \sim \text{Unif}(\{\boldsymbol{c} \in \mathbb{R}^N : \|\boldsymbol{c}\|_2 = \rho, \|\boldsymbol{c}\|_0 = N_0\})$.
Run gradient flow (6.3) up to time $T = T_0 + T_1$.
Return $\boldsymbol{\theta}(T), \boldsymbol{c}(T)$.

---

We recall the basic concentration properties of Gaussian random variables.

**Lemma G.3** (Concentrations of Gaussian random variables). *Let $\delta \in (0, 1/4)$, $N \in \mathbb{N}$, and $b_1, \ldots, b_N$ be i.i.d. random variables drawn from $\mathcal{N}(0, \tau^2)$. Then there exists a universal constant $C' > 0$ such that the following two events hold simultaneously with probability at least $1 - \delta$:*

$$\max_j |b_j| \leq C' \tau \sqrt{\log(N/\delta)},$$

$$\sum_j b_j^2 \leq N\tau^2 + C'\tau^2 \max \left\{ \log(1/\delta), \sqrt{N \log(1/\delta)} \right\}.$$

Recall $f(\boldsymbol{\theta}, \boldsymbol{c}; \boldsymbol{x}) = \frac{1}{\sqrt{N}} \sum_{i=1}^{N} c_i \phi(\sigma_i \langle \boldsymbol{\theta}, \boldsymbol{x} \rangle + b_i) = \boldsymbol{c}^\top \Phi(\langle \boldsymbol{\theta}, \boldsymbol{x} \rangle)$, where we denote the feature vector of first layer as $\Phi(\langle \boldsymbol{\theta}, \boldsymbol{x} \rangle) = (\frac{1}{\sqrt{N}} \phi(\sigma_i \langle \boldsymbol{\theta}, \boldsymbol{x} \rangle + b_i))_{i=1}^{N}$. We have the following bound for the feature vector.

**Lemma G.4** ($\ell_2$-norm of random features, Corollary D.5 in Bietti et al. [15]). *Let $\delta \in (0, 1/4)$ and $b_1, \ldots, b_N$ be i.i.d. random variables drawn from $\mathcal{N}(0, \tau^2)$. Then there exists a universal constant $C' > 0$ such that the following holds for all $z \in \mathbb{R}$ with probability at least $1 - \delta$ over the random features,*

$$\|\Phi(z)\| \leq |z| + C'\tau(1 + \sqrt{\log(1/\delta)/N}) \leq |z| + 2C'\tau\sqrt{\log(1/\delta)}.$$

We restate and prove our bound below.

**Corollary 6.3.** *Under the settings of Theorem 6.1 in Bietti et al. [15] (provided in Theorem G.2),*

$$\Gamma \leq \tilde{O}\left(\sqrt{\frac{d^{\frac{s}{2}+1}}{n\lambda^2} + \lambda^2 d}\right),$$

*with high probability as $n, d \to \infty$. As long as $\lambda = o_d(1/\sqrt{d})$ and $n = \tilde{\Omega}(d^{\frac{s}{2}+2})$, $\Gamma = o_{n,d}(1)$. Taking $\lambda = \Theta(\frac{d^{\frac{s}{2}}}{n})^{\frac{1}{4}}$, we have*

$$\Gamma \leq \tilde{O}\left(\left(\frac{d^{\frac{s}{2}+2}}{n}\right)^{\frac{1}{4}}\right).$$

*Proof.* Recall

$$\Gamma = \frac{2}{n}\sqrt{L_\mathcal{S}(\boldsymbol{\theta}_0, \boldsymbol{c}_0) - L_\mathcal{S}(\boldsymbol{\theta}_T, \boldsymbol{c}_T)}\sqrt{\sum_{i=1}^{n} \int_0^T \|\nabla_{\boldsymbol{w}}\ell(\boldsymbol{w}_t, \boldsymbol{z}_i)\|^2 \, dt}.$$

We first calculate the order of the $\sqrt{L_\mathcal{S}(\boldsymbol{\theta}_0, \boldsymbol{c}_0) - L_\mathcal{S}(\boldsymbol{\theta}_T, \boldsymbol{c}_T)}$. Recall

$$L_\mathcal{S}(\boldsymbol{\theta}, \boldsymbol{c}) = \frac{1}{n}\sum_{i=1}^{n}(f(\boldsymbol{\theta}, \boldsymbol{c}; \boldsymbol{x}_i) - y_i)^2 + \lambda\|\boldsymbol{c}\|^2.$$

Then one can claim that

$$L_\mathcal{S}(\boldsymbol{\theta}_0, \boldsymbol{c}_0) = \frac{1}{n}\sum_{i=1}^{n}(f(\boldsymbol{\theta}_0, \boldsymbol{c}_0; \boldsymbol{x}_i) - y_i)^2 + \lambda\|\boldsymbol{c}_0\|^2$$

$$= \frac{1}{n}\sum_{i=1}^{n}\left(f(\boldsymbol{\theta}_0, \boldsymbol{c}_0; \boldsymbol{x}_i)^2 + y_i^2 - 2y_i f(\boldsymbol{\theta}_0, \boldsymbol{c}_0; \boldsymbol{x}_i)\right) + \lambda\|\boldsymbol{c}_0\|^2.$$

By Lemma G.3 $\phi(\sigma_i \langle \boldsymbol{\theta}_0, \boldsymbol{x}_i \rangle + b_i) = \phi(\tilde{O}(1) + \tilde{O}(1)) = \tilde{O}(1)$. Since $\mathbb{E}_{\boldsymbol{c}_0}\left[f(\boldsymbol{\theta}_0, \boldsymbol{c}_0; \boldsymbol{x}_i)^2\right] = \mathbb{E}_{\boldsymbol{c}_0}\left[\frac{1}{N}\sum_{i=1}^{N} c_i^2\phi(\sigma_i \langle \boldsymbol{\theta}_0, \boldsymbol{x}_i \rangle + b_i)^2\right] = \tilde{O}(\frac{\|\boldsymbol{c}_0\|^2}{N}) = \tilde{O}(\frac{\rho^2}{N}) = \tilde{O}(1)$, by Chebyshev's inequality $f(\boldsymbol{\theta}_0, \boldsymbol{c}_0; \boldsymbol{x}_i) = \tilde{O}(1)$. Since also $y_i = \tilde{O}(1)$ and $\lambda\|\boldsymbol{c}_0\|^2 = \tilde{O}(1)$, one can verify $L_\mathcal{S}(\boldsymbol{\theta}_0, \boldsymbol{c}_0) = \tilde{O}(1)$. Therefore $L_\mathcal{S}(\boldsymbol{\theta}_0, \boldsymbol{c}_0) - L_\mathcal{S}(\boldsymbol{\theta}_T, \boldsymbol{c}_T) = \tilde{O}(1)$. Since $L_\mathcal{S}(\boldsymbol{\theta}_T, \boldsymbol{c}_T)$ is non-increasing in gradient flow, $\lambda\|\boldsymbol{c}_t\|^2 = \tilde{O}(1)$ during training.

Then we calculate the $\sum_{i=1}^{n} \int_0^T \|\nabla_{\boldsymbol{w}}\ell(\boldsymbol{w}_t, \boldsymbol{z}_i)\|^2 \, dt$. By Lemma G.4, the sample gradient for $\boldsymbol{\theta}$ and an upper bound for its $\ell_2$ norm is given by

$$\nabla_{\boldsymbol{\theta}}\ell(\boldsymbol{\theta}, \boldsymbol{c}; \boldsymbol{x}_i, y_i) = -\boldsymbol{c}^\top\Phi'(\langle \boldsymbol{\theta}, \boldsymbol{x}_i \rangle)(y_i - \boldsymbol{c}^\top\Phi(\langle \boldsymbol{\theta}, \boldsymbol{x}_i \rangle))\boldsymbol{x}_i$$

$$\|\nabla_{\boldsymbol{\theta}}\ell(\boldsymbol{\theta}, \boldsymbol{c}; \boldsymbol{x}_i, y_i)\| = \|\boldsymbol{c}\|\,(\mathrm{Lip}(f)_*\|\boldsymbol{x}_i\| + \|\xi_i\| + \|\boldsymbol{c}\|\,(\|\boldsymbol{x}_i\| + C'\tau\sqrt{\log(1/\delta)}))\,\|\boldsymbol{x}_i\|$$

$$= \tilde{O}(\|\boldsymbol{c}\|^2\|\boldsymbol{x}_i\|^2).$$

Similarly, the sample gradient for $\boldsymbol{c}$ is

$$\nabla_{\boldsymbol{c}}\ell(\boldsymbol{\theta}, \boldsymbol{c}; \boldsymbol{x}_i, y_i) = 2\Phi(\langle \boldsymbol{\theta}, \boldsymbol{x}_i \rangle)(\boldsymbol{c}^\top\Phi(\langle \boldsymbol{\theta}, \boldsymbol{x}_i \rangle)) - f_*(\langle \boldsymbol{\theta}, \boldsymbol{x}_i \rangle - \xi_i)$$

$$\|\nabla_{\boldsymbol{c}}\ell(\boldsymbol{\theta}, \boldsymbol{c}; \boldsymbol{x}_i, y_i)\| = 2(\|\boldsymbol{x}_i\| + C'\tau\sqrt{\log(1/\delta)})(\|\boldsymbol{c}\|\,(\|\boldsymbol{x}_i\| + C'\tau\sqrt{\log(1/\delta)})$$

$$+ \mathrm{Lip}(f)_*\|\boldsymbol{x}_i\| + \|\xi_i\|) = \tilde{O}(\|\boldsymbol{c}\|\,\|\boldsymbol{x}_i\|^2).$$

As we have shown, $\|c\|^2 = O(\frac{1}{\lambda})$ and by Lemma G.3, $\max_i \|x_i\| = O(\sqrt{d \log(n/\delta)})$. Therefore,

$$\|\nabla_{\boldsymbol{w}} \ell(\boldsymbol{w}_t, \boldsymbol{z}_i)\|^2 = \|\nabla_{\boldsymbol{\theta}} \ell(\boldsymbol{\theta}, \boldsymbol{c}; \boldsymbol{x}_i, y_i)\|^2 + \|\nabla_{\boldsymbol{c}} \ell(\boldsymbol{\theta}, \boldsymbol{c}; \boldsymbol{x}_i, y_i)\|^2 = \tilde{O}\Big(\frac{d^2}{\lambda^2}\Big).$$

Then take $T = T_0 + T_1$, we have

$$\sum_{i=1}^{n} \int_0^T \|\nabla_{\boldsymbol{w}} \ell(\boldsymbol{w}_t, \boldsymbol{z}_i)\|^2 \, dt \leq n \cdot \tilde{O}\Big(\frac{d^2}{\lambda^2}\Big) \cdot (T_0 + T_1)$$

$$\leq n \cdot \tilde{O}(\frac{d^2}{\lambda^2}) \cdot \left( \tilde{\Theta}(d^{\frac{s}{2}-1}) + \tilde{\Theta}(\frac{\lambda^4 n}{d + N}) \right)$$

$$= n \cdot \tilde{O}\left( \frac{d^{\frac{s}{2}+1}}{\lambda^2} + \lambda^2 dn \right).$$

Combining the results, we have

$$\Gamma \leq \frac{2}{n} \sqrt{\tilde{O}(1) \cdot n \cdot \tilde{O}\left( \frac{d^{\frac{s}{2}+1}}{\lambda^2} + \lambda^2 dn \right)}$$

$$= \tilde{O}\left( \sqrt{\frac{d^{\frac{s}{2}+1}}{n\lambda^2} + \lambda^2 d} \right).$$

As long as $\lambda = o_d(1/d)$ and $n = \tilde{\Omega}(d^{\frac{s}{2}+2})$, $\Gamma = o_{n,d}(1)$. Optimizing the choice of $\lambda = (\frac{d^{\frac{s}{2}}}{n})^{\frac{1}{4}}$, we have

$$\Gamma \leq \tilde{O}\left( \left( \frac{d^{\frac{s}{2}+2}}{n} \right)^{\frac{1}{4}} \right).$$

Hence, we complete the proof. □

