# OpenReview forum: "Generalization Bound of Gradient Flow through Training Trajectory and Data-dependent Kernel"
_NeurIPS.cc/2025/Conference — NeurIPS 2025 poster_

### Official Review · Reviewer_m74d · 2025-06-24

**Clarity:** 3
**Significance:** 3
**Originality:** 2
**Rating:** 5
**Confidence:** 3

**Summary:**

This paper moves beyond the static NTK framework by using the Loss Path Kernel to capture training dynamics. It derives a novel generalization bound using Rademacher complexity, whose leading term is determined by the term $\int_0^T \| \nabla_w l(w_t,z) \|^2 dt$. This approach yields a tighter, data-dependent bound about $O(\frac{1}{n})$ that directly links optimization dynamics to generalization, surpassing existing results.

**Questions:**

1、Do we have a way to bound the term $\int_0^T \| \nabla_w l(w_t,z) \|^2 dt$

2、Do Gradient Descent and Gradient Flow have a difference in the order of the resulting bound?

**Ethical Concerns:**

["NO or VERY MINOR ethics concerns only"]

**Final Justification:**

I have no more questions and I will keep my original positive score.

**Limitations:**

The proposed bound $\Gamma$ cannot be determined theoretically a priori. While the authors claim it can be used for monitoring and prediction in practice, they do not adequately discuss its high computational cost. This makes their claim impractical for large-scale applications.

**Paper Formatting Concerns:**

No formatting concerns.

**Quality:**

3

**Strengths And Weaknesses:**

Strengths: This paper has enough technical contribution and beautiful results. It derives a tighter bound on the generalization error, and the theory is in line with many known phenomena, like why memorizing random labels hurts generalization. The authors also verified the bound on CIFAR-10 with ResNet, and the experiments are consistent with the theoretical results, which is a strong point.

Weaknesses: My main concern is about the exact order of the upper bound. The main term, which is based on the integral of gradients $\int_0^T \| \nabla_w l(w_t,z) \|^2 dt$, can only be calculated after training is finished. This means the bound is a post-hoc analysis tool, and cannot be used to predict generalization or guide the training process beforehand. Also, computing this term is very expensive as it requires the full gradient history, making it hard to use in practice.

---

> ### Author Rebuttal · Authors · 2025-07-31
>
> We thank the reviewer for the positive feedback, thoughtful comments, and for appreciating the novelty and value of the work! In response to the reviewer’s inquiries, we present the following detailed explanation.
>
> ---
> **Q1. The exact order of the upper bound.**
> A1. A naive bound of $\Gamma$ is $L\sqrt{T/n}$ using the Lipschitz property based on our assumption. Combining with the $\epsilon$ term, our bound is at most $O(\sqrt{T/n})$ _in the worst case_. However, when the model converges fast, $\Gamma$ may not increase with $\sqrt{T}$, and the $\epsilon$ can also achieve a faster rate of $O(n^{-3/4})$ when $T$ is small. Our Table 1 shows when $\Gamma$ will dominate our generalization bound. For some special cases, e.g., NTK regime, we refer to Section 6 for more details.
>
>
> **Q2. The bound is a post-hoc analysis tool, and cannot be used to predict generalization or guide the training process beforehand.**
>
> A2. In the case studies in Section 6, we can predict the generalization before the training. See Corollary 6.1, 6.2, 6.3. In these case studies, we have made additional assumptions regarding the dataset, optimizer, and training dynamics. Consequently, we can further simplify our generalization error bound and obtain a more precise analysis of the leading $\Gamma$ terms. If we do not have detailed assumptions about the learning dynamics, we may need to carry out the training process sufficiently to extract enough information to determine our generalization error bound. Potentially, a practical application of our framework is to train a model with a smaller size on a smaller dataset and then use its observed training trajectory to compute the LPK‐based generalization bound. This “mini‐experiment” provides an early, data‐driven estimate of how well the full‐scale model is likely to generalize without the computational cost of training a very large network.
>
> By monitoring the evolution of our bound $\Gamma$ during training, we can predict the overfitting for overparameterized NNs and identify optimal stopping time for training without access to test data. Our bound can also be used as a regularization in the training. Moreover, our bound can also serve as a proxy to compare model architectures in Neural Architecture Search (NAS), e.g., [1, 2, 3, 4].
>
> [1] Mellor, J., Turner, J., Storkey, A., and Crowley, E. J. Neural architecture search without training. ICML 2021.
>
> [2] Chen, W., Gong, X., and Wang, Z. Neural architecture search on imagenet in four gpu hours: A theoretically inspired perspective. ICLR 2021.
>
> [3] Mok, J., Na, B., Kim, J.-H., Han, D., and Yoon, S. Demystifying the neural tangent kernel from a practical perspective: Can it be trusted for neural architecture search without training? CVPR 2022.
>
> [4] Chen, Y., Huang, W., Wang, H., Loh, C., Srivastava, A., Nguyen, L. M., and Weng, T.-W. Analyzing generalization of neural networks through loss path kernels. NeurIPS 2023.
>
>
> **Q3. Computing this term is very expensive as it requires the full gradient history, making it hard to use in practice.**
>
> A3. The computation of our bound is tractable. In the $\Gamma$ term (Eq. (3)), as Reviewer xevc,
>  mentioned $L_S(w_0) - L_S(w_T)$ can be easily computed from the training loss. For $\sum_{i=1}^n \int_0^T ||\nabla_w \ell(w_t, z) ||^2 dt$, we can compute the per-sample gradients (which can be computed in parallel for all the samples) at each training step, calculate their 2-norm, and sum over the training steps. The computation cost of the bound is similar to that of gradient descent training of the model. For large datasets, we can estimate $\sum_{i=1}^n \int_0^T ||\nabla_w \ell(w_t, z) ||^2 dt$ with a subset of training samples, such as the batch data used at each training step in SGD. We will add a detailed discussion of the computational complexity of evaluating the LPK term $\Gamma$.
>
>
> **Q4. Do we have a way to bound the term $\int_0^T ||\nabla_w \ell(w_t, z)|| dt$?**
>
> A4. A naive bound of $\int_0^T ||\nabla_w \ell(w_t, z)|| dt$ is $L^2 T$ using the Lipschitz property. It is hard to give a tighter bound for general cases. But for more specific models, such as the case studies in Sec 6, we can derive tighter bounds for this quantity.
>
> For the NTK case, in line 730, we bound it by $\sum_i^n \int_0^T ||\nabla_w \ell(w_t, z_i)|| dt \leq \frac{n \lambda_{max} \|\|f(w_0, X) - y\|\|^2 }{2 \lambda_{min}}  ( 1 - e^{-\frac{2 \lambda_{min} T}{n}})$.
>
> For the kernel regression, in line 744, we bound it by $\sum_i^n \int_0^T ||\nabla_w \ell(w_t, z_i)|| dt \leq \frac{K_{max} n}{2}||w_0 - w^*||^2$.
>
> **Q5. Do Gradient Descent and Gradient Flow have a difference in the order of the resulting bound?**
>
> A5. In the early phase of this paper, we computed the gradient flow and found that the difference between the gradient flow and gradient descent with a small learning rate is very small, as also shown in [5, 6]. Therefore, the bounds of gradient flow and GD should not differ much under certain conditions of the learning rate. We will present a simulation to control the difference between the GD and gradient flow dynamics for two-layer neural networks on a portion of the MNIST dataset.
>
> Our experimental findings demonstrate a strong correlation between our bound and gradient descent, particularly when employing reasonable learning rates. In our simulation, two-layer neural networks (NNs) and ResNets were trained with learning rates of 0.01 and 0.001, respectively, which are widely adopted learning rate settings in practical applications. These experimental results also suggest the potential extension of our theory to discrete SGD with a constant learning rate, which we intend to explore further in future research endeavors. Furthermore, it is noteworthy that [5] indicates that deep neural networks with homogeneous activations can be approximated by gradient descent. Consequently, we can potentially transfer our analysis of gradient flow for neural networks to a gradient descent training framework based on this observation. We will elaborate on this limitation in the revised version of our paper.
>
>
> [5] Continuous vs. Discrete Optimization of Deep Neural Networks. NeurIPS 2021.
>
> [6] Chen, Y., Huang, W., Wang, H., Loh, C., Srivastava, A., Nguyen, L. M., and Weng, T.-W. Analyzing generalization of neural
>  networks through loss path kernels. NeurIPS 2023.
>
> ---
> We are happy to address any further questions and feedback.

---

> > ### Comment · Reviewer_m74d · 2025-08-09
> >
> > Thank you for your detailed response. I will keep my positive score.

---

### Official Review · Reviewer_xevc · 2025-06-24

**Clarity:** 3
**Significance:** 3
**Originality:** 3
**Rating:** 4
**Confidence:** 3

**Summary:**

This paper establishes new generalization error bounds for the gradient flow algorithm. The method is an extension of the previous work of Chen et al. on loss path kernel. The authors integrate uniform stability and uniform convergence analysis with loss path kernel, leading to tighter generalization bounds. The results demonstrate that the generalization gap is highly dependent on the norm of the training loss gradients along the optimization trajectory. Further extension to stochastic gradient flow and several applications on the NTK regime and kernel ridge regression are discussed, and the results are empirically evaluated to show high correlation with the true generalization gap.

**Questions:**

* Can the authors elaborate more on the impact of the big O terms on the tightness and computational tractability of these bounds? How are these bounds empirically compared to the previous one by Chen et al?
* Can these bounds provide some new insights for the design of learning algorithms? From eq. (3), what I can tell is that minimizing the norm of gradients may help, encouraging techniques like gradient clipping. The other part in eq. (3) looks like a fixed constant, as we always get something like $L_S(w_0) \approx 0.5$ and $L_S(w_T) \approx 0$ in case of binary classification. Are there any other insights we can tell from these results?

**Ethical Concerns:**

["NO or VERY MINOR ethics concerns only"]

**Final Justification:**

My main concerns are fully addressed after the rebuttal. I believe this paper offers a solid improvement over the previous work of Chen et al. I'm maintaining my current rating.

**Limitations:**

yes

**Paper Formatting Concerns:**

No concerns.

**Quality:**

4

**Strengths And Weaknesses:**

Strength:
* The paper addresses an important problem in learning theory. The authors improve the results of a previous work by Chen et al, tightening the bound and easing the computation in practice. Empirical results have shown the superiority of these results compared to some other bounds in the literature.

Weakness:
* To my understanding, using the method of uniform stability or uniform convergence requires additional assumptions like Lipschitzness and smoothness, while the previous one of Chen et al does not need such assumptions. While they are prevalent ones also seen in many generalization analysis papers, they are actually quite strong and make the bound computationally intractable, since the Lipschitz or smoothness parameters are hard to estimate in practice.
* It is not directly clear how to compare the tightness between the bounds of this paper and the previous one by Chen et al. While the $\Gamma$ part of the bound is improved as clearly seen, these new bounds also introduce some extra terms of stability constants. There perhaps also miss some comparison between these bounds and those ones derived solely based on stability or uniform convergence approaches. The authors claimed a tighter bound, but this is not directly clear to me.
* For the same reason, the empirical comparison may not be sufficient since only the $\Gamma$ part of the bound is computed and plotted. It is not clear if the bound is still tight when added with other stability or uniform convergence terms.

---

> ### Author Rebuttal · Authors · 2025-07-31
>
> We express our gratitude for the positive feedback, thoughtful review, and the reviewer’s appreciation for our work. In response to the reviewer’s suggestions and inquiries, we will incorporate a comparative analysis with the results of Chen et al. and provide a concise summary of the improvements made to our paper.
>
> ---
> **Q1. The Liptichitzness and smoothness assumptions are quite strong and make the bound computationally intractable, since the Lipschitz or smoothness parameters are hard to estimate in practice.**
>
> A1. Thanks for the question. However, we would like to note that the $\Gamma$ term in our bound does not depend on the Lipschitz or smoothness constant. As we have compared in Table 1, we conclude that in many cases, $\Gamma$ is the dominant term compared with $\epsilon$. In these cases, $\epsilon$ will be small compared with $\Gamma$ as $n\to\infty$. Hence, we can use the $\Gamma$ to represent our generalization bound. For large-scale simulations, our $\Gamma$ can be applied to bound the generalization gap, and it empirically tracks the actual generalization gap well.
>
> To improve the results of Chen et al. and obtain a better LPK-type generalization bound, we need to include the Lipschitz and smoothness assumptions. Chen et al. do not require these assumptions because they did not perform any optimization analysis. Their bound calculates the supremum over an infinite set, which cannot be accurately estimated. Additionally, their bounds must be evaluated on (infinitely many) datasets different from the training set, which is impossible in practice. By explicitly enforcing Lipschitz and smoothness conditions, one transforms an intractable supremum‐based LPK guarantee into a finite-sample, data-dependent, and computationally verifiable bound. This modification not only tightens the theoretical rates but also aligns the analysis with the practical realities of deep‐network training. We will add a detailed comparison with Chen et al. in the revision of our paper.
>
> **Q2. Compare the tightness between the bounds of this paper and the previous one by Chen et al. Miss some comparison between these bounds and those ones derived solely based on stability or uniform convergence approaches.**
>
> A2. Our bound is clearly tighter than the previous one by Chen et al. Our bound has a rate of $O(\sqrt{T/n})$ in the worst case, while the bound in Chen et al can be as large as $O(\sqrt{T})$ since the $\sum_i \delta(z_i, z_j)$ term in their bound can be $O(L^2 T n^2)$.
>
> Stability-based bounds are exponential with $T$ for non-convex loss with constant learning rate, as we showed in Lemma 4.3, while our bound is at most $O(\sqrt{T/n})$ in the worst case. Uniform convergence bounds such as [1, 2, 3, 4] listed below are usually based on the norm of the weight matrices. These bounds are independent of the algorithm and cannot capture the inductive bias of the learning algorithm. In contrast, our bound is data-dependent and algorithm-dependent. Empirically, our bound is also orders of magnitude tighter than [3], as shown in Fig. 2.
>
> [1] Bartlett, P. L. and Mendelson, S. Rademacher and gaussian complexities: Risk bounds and structural results. JMLR 2002.
>
> [2] Neyshabur, B., Tomioka, R., and Srebro, N. Norm-based capacity control in neural networks. COLT 2015.
>
> [3] Bartlett, P. L., Foster, D. J., and Telgarsky, M. J. Spectrally-normalized margin bounds for neural networks. NeurIPS 2017.
>
> [4] Neyshabur, B., Li, Z., Bhojanapalli, S., LeCun, Y., and Srebro, N. The role of over-parametrization in generalization of neural networks. ICLR 2019.
>
> **Q3. Impact of the big O terms on the tightness and computational tractability of these bounds? How are these bounds empirically compared to the previous one by Chen et al?**
>
> A3. As we compared in Table 1, in many cases, $\Gamma$ is the dominant term compared with $\epsilon$. In these cases, the big-O terms are small as $n$ increases. Hence, we can use $\Gamma$ to track the trend of the generalization error bound through different training dynamics.
>
> Our bound is tighter than the previous one by Chen et al. As we mentioned above in A2, our bound has a rate of $O(\sqrt{T/n})$ in _the worst case_, while the bound in Chen et al can be as large as $O(\sqrt{T})$. Numerically, we will add an empirical estimate of the big-O terms for the MNIST dataset and small neural networks, and compare our bound with Chen et al in the revised version.
>
> **Q4. Can these bounds provide some new insights for the design of learning algorithms?**
>
> A4. Thank you for the insightful question. The primary motivation of this paper is to investigate the generalization capabilities of neural networks. It would be more insightful if our theory could provide practical insights for training neural networks. We concur with the reviewer’s observation that our theory indicates gradient clipping. Moreover, by monitoring the evolution of our bound during training, we can potentially predict and analyze the overfitting for overparameterized NNs and identify optimal stopping time for training without access to test data. We could potentially utilize a smaller model and a smaller dataset to train and approximate our generalization bound. Subsequently, we can employ this bound to predict the generalization capability of larger models. Our bound can also be used as a regularization in the training. Moreover, our bound can also serve as a proxy to compare model architectures in Neural Architecture Search (NAS), e.g., [5, 6, 7, 8].
>
> [5] Mellor, J., Turner, J., Storkey, A., and Crowley, E. J. Neural architecture search without training. ICML 2021.
>
> [6] Chen, W., Gong, X., and Wang, Z. Neural architecture search on imagenet in four gpu hours: A theoretically inspired perspective. ICLR 2021.
>
> [7] Mok, J., Na, B., Kim, J.-H., Han, D., and Yoon, S. Demystifying the neural tangent kernel from a practical perspective: Can it be trusted for neural architecture search without training? CVPR 2022.
>
> [8] Chen, Y., Huang, W., Wang, H., Loh, C., Srivastava, A., Nguyen, L. M., and Weng, T.-W. Analyzing generalization of neural networks through loss path kernels. NeurIPS 2023.
>
> ---
>
> We are happy to address any further questions and feedback.

---

> > ### Comment · Reviewer_xevc · 2025-08-05
> >
> > Thank you for the detailed response. My concerns are fully addressed after the rebuttal, and I'm willing to keep my current score.

---

### Official Review · Reviewer_cwrL · 2025-06-26

**Clarity:** 4
**Significance:** 3
**Originality:** 3
**Rating:** 4
**Confidence:** 3

**Summary:**

The paper presents a novel generalization bound for parametric models trained with gradient flow. It builds on a recent result showing that the loss at the end of training of any parametric model is a kernel machine for a certain data-dependent kernel called Loss Path Kernel (LPK). Therefore the class of models realizable by gradient flow could be enumerated by corresponding LPKs for all possible training datasets.

If there was a single kernel, one could use a classical bound for Rademacher complexity which scales as a square root of a kernel Gram matrix trace over the dataset size $n$. One could generalize this bound for a class of kernel machines realizable by a set of kernels, but such a bound does not generally decrease with $n$.

The main novelty of the present paper is a concentration result on LPKs: that is, as $n$ grows to infinity, all LPKs concentrate around their average over training datasets. This leads to a generalization gap bound whose main term is the classical Rademacher complexity bound mentioned above evaluated on the LPK on the training dataset.

The main term of the bound closely tracks the actual generalization gap times a constant. This constant is relatively small $(\approx 3)$ for a binary CIFAR classification using a two-layer MLP, but much larger $(\approx 1e5)$ when ResNet18 is considered.

**Questions:**

1. I haven't looked at the proof, but do you have any intuitive explanation why the epsilon term in your Theorem 5.2 is a minimum between two terms, not a maximum? Because naively when you construct an upper-bound, you have to take the largest between competing terms, not the smallest.
2. Have you tried to compute your bound up to a number? How close is it to gamma on your experiments? Does it also track the actual generalization gap well?

**Ethical Concerns:**

["NO or VERY MINOR ethics concerns only"]

**Final Justification:**

The authors responded quite satisfactory to some of my concerns; for this reason, I raising my score from 3 to 4.

**Limitations:**

I think the point I discuss above in Weaknesses has to be mentioned as a limitation.
I will increase my score if the authors either (a) prove me wrong concerning this point, or (b) correct/remove the Numerical Experiments section and include the aforementioned point as a limitation.

**Quality:**

3

**Strengths And Weaknesses:**

**Strengths**

1. I believe, the concentration of LPKs is a natural, novel and generally useful result, closely related to algorithmic stability.
2. The paper features a very nicely written background section which makes it easy to read and comprehend.
3. The main term of the bound tracks the time evolution of the generalization gap really well, even in realistic scenarios like ResNet18 trained on CIFAR10, up to a multiplicative constant. Maybe, it could be useful for predicting the optimal early stopping time?

**Weaknesses**

1. The regularity assumption on the loss as a function of model weights (Assumption 4.2) looks too strong for me. While it is quite natural for some models, like Random Feature Model, it is too restrictive for neural networks with at least two trainable layers. Indeed, consider a two-layer MLP with a sigmoid activation function, no bias terms, and a bounded (say, also sigmoid) loss. Suppose the input layer weights are zeros, while the output layer weights are very large. Then even a very small change of input weights results in a finite change of the loss. This shows that the Lipschitz property is not satisfied even for this toy model. Having a non-bounded activation function, non-bounded loss, and more layers cannot make the situation better.
2. If the above is correct, the bound cannot be applied to any of the scenarios considered in Section 7, Numerical Experiments.

**Minor comments**

1. The work [1] constructs a generalization bound for neural nets close to being linear; this is also a special case of algorithmic stability which heavily restricts the hypothesis class (i.e. only weights close to a linear network could be found), hence could be mentioned in the Related Work section.

[1] https://openreview.net/pdf?id=tRpWaK3pWh

---

> ### Author Rebuttal · Authors · 2025-07-31
>
> We appreciate the positive feedback, thoughtful review, and the reviewer’s appreciation for the work! We will add a limitation section, provide further discussion in the numerical experiments section, and address the points the reviewer mentioned in our revised paper. We also thank the reviewer for mentioning the paper, “A Generalization Bound for Nearly-Linear Networks”. We will have a discussion of this paper in our literature review.
>
> ---
> **Q1. The Lipschitzness and smoothness assumptions.**
>
> A1. Thanks for the question and comments. The Lipschitzness and smoothness assumptions are standard assumptions in optimization and generalization error bound analysis of neural networks [1, 2, 4]. There is also considerable empirical evidence that neural networks are Lipschitz, and many efforts have been made to estimate the Lipschitz constant (see e.g. [3, 4]).  In practice, training with weight decay or early stopping confines the weights to a compact subset of parameter space, on which the loss is indeed (locally) Lipschitz.
>
> For a two-layer network with smooth activation functions, as long as the norms of the weight matrices are bounded, there should be a finite Lipschitz constant. For the toy model the reviewer is concerned with, the Lipschitz constant could be large. But the Lipschitz constant will still be finite as long as the weight matrices are bounded and the activation function is Lipschitz. This is why in many previous generalization bound papers, the final bound is determined by the Frobenius norm of the weight matrices, e.g., [4, 5, 6, 7, 8]. However, this Lipschitz constant can be significantly large depending on the model’s size, although the trend is consistent with the true generalization gap. This is because, as the reviewer mentioned, the bound in the ResNet18 simulation is much larger.
>
> We will add additional discussion about the limitations of Lipschitzness and smoothness assumptions in the conclusion section. In the simulations, we will also add an experiment to estimate the Lipschitz constant and weight matrix norm during training. This will enable us to ascertain that these terms do not exhibit explosive growth and are bounded by a constant. In the event that this is the case, it will provide empirical evidence of our simulation’s consistency with our theoretical framework. Furthermore, our simulations show that our $\Gamma$ in the theorems has already served as a good bound of the generalization gap across diverse architectures and datasets.  Even when a network may not strictly satisfy the global Lipschitz or smoothness conditions, we find that our bound still tracks its behavior remarkably well. These results suggest that, with modest technical extensions, our theory could be broadened to cover a wider class of models. This is one of the reasons why various simulations on real-world data and various neural networks have been presented in this paper. We will conclude by outlining the challenges and potential strategies for relaxing these regularity assumptions in future work in the revision of our paper.
>
>
> [1] Hardt, M., Recht, B., and Singer, Y. Train faster, generalize better: Stability of stochastic gradient descent. ICML 2016.
> [2] Optimization Methods for Large-Scale Machine Learning.
> [3] Efficient and Accurate Estimation of Lipschitz Constants for Deep Neural Networks. NeurIPS 2019.
> [4]  Generalization bounds of stochastic gradient descent for wide and deep neural networks. NeurIPS 2019.
> [5] Bartlett, P. L. and Mendelson, S. Rademacher and gaussian complexities: Risk bounds and structural results. JMLR 2002.
> [6] Neyshabur, B., Tomioka, R., and Srebro, N. Norm-based capacity control in neural networks. COLT 2015.
> [7] Bartlett, P. L., Foster, D. J., and Telgarsky, M. J. Spectrally-normalized margin bounds for neural networks. NeurIPS 2017.
> [8] Neyshabur, B., Li, Z., Bhojanapalli, S., LeCun, Y., and Srebro, N. The role of over-parametrization in generalization of neural networks. ICLR 2019.
>
> **Q2. An intuitive explanation why the epsilon term in your Theorem 5.2 is a minimum between two terms, not a maximum.**
>
> A2. The epsilon term is the minimum between the two terms because the two bounds both hold at the same time, and we can take the minimum one. The mechanism is different from what the reviewer mentioned. Here we have two different methods for the epsilon term. So we take the minimum to get a sharper bound.
>
>
> **Q3. Have you tried to compute your bound up to a number? How close is it to gamma on your experiments? Does it also track the actual generalization gap well?**
>
> A3. Thanks for your suggestion! We have not tried to do that since estimating the Lipschitz/smoothness constant, and the constant in the big-O terms is hard. In the revision, we will numerically estimate our bound, including the $\epsilon$ term for a two-layer neural network during training to see how it tracks or affects the generalization gap.
>
> However, as we have compared in Table 1, we want to emphasize that in many cases, $\Gamma$ is the dominant term compared with $\epsilon$. In these cases, $\epsilon$ will be small compared with $\Gamma$ as $n\to\infty$. Hence, we can use the $\Gamma$ to represent our generalization bound. For large-scale simulations, our $\Gamma$ can be applied to bound the generalization gap, and it empirically tracks the actual generalization gap well. Also, for future work, we hope to develop a new method to estimate our LPK term so that it could be useful for predicting the optimal early stopping time.
>
>
> **Q4. Missing related work.**
>
> A3. Thanks for the related work! We will add and discuss the paper in the revised version.
>
> ---
>
> We are happy to address any further questions and feedback.

---

> > ### Comment · Reviewer_cwrL · 2025-08-04
> >
> > I am grateful to the authors for their detailed and elaborate response!
> >
> > **Q1**
> >
> > You cite [3,4] as works where a Lipschitz constant is estimated. However, [3] estimates the Lipschitz constant of a neural network as a function of its input, not its weights, while your Assumption 4.2 asks the loss function to be Lipschitz in weights. At the same, after skimming through [4], I do not see where they estimate the Lipschitz constant.
> >
> > If the weights are guaranteed to lie within a ball during training then the loss function is indeed Lipschitz in weights. However, for you results to be legit, the Lipschitz constant $L$ has to be uniform over all possible training datasets. That is, there should exist a ball in a parameter space such that no matter which training dataset you sample, the trained weights lie within that ball. It is not a-prioiri clear why this should hold. Even if you add early stopping or weight decay, does it (sharply) constrain the parameter within a ball whose radius does not depend on the dataset? This is not clear to me.
> >
> > **Q2**
> >
> > Thanks for clarification!

---

> ### Author Response · Authors · 2025-08-06
> **Official Comment by authors**
>
> Thanks for your follow-up question! We really appreciate you taking the time to review our paper.
>
> We apologize for the ambiguous references in the rebuttal. You are correct that [3] indeed estimates the Lipschitz constant of the input. [4] does not estimate the Lipschitz constant but rather derives a generalization bound using the Lipschitz condition of the loss function and activation function. On the other hand, we want to mention [9], which estimates the upper bound of the Lipschitz constant of NN parameters. They show the Lipschitz constant is upper-bounded as long as the parameters and input are bounded.
>
> In general, it could be hard to guarantee the global Lipschitzness and boundedness of parameters in general settings. But for some specific setting, such as the NTK case, the parameters can be bounded in a ball [10, 11, 12]. Additionally, we can show the boundedness of the parameter space under the gradient flow with an $\ell_2$ weight decay. Suppose that we consider loss function $J(w)=L_S(w)+\frac{\lambda}{2}\|\|w\|\|^2$, and gradient flow $
> \dot w(t)=-\nabla J(w(t)),\quad w(0)=w_0.$ We can check that  $J(w(t))\le J(w_0)$ for all $t\ge0$ because of $\frac{d}{dt}J(w(t))=\langle\nabla J(w),\dot w\rangle=-\|\|\nabla J(w(t))\|\|^2\le0$. Since $L_S(w)\ge0$, $\frac{\lambda}{2}\|\|w(t)\|\|^2\le J(w(t))\le J(w_0)=L_S(w_0)+\frac{\lambda}{2}\|\|w_0\|\|^2.$
> Hence, we can show that
> $\||w(t)\||\le\sqrt{\||w_0\||^2+\frac{2}{\lambda}L_S(w_0)}\equiv R$
> which depends only on $w_0$, $\lambda$, and $L_S(w_0)$ (the initial loss). $R$ is bounded whenever the loss is bounded.
>
> In deep learning, there are many other ways to potentially guarantee the global Lipschitzness and boundedness of training parameters, such as gradient clipping, projected GD [13], spectral normalization [14], and explicitly parametrizing weight matrices as orthogonal matrices [15]. We leave the extension of our assumption to local Lipschitzness as future work.
>
> We will add a detailed discussion on the Lipschitzness and smoothness assumptions in our limitations section. Thank you for your thoughtful suggestions and constructive feedback!
>
> ---
>
> [9] Local Lipschitz Bounds of Deep Neural Networks.
>
> [10] Gradient Descent Provably Optimizes Over-parameterized Neural Networks. ICLR 2019.
>
> [11] Gradient Descent Finds Global Minima of Deep Neural Networks. ICML 2019.
>
> [12] A Convergence Theory for Deep Learning via Over-Parameterization.ICML 2019.
>
> [13] Stability of Stochastic Gradient Descent on Nonsmooth Convex Losses. NeurIPS 2020.
>
> [14] Optimization and Generalization Guarantees for Weight Normalization
>
> [15] Full-Capacity Unitary Recurrent Neural Networks. NIPS 2016.

---

> > ### Comment · Reviewer_cwrL · 2025-08-08
> >
> > Okay, thank you for the clarification!
> > Now I see that a bounded loss function + weight decay imply uniformly bounded weights, which imply a Lipschitz constant uniform over training datasets.
> > I have no concerns for now, and will increase my final score.

---

### Official Review · Reviewer_R6uw · 2025-07-04

**Clarity:** 3
**Significance:** 3
**Originality:** 4
**Rating:** 4
**Confidence:** 5

**Summary:**

This paper proposes a novel theoretical framework to derive algorithm-dependent generalization bounds for gradient flow (GF), grounded in Rademacher complexity and a data-dependent kernel—the loss path kernel (LPK). The authors leverage the dynamics of gradient flow to build a function class characterized by the LPK and provide uniform convergence bounds with high probability. Key contributions include tighter generalization bounds than prior works, extensions to stochastic gradient flow (SGF), and experimental evidence that the derived bounds correlate with the true generalization gap. The results also recover known bounds in NTK and kernel ridge regression regimes, while demonstrating the advantages of feature learning in neural networks.

**Questions:**

1. How robust is the LPK-based generalization bound to architectural variations? The experiments mainly focus on ResNets and two-layer NNs. It would be valuable to understand if the same correlation between the bound and generalization gap holds for transformers or larger language models.

2. Can the authors extend the results to discrete-time gradient descent or SGD with fixed learning rate? The current analysis assumes gradient flow with infinitesimal steps. A more practical extension would improve relevance.

3. What is the computational complexity of evaluating the LPK term?

4. How sensitive is the bound to initialization? Since LPK integrates gradient information across the trajectory, the choice of initialization could significantly impact the generalization bound. Clarifying this would help with practical deployment.

**Ethical Concerns:**

["NO or VERY MINOR ethics concerns only"]

**Final Justification:**

After considering the rebuttal and discussion, I maintain my positive score.

Resolved issues:

1. The authors clarified that their framework extends to stochastic gradient flow and provided empirical evidence with practical learning rates, supporting relevance beyond pure gradient flow.

2. They explained that computing the loss path kernel (LPK) is tractable and can be approximated efficiently using mini-batch estimates.

3. They addressed initialization sensitivity and showed stability of the bound across architectures of varying depth (e.g., ResNet-18 vs. ResNet-34).

Unresolved issues:

1. No full discrete-time theoretical extension to SGD with finite step sizes is yet provided; this remains an avenue for future work.

2. Empirical validation is still limited to ResNets and two-layer networks; broader architectural coverage (e.g., transformers, large language models) is not demonstrated.

3. For large training times 𝑇, the bound converges to weaker known forms, limiting novelty in that regime.

Weighting:

1. I assign high weight to the theoretical novelty, integration of algorithm-dependent generalization bounds with training dynamics, and the strong empirical correlation between the bound and generalization gap in tested settings.

2. The remaining practical and empirical limitations temper the overall impact but do not diminish the significance of the core contribution.

**Limitations:**

Yes

**Quality:**

4

**Strengths And Weaknesses:**

Strengths:

1. Theoretical Novelty and Depth: The paper offers a nontrivial integration of algorithmic stability and uniform convergence through a novel kernel—the LPK—capturing the full training trajectory. This provides a compelling alternative to static kernels like NTK.

2. Unification and Generalization: The results unify classical kernel generalization bounds with modern deep learning training via gradient flow, while being applicable across convex, strongly convex, and non-convex regimes.

3. Empirical Validation: Numerical results on CIFAR-10 with ResNet and two-layer NNs convincingly support the theoretical claims. The derived bounds track the generalization gap and explain overfitting dynamics and label noise effects.

Weaknesses:

1. Limited Practical Scope: Although insightful, the theoretical analysis is limited to full-batch gradient flow with infinitesimal step size, which does not align with practical stochastic optimization scenarios in deep learning.

2. LPK Computation Complexity: Although discussed as “data-dependent,” computing LPK in practice may be computationally demanding, especially when integrating over long training trajectories.

3. When training time T is large, the asymptotic generalization bound aligns with standard bounds via Radamacher complexity analysis and poorer than that via covering number analysis.

---

> ### Author Rebuttal · Authors · 2025-07-31
>
> We thank the reviewer for the positive feedback, thoughtful review, and for appreciating the merits of the work!
>
> ---
> **Q1. The theoretical analysis is limited to full-batch gradient flow with infinitesimal step size, which does not align with practical stochastic optimization scenarios in deep learning.**
>
> A1. Although our analysis is based on gradient flow, our bound correlates well with the true generalization gap even for constant learning rates, as shown in our experiments. Also, we have the theory of stochastic gradient flow in section 5.2, which extends full-batch gradient flow to mini-batch stochastic gradient flow. In our simulation, the two-layer neural networks (NNs) and ResNets are trained with learning rates of 0.01 and 0.001, which are common learning rate setups in practice. These experimental results also show that our theory can be potentially extended to discrete SGD with a constant learning rate. We will leave this to future research work. We want to emphasize that using gradient flow to do theoretical analysis is easier and does not lose the general properties of the training dynamics for NNs. The gradient flow dynamics provides a clean, continuous-time model that captures the core features of neural-network training without sacrificing generality. Many qualitative phenomena observed under SGD (e.g., phases of feature learning, generalization plateaux, late overfitting) have been shown to persist in the gradient-flow regime. Moreover, even the training dynamic of gradient flow for two-layer NNs is still underexplored and an ongoing research direction currently, e.g., [43] ("Dynamical decoupling of generalization and overfitting in large two-layer networks") in our reference. Additionally, we want to mention that [1] shows that deep neural networks with homogeneous activations, gradient flow trajectories can be approximated by gradient descent, so we can potentially transfer our analysis of gradient flow for neural networks into a gradient descent training based on this result. We will have a further discussion of this limitation in the revision of our paper.
>
> [1] Continuous vs. Discrete Optimization of Deep Neural Networks. NeurIPS 2021.
>
>
> **Q2. Computing LPK in practice may be computationally demanding.**
>
> A2. The computation of our bound is tractable. In the $\Gamma$ term (Eq. (3)), $L_S(w_0) - L_S(w_T)$ can be easily computed from the training loss. For $\sum_{i=1}^n \int_0^T ||\nabla_w \ell(w_t, z) ||^2 dt$, we can compute the per-sample gradients (which can be computed in parallel for all the samples) at each training step, calculate their 2-norm, and sum over the training steps. The computation cost of the bound is similar to that of gradient descent training of the model. For large datasets, we can estimate $\sum_{i=1}^n \int_0^T ||\nabla_w \ell(w_t, z) ||^2 dt$ with a subset of training samples, such as the batch data used at each training step in SGD. We will add a detailed discussion of the computational complexity of evaluating the LPK term $\Gamma$.
>
>
>
> **Q3. When training time T is large, the asymptotic generalization bound aligns with standard bounds via Radamacher complexity analysis and poorer than that via covering number analysis.**
>
> A3. Standard bounds via Radamacher complexity and covering number analysis are usually based on the norm of the weight matrices [2, 3, 4, 5]. These bounds are independent of the algorithm and cannot capture the inductive bias of the learning algorithm. In contrast, our bound is data-dependent and algorithm-dependent. Empirically, our bound is also orders of magnitude tighter than [4], as shown in Fig. 2.
>
> We would appreciate it if the reviewer could clarify which Rademacher complexity bound and covering number bound they are referring to.
>
> [2] Bartlett, P. L. and Mendelson, S. Rademacher and gaussian complexities: Risk bounds and structural results. JMLR 2002.
> [3] Neyshabur, B., Tomioka, R., and Srebro, N. Norm-based capacity control in neural networks. COLT 2015.
> [4] Bartlett, P. L., Foster, D. J., and Telgarsky, M. J. Spectrally-normalized margin bounds for neural networks. NeurIPS 2017.
> [5] Neyshabur, B., Li, Z., Bhojanapalli, S., LeCun, Y., and Srebro, N. The role of over-parametrization in generalization of neural networks. ICLR 2019.
>
> **Q4. How robust is the LPK-based generalization bound to architectural variations?**
>
> A4. Thanks for the question! We will test our bounds for transformers or larger language models later. Vision transformers (ViT) satisfy our theory. For language models, we can treat each sentence as a data point $x_i$ and its corresponding response as a label $y_i$. Our analysis does not require the output dimension of the model to be one dimension. The same bound will hold for the language model and ViT. As a remark, our LPK-based generalization bound exhibits intricate and multifaceted dependencies on the dataset, training dynamics, and the architecture. In Figures 1 and 4 (Appendix A), we compare the predicted bound against the actual generalization gap for ResNet-18 versus ResNet-34 on CIFAR-10.  Despite the difference in depth and width, the bound tracks the observed gap closely for both architectures, demonstrating its quantitative stability under architectural scaling.
>
>
>
>
>
> **Q5. Extend the results to discrete-time gradient descent or SGD with fixed learning rate**
>
> A5. Our results could be extended to discrete-time GD and SGD using an idea similar to [6], by considering the taylor expansion of GD at each time step: $\ell(w_{t+1}, z) - \ell(w_{t}, z) = \nabla_w \ell(w_{t}, z)^\top (w_{t+1} - w_{t}) + O(||w_{t+1} - w_{t}||)$, where $\nabla_w \ell(w_{t}, z)^\top (w_{t+1} - w_{t})$ can be analyzed like a linear function class and $O(||w_{t+1} - w_{t}||)$ can be controlled with learning rate. We defer this to future work, as it requires additional effort and lies beyond the scope of the present study.
>
> [6] Learning Trajectories are Generalization Indicators. NeurIPS 2023.
>
>
> **Q6. Computational complexity of evaluating the LPK term**
>
> A6. As we replied in Q2, the computational complexity of computing our bound is similar to the auto-gradient in GD training of the model. In our simulation, it won’t cost too much since we can compute the gradient at each step of the GD training and compute the LPK term along the training process. We will have a detailed discussion of the computational complexity in our revised paper.
>
>
> **Q7. How sensitive is the bound to initialization?**
>
> A7. Thanks for the insightful question! The choice of initialization could indeed influence the training trajectory and the generalization bound. Our analysis assumes a fixed random initialization. The final bound does not depend directly on the initialization, but rather indirectly through the training trajectory. It would be very interesting to consider the influence of the initialization on the bound, which may require more involved analysis of optimization. Intuitively, the initialization close to the stationary points may induce a smaller bound and better generalization ability.
>
> In our case study of feature learning in Sec 6.3, we show that a mean-field type initialization enables feature learning and can obtain better sample complexity compared with NTK initialization from our generalization bound. This may indicate how initialization affects the generalization error. Our generalization bound captures this effect.
>
> ---
>
> We are happy to address any further questions and feedback.

---

> > ### Comment · Reviewer_R6uw · 2025-08-09
> >
> > Thank you for taking the time to address my questions. I appreciate your clear response, and I’m happy to keep my positive score.

---

### Decision · Program_Chairs · 2025-09-17

**Decision:**

Accept (poster)

**Comment:**

The paper investigates the theoretical generalization properties of gradient-based optimization. The authors present a generalization bound for gradient flow that parallels classical Rademacher complexity bounds known for kernel methods, with the key claimed innovation being a data-dependent kernel termed the loss path kernel (LPK). The authors claim that unlike static constructions such as the NTK, the LPK evolves with the optimization trajectory, capturing both data and training dynamics to yield sharper and more informative guarantees. The results presented by the authors emphasize the role of the gradient norms of the training loss throughout optimization in determining generalization performance. The bound recovers known kernel regression results for overparameterized neural networks, while also illustrating how neural networks enable feature learning beyond kernel methods. The authors also carry out some empirical evaluations to further support their theory.

Reviewers noted several strengths: the framework captures training trajectories in a novel way, unifies kernel-based generalization analysis with gradient dynamics, and is supported by experiments that show the proposed bounds track the generalization gap on datasets such as CIFAR with ResNets and two-layer networks. The work is also valued for its clear proofs and connections to feature learning. On the other hand the reviewers had some concerns that the analysis remains idealized, as it focuses on continuous-time gradient flow rather than the discrete SGD used in practice. Another concern was that the computation of LPK can be costly, and some of the technical assumptions, such as Lipschitzness and smoothness, may be restrictive. The authors addressed many of these concerns in their rebuttal. Overall, while some practical questions remain, the consensus is that the paper makes a meaningful theoretical advance, and I thus recommend acceptance.